# THINK THRICE BEFORE YOU ACT: PROGRESSIVE THOUGHT REFINEMENT IN LARGE LANGUAGE MODELS

**Chengyu Du**[14], **Jinyi Han**[2], **Yizhou Ying**[1], **Aili Chen**[14], **Qianyu He**[1], **Haokun Zhao**[1],

**Sirui Xia**[14], **Haoran Guo**[5], **Jiaqing Liang**[3][†], **Zulong Chen**[4], **Liangyue Li**[4], **Yanghua Xiao**[1][†]

[1] Shanghai Key Laboratory of Data Science, School of Computer Science, Fudan University
[2] Shanghai Institute of Artificial Intelligence for Education, East China Normal University
[3] School of Data Science, Fudan University   [4] Alibaba Group   [5] RhineAI
```
{cydu24, yzying24, alchen24, qyhe21, hkzhao23, srxia24}@m.fudan.edu.cn,
{liangjiaqin, shawyh}@fudan.edu.cn,{jinyihan099, rhineailab}@gmail.com,
{zulong.czl, liangyue.lly}@alibaba-inc.com
```

## ABSTRACT

Recent advancements in large language models (LLMs) have demonstrated that progressive refinement, rather than providing a single answer, results in more accurate and thoughtful outputs. However, existing methods often rely heavily on supervision signals to evaluate previous responses, making it difficult to effectively assess output quality in more open-ended scenarios. Additionally, these methods are typically designed for specific tasks, which limits their generalization to new domains. To address these limitations, we propose Progressive Thought Refinement (PTR), a framework that enables LLMs to progressively refine their responses. PTR operates in two phases: (1) Thought data construction stage: We propose a *weak and strong model collaborative selection* strategy to build a high-quality progressive refinement dataset to ensure logical consistency from thought to answers, and the answers are gradually refined in each round. (2) Thought-Mask Fine-Tuning Phase: We design a training structure to mask the "thought" and adjust loss weights to encourage LLMs to refine prior thought, teaching them to implicitly understand "how to improve" rather than "what is correct." Experimental results show that PTR significantly enhances LLM performance across ten diverse tasks (avg. from 49.6% to 53.5%) without task-specific fine-tuning. Notably, in more open-ended tasks, LLMs also demonstrate substantial improvements in the quality of responses beyond mere accuracy, suggesting that PTR truly teaches LLMs to self-improve over time. Our work is now open-source. [1]

## 1 INTRODUCTION

> "*Think thrice before you act.*"

— Confucius

Recent advancements in large language models (LLMs) have highlighted that progressive refinement is more important than simply providing a single answer (Yang et al., 2023b; Madaan et al., 2023b). Humans often rely on a combination of two thinking systems to solve problems, known as *System 1* and *System 2* (Kahneman, 2011). *System 1* facilitates quick, intuitive responses but often lacks the depth required to handle complex reasoning tasks. In contrast, *System 2* engages in progressive refinement, gradually improving a solution by starting with a rough approximate thought and iteratively adding detail and accuracy. Recent work, such as GPT-o1 (OpenAI, 2024), demonstrates that LLMs perform better by adopting progressive thought refinement. This approach leads to more accurate and thoughtfully considered outcomes, similar to how the human brain processes complex tasks.

---

[†] Corresponding authors.

[1] https://github.com/cydu24/Progressive-Thought-Refinement

Progressive refinement ability is imperative for LLMs because it significantly enhances the quality of responses by gradually improving accuracy and depth. Previous methods heavily rely on supervision signals, such as correctness assessments, to assess response quality. For example, labeled datasets with feedback are used to fine-tune models as verifiers (Han et al., 2024; Havrilla et al., 2024; Welleck et al., 2023), facilitating self-assessment and iterative improvement. Additionally, Reinforcement Learning (RL) reward functions are also employed to guide models toward generating better answers (Chen et al., 2024; Yuan et al., 2024; Rosset et al., 2024a; Akyurek et al., 2023). However, evaluating answers based on supervision signals has limitations, as annotators often **struggle to provide accurate labels** without clear, comprehensive criteria. This is particularly challenging in open-ended tasks, such as text generation and summarizing, where the distinction between "correct" and "incorrect" is blurred, making it difficult to define and evaluate response quality.

Due to significant variations in supervision signals and evaluation criteria across tasks, previous self-improvement approaches have primarily aimed to enhance accuracy within specific domains. Examples include enabling LLMs to self-debug for improved code generation (Chen et al., 2023; Tony et al., 2024; Liang et al., 2023) and solving math problems through progressive step validation (Wang et al., 2023a; Lightman et al., 2023; Uesato et al., 2022a). These methods often rely on task-specific pipelines or reward models, making generalization difficult. The key limitation is that errors addressed in one domain may not apply to other tasks, since different tasks exhibit varying error types. Consequently, transferring these approaches to new tasks often fails (Tian et al., 2024) , and models trained with these methods have **limited generalization capabilities**, struggling to improve performance beyond their training domains.

To address these challenges, we introduce PTR (**P**rogressive **T**hought **R**efinement), a framework specifically designed to stimulate the model's intrinsic refinement ability. Our PTR method comprises a progressive refinement dataset construction phase and a weighted thought-mask fine-tuning phase. During the progressive refinement dataset construction phase, we obtain queries from open-domain datasets and employ a *weak-strong model collaborative selection* strategy to construct high-quality *thoughts* and *refined answers* dataset.

This strategy not only ensures improvement from thoughts to answers but also **eliminates the need for accurate labels**. In the fine-tuning phase, we employ *weighted thought-mask fine-tuning* to teach LLMs to implicitly understand "how to improve" rather than supervising them with "what is correct". Specifically, we reformulate the masked data structure and redesign the loss of weighting to encourage LLMs to improve responses based on previous thoughts and ensuring logical consistency between the thought process and the final answer.

Our experimental results show that LLMs trained with PTR can improve the quality of their previous answers across ten tasks, including knowledge reasoning, code generation, mathematical reasoning, comprehension, summarizing, and text generation. The average performance across these tasks improved from 49.6% to 53.5%, with a significant improvement on the MMLU task, where accuracy increased from 57.1% to 64.1% for Qwen2-8B. Notably, these improvements occur **without task-specific fine-tuning**, demonstrating that our method activates the model to learn progressive refinement from the PTR dataset. Moreover, in more open-ended tasks, LLMs have also demonstrated further improvements in answer quality and formatting beyond correctness.

Our contributions are threefold:

- We propose the PTR method to stimulate models' progressive refinement abilities and enhance generalization across various tasks without additional task-specific fine-tuning.
- We design an efficient *weak-strong model collaborative selection* strategy to construct high-quality PTR datasets without extra feedback.
- We introduce a novel *weighted thought-mask fine-tuning* method to instill general progressive refinement capabilities in LLMs.

## 2 RELATED WORK

**Progressive Refinement with External Feedback** Existing work often relies on external tools or stronger LLMs to provide feedback for refinement. For example, external tools are used to critique and provide feedback on the primary model's responses (Yang et al., 2023a; Chen et al.,

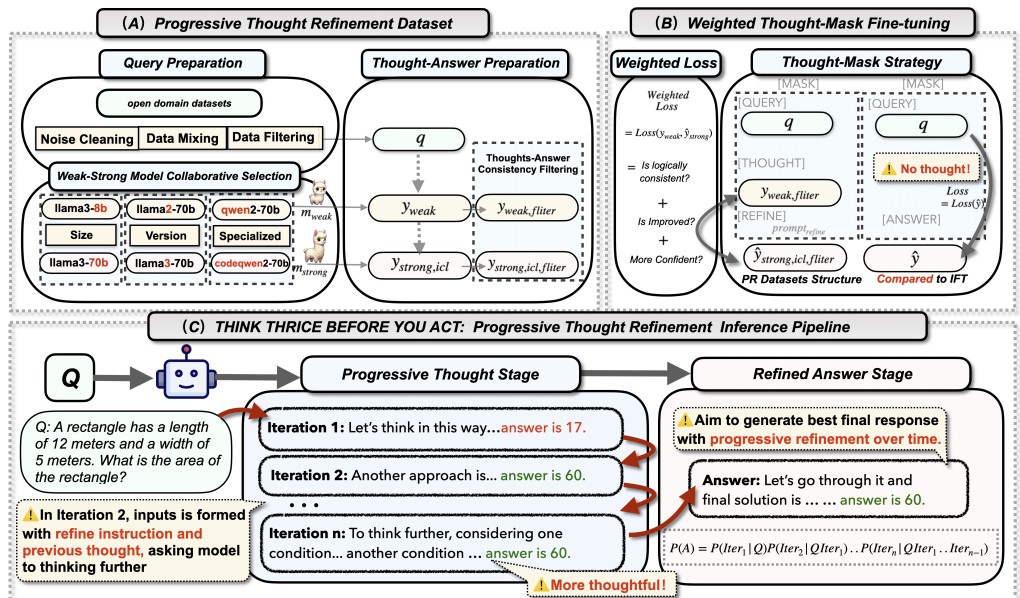

Figure 1: Illustration Our approaches. (A) Pipeline of our progressive refinement Dataset construction. We first prepare queries from the general open domain datasets, and pre-processing queries in three steps. Then we use a strong weak model collaborative selection strategy to generate thoughts and answers for each query. We also implement In-context Learning (ICL) and Consistency Filtering to ensure the quality of the thought process. (B) The illustration of Weighted Thought Masking Fine-tuning. Aiming at training the model to produce a better response in the next attempt and ensure logical consistency during the thought process. The difference between our method and IFT is that we use thought-mask techniques to ask model to generate better responses. (C) Pipeline of our PTR. Given a query $Q$, LLMs think progressively and refine their responses based on their own previous thought and refinement instruction. LLMs refined its mistakes on the second attempt, as well as gave a more thoughtful answer at a later iteration.

2023; Charalambous et al., 2023; Nijkamp et al., 2023; Yao et al., 2022; Gou et al., 2023). Models have improved their code generation capabilities by leveraging error messages from the Python interpreter (Wang et al., 2023b) and by teaching large language models to debug and explain their own code, allowing them to identify and fix errors without human feedback (Chen et al., 2023). Similarly, compiler feedback has been utilized in code generation (Chen et al., 2024; Olausson et al., 2023). Additionally, some approaches utilize criticisms or constraints generated by stronger models (Pan et al., 2023; Du et al., 2023; Bai et al., 2022; Huang et al., 2023a), such as using a strong model to verify the correctness of another model's math solutions (Wang et al., 2023a), thereby relying on external information sources to guide improvements. Although models can self-correct through external feedback (Pan et al., 2023), this approach does not fully tap into their intrinsic progressive refinement capabilities. Moreover, it requires task-specific feedback models or tools, increasing the cost of adapting to a broader range of tasks. Furthermore, current LLMs struggle to self-correct reasoning errors without external feedback (Huang et al., 2023b). Our work aims to unlock the model's inherent Progressive Refinement ability, enabling it to perform progressive refinement across all domains without relying on external tools.

**Prompting for Progressive Refinement** Various Prompting methods have been introduced to enhance Progressive Refinement, such as prompting LLMs to generate explanations and self-correct code (Li et al., 2023a), or encouraging them to generate alternative solutions and revision suggestions (Zhang et al., 2024). Some methods iteratively improve outputs by generating feedback through task-specific prompts (Madaan et al., 2023a), or guide models to generate fine-grained feedback in mathematical problem-solving, further enhancing solution accuracy and quality (Xue et al., 2023). The Reflexion method enables language models to operate effectively in specific environments by allowing them to reflect and adjust their actions when encountering errors (Shinn et al., 2023b). However, these approaches often require carefully designed, task-specific prompts or even oracle ground-truth

answers (Shinn et al., 2023a), making LLMs highly sensitive to evaluating response and achieving optimal performance (Wu et al., 2024a). Without external tools, LLMs have limited self-correction capabilities when relying solely on prompting (Huang et al., 2023b; Zheng et al., 2024).

**Fine-Tuning for Progressive Refinement** In current progressive refinement work, fine-tuning typically relies on reward models or verifiers to assess the accuracy of model outputs based on predefined criteria (Wang et al., 2023a; Lightman et al., 2023; Uesato et al., 2022a). For instance, some research focuses on improving the model's ability to identify and correct mistakes (Han et al., 2024), while others progressively validate solutions, such as in solving math problems (Uesato et al., 2022b). Additionally, reinforcement learning (RL) (Chen et al., 2024; Yuan et al., 2024; Rosset et al., 2024a; Akyurek et al., 2023) has been applied to align model outputs with correct responses. For example, researchers create preference-based datasets to align outputs with human values and reduce harmful content (Wang et al., 2024; Rosset et al., 2024b). Similarly, ROUGE has been used as a reward function in text summarizing tasks to optimize generated summaries (Akyurek et al., 2023). While these methods effectively train models, they focus on building task-specific datasets and reward functions tailored to particular objectives. In contrast, our approach redefines the fine-tuning objective to bolster the model's capacity for progressive refinement. Rather than relying on domain-specific datasets, our model is trained to iteratively enhance its responses—starting from initial thoughts and evolving toward increasingly refined answers.

## 3 PROGRESSIVE THOUGHT REFINEMENT FRAMEWORK

Our proposed framework, Progressive Thought Refinement (PTR), comprises two stages, as illustrated in Figure 1: (1) Progressive Thought Refinement Dataset Construction and (2) Progressive Weighted Thought-Mask Fine-tuning. The primary objective of this framework is to enhance models' progressive refinement abilities, enabling them to handle diverse and unfamiliar tasks without relying on task-specific fine-tuning. Since fine-tuning models for every task is impractical, our approach utilizes general queries, thoughts, and answers to help models comprehend progressive refinement. This strategy gradually improves their capacity to tackle complex tasks through progressive refinement.

### 3.1 PROGRESSIVE THOUGHT REFINEMENT DATASET CONSTRUCTION

In the first stage, we construct a progressive refinement dataset that includes Queries, Thoughts, and Answers. The Thoughts capture a sequence of different reasoning attempts, which may be varied, incomplete, or even incorrect, reflecting the model's initial exploration of the problem. In contrast, the Answers provide more confident and well-reasoned responses. This structured approach helps the model implicitly understand the difference between initial thoughts and improved answers, enabling it to generate more thoughtful and in-depth responses over time.

#### 3.1.1 QUERY PREPARATION

To enhance the model's generalization, we avoid creating domain-specific datasets. Instead, we use queries from open-domain general datasets (details in Appendix B.6), ensuring the model develops general refinement abilities rather than specializing in specific areas. Our data preprocessing involves three key steps. First, we perform data cleaning to remove noise and irrelevant content, such as images or URLs. Second, to prevent data leakage, we exclude domain-specific testing queries during training. Finally, we incorporate traditional SFT data (queries and answers) into our dataset to mitigate the risk of catastrophic forgetting.

#### 3.1.2 THOUGHT-ANSWER PREPARATION

We strategically select weak and strong models to generate sequences of thoughts and improved answers from an initial query. The objective is to ensure that the final answer is progressively improved through multiple iterations rather than relying on a single-step response. We also employ **In-Context Learning (ICL)** (Dong et al., 2024) and consistency filtering to ensure logical coherence between thoughts and answers.

**Weak-Strong Model Collaborative Selection Criteria** To ensure the final answer shows significant improvement over the initial thought sequence, we adopt a weak-strong model collaborative

selection strategy. Let $\theta_w$ and $\theta_s$ represent the abilities of the weak and strong models, respectively, with the goal of ensuring $\theta_s \gg \theta_w$. We employ three key strategies: *Model Parameter Strength*, *Model Version (New vs. Old)*, and *Domain-Specific Fine-Tuning*. These selection strategies ensure the quality of the final answer surpasses that of the previous thoughts. Additionally, we validate that the strong model performs significantly better than the weak model through Wilcoxon significance tests, as shown in Appendix B.5.

**Thought Generation by the Weak Model** The weak model generates a sequence of thoughts based on the input query $q_i$, with $\hat{y}_{i,w}^t$ representing the initial thought at the $t$-th attempt. We denote the strong model as $\pi_{\text{strong},\theta_s}$ and the weak model as $\pi_{\text{weak},\theta_w}$. These initial thoughts may contain errors but provide a foundation for further refinement:

$$S_{\text{i, thought}} = \left[ \hat{y}_{i,w}^1, \hat{y}_{i,w}^2, \ldots, \hat{y}_{i,w}^t \right] = \pi_{\text{weak},\theta_w}(\cdot \mid q_i). \tag{3.1}$$

Multiple weak models can be used to generate these thoughts, or a single weak model can produce multiple attempts. Since the weak model's thoughts need not be correct, constructing these thoughts remains cost-effective.

**Answer Refinement by the Strong Model** To achieve progressive refinement, we leverage the strong model to produce increasingly improved answers. We use ICL to ensure logical coherence between the outputs of the strong and weak models and to avoid randomness. This guides the strong model to generate better answers based on prior thoughts. Specifically, the strong model takes the sequence of thoughts $S_{\text{i, thought}}$ and query $q_i$ as input and generates the final answer $\hat{y}_{i,s,\text{icl}}$:

$$\hat{y}_{i,s,\text{icl}} = \pi_{\text{strong},\theta_s}(\cdot \mid S_{\text{i, thought}}, q_i). \tag{3.2}$$

**Thoughts-Answer Consistency Filtering** To further ensure that the thought process exhibits logical coherence, we apply consistency filtering to remove inconsistent outputs. If the consistency score is below a certain threshold, the pair is considered inconsistent and removed, ensuring that only coherent thought sequences are used for the final output (see Appendix A.1).

## 3.2 Progressive Weighted Thought-Mask Fine-tuning

In the second stage, we perform weighted thought-mask fine-tuning using the datasets constructed previously, consisting of the input query $q_i$, the initial thought sequence $S_{\text{i, thought}}$, and the final answer $\hat{y}_{i,s,\text{icl}}$. Formally, the dataset is represented as:

$$\tilde{\mathcal{D}} = \left\{ \left( q_i, S_{\text{i, thought}}, \hat{y}_{i,s,icl} \right) \right\}_{i=1}^N \tag{3.3}$$

**Thought Mask Mechanism** To help the model understand the improvement between the thought process and the answer—rather than focusing solely on the final answer—we introduce a thought mask mechanism. This mechanism selectively hides parts of the thought process during training, as shown in Figure 1 (B). It calculates the loss based only on the accuracy of the refined final answer, ensuring the model focuses on enhancing the quality of its ultimate response. Additionally, we provide refinement instructions (e.g., "Please continue thinking and refine your answer") after each thought process to prompt better refinement in subsequent iterations.

**Weighted Supervised Learning** We adopt a weighted supervised learning approach that enables the model to focus on refining its answers by progressively improving its thought process. Specifically, we perform weighted supervised learning that emphasizes both the accuracy of the final answers and the logical consistency of the thought process. The loss function optimizes the model in three key areas: generating accurate final answers, maintaining consistency in reasoning and ensuring that the model's confidence increases progressively throughout the thought process.

$$\mathcal{L}_{\text{PTR}}(\theta) = \sum_{(q_i, S_{i,\text{thought}}, y_n) \in \tilde{\mathcal{D}}} \left[ -\lambda_1 \log \Pr(y_n \mid q_i, S_{i,\text{thought}}; \theta) \right.$$
$$\left. + \lambda_2 \sum_{t=2}^n \mathcal{F}_{\text{cons}}(y_t, y_{t-1}) + \lambda_3 \sum_{t=1}^n \beta_t \left( 1 - \Pr(y_t \mid q_i, S_{i,\text{thought}}; \theta) \right) \right]. \tag{3.4}$$

Unlike standard supervised fine-tuning, which trains the model to produce a single response $\hat{y}$ given $\boldsymbol{x}$, $-\lambda_1 \log \Pr(y_n \mid q_i, S_{i,\text{thought}}; \theta)$ focuses exclusively on the accuracy of the final response generated after the thought refinement process. $\mathcal{F}_{\text{cons}}(y_t, y_{t-1})$ ensures that the current response remains logically consistent with the previous thought sequence by computing the CosineSimilarity with the Sentence-BERT (Reimers & Gurevych, 2019) model (see Appendix A.2). $\left(1 - \Pr(y_t \mid q_i, S_{i,\text{thought}}; \theta)\right)$ represents the model's uncertainty or error probability at each reasoning step, which measures the confidence of the model's predictions. The term $\beta_t$ represents the confidence score at each reasoning step, which increases as reasoning progresses to encourage higher certainty in later steps (see Appendix A.3). We also investigate sensitivity analysis and ablation study in Appendix A.5.

## 4 EXPERIMENTS

The goal of our experiments is to demonstrate the effectiveness of PTR in enabling language models to progressively enhance their responses. Specifically, we aim to answer the following questions: (1) Can the PTR method activate the model's progressive refinement ability? (2) Does our method demonstrate generalization? (3) Does progressive refinement ability emerge during training? (4) Is our method robust across different LLMs and instructions? (5) How many iterations are required for our method to achieve optimal performance?

**PTR dataset** Our model has trained on our PTR (Progressive Thought Refinement) dataset, derived queries from the WizardLM dataset (Xu et al., 2023). After thorough cleaning in Section 3.1.1, we reduced the original dataset from approximately 50k QA pairs to 40k high-quality QA pairs.

**Evaluation Tasks** In our experiments, we perform generalization over ten datasets across different tasks. For general tasks, we use MMLU (Hendrycks et al., 2020), and for coding tasks, we use HumanEval (Chen et al., 2021) (abbreviated as H-Eval). DROP (Dua et al., 2019) is used for comprehension tasks (abbreviated as Comp), and XSum (Narayan et al., 2018) is applied for summary tasks. We use GSM8K (Cobbe et al., 2021) and MATH (Hendrycks et al., 2021) for math-related tasks. For complex reasoning tasks, we use ARC and GPQA (Rein et al., 2023). For knowledge reasoning, we utilize Winogrande (Sakaguchi et al., 2019) (abbreviated as Wino) and CommonsenseQA (Talmor et al., 2019) (abbreviated as Comm).

**Evaluation Settings** We use greedy decoding (with temperature set to 0) for final generation, as lower temperature yields better performance shown in Appendix A.4. We utilize zero-shot prompting (Kojima et al., 2023) for both answer sampling and evaluations. All of our experiments are conducted on workstations equipped with eight NVIDIA A800 PCIe GPUs with 80GB memory, running Ubuntu 20.04.6 LTS and PyTorch 2.0.1.

**Baselines** We compare our model with base models and prior approaches: (1) **Prompt**: Directly prompting the model to refine its answer (Huang et al., 2023b). (2) **IFT**: Instruction Fine-Tuning by directly fine-tuning the input-output pairs from strong models on the PRD dataset to show that improvements are not due to knowledge distillation. (3) **RL**: Perform one reinforcement learning training (Wu et al., 2024b) iteration on the PRD dataset to compare with our method. Specifically, we use the thoughts and answers of the PRD dataset to construct preference data, and prefer model to produce stronger answers through DPO (Rafailov et al., 2024). We compare these methods on the PRD dataset under the same settings as in the previous section.

### 4.1 CAN THE PTR METHOD ACTIVATE THE MODEL'S PROGRESSIVE REFINEMENT ABILITY?

**PTR Activates Progressive Refinement Ability** As shown in Table 1, to emphasize the progressive refinement ability, we conduct tests on a broad range of tasks B.6. The result demonstrates that our PTR activate models substantially refine their responses across multiple iterations in the majority of tasks. For instance, in the Qwen2-7B model, accuracy on MMLU increased by 7.0%, from 57.1% (Base model Prompt Iteration 1) to 64.1% (PTR Iteration 2). On several additional tasks, PTR also showed improvements, with the average score across all tasks increasing by 3.9%-rising from 49.6% to 53.5%. However, the **Prompting** method results show that both two base models degrade in performance when asked to refine, producing worse answers compared to initial responses. These results indicate that PTR effectively enables base models to improve based on previous thoughts.

| Method | Iters | Gene. MMLU Acc | Code H-Eval Pass@1 | Comp. DROP Acc | Sum. Xsum Sim | Math GSM8k Acc | Math Acc | Reasoning ARC Acc | GPQA Acc | Knowledge Wino Acc | Comm Acc | Avg |
|---|---|---|---|---|---|---|---|---|---|---|---|---|
| | | | | | | *Qwen2-7b* | | | | | | |
| Prompt | Iter.1 | 57.1 | 56.1 | 20.9 | 47.3 | 79.1 | 48.2 | 60.6 | 24.6 | 66.8 | 55.7 | 51.6 |
| | Iter.2 | 50.1 | 37.6 | 18.7 | 43.2 | 78.4 | 47.6 | 37.9 | 22.3 | 50.4 | 42.1 | 42.8 |
| IFT | Iter.1 | 57.7 | 50.2 | 21.1 | 45.5 | 75.4 | 45.6 | 54.9 | 22.3 | 66.8 | 46.1 | 48.5 |
| | Iter.2 | 52.4 | 40.2 | 17.2 | 37.9 | 71.0 | 43.2 | 36.6 | 21.9 | 62.8 | 40.3 | 42.3 |
| RL | Iter.1 | 56.5 | 48.3 | 21.7 | 47.6 | 71.2 | 47.3 | 60.4 | 20.3 | 65.0 | 51.6 | 48.9 |
| | Iter.2 | 55.1 | 42.2 | 20.9 | 44.3 | 58.6 | 44.5 | 35.3 | 20.9 | 63.8 | 43.5 | 42.9 |
| **PTR**(our) | Iter.1 | 59.2 | 52.3 | 19.0 | 45.9 | 76.7 | 47.6 | 58.6 | 23.2 | 66.4 | 47.9 | 49.6 |
| | Iter.2 | **64.1** | 57.2 | 21.2 | **49.8** | 79.6 | 48.6 | 62.7 | **25.6** | 66.4 | 54.9 | 53.0 |
| | Iter.3 | 63.2 | **57.6** | **21.5** | 49.6 | **79.9** | **48.9** | 65.2 | 25.6 | 66.8 | 56.5 | **53.5** |
| | | | | | | *Llama3-8B* | | | | | | |
| Prompt | Iter.1 | 66.4 | 54.0 | 47.3 | 64.5 | 76.4 | 25.1 | 75.1 | 34.6 | 67.9 | 41.6 | 55.2 |
| | Iter.2 | 34.4 | 50.1 | 35.9 | 62.1 | 70.5 | 20.9 | 56.4 | 30.1 | 66.8 | 43.9 | 47.1 |
| IFT | Iter.1 | 49.1 | 38.4 | 52.8 | 47.9 | 55.4 | 21.3 | 63.0 | 34.1 | 63.3 | 37.1 | 46.2 |
| | Iter.2 | 40.4 | 34.2 | 24.7 | 42.9 | 51.1 | 18.6 | 54.4 | 28.8 | 62.2 | 28.7 | 38.5 |
| RL | Iter.1 | 51.8 | 32.4 | 46.8 | 65.9 | 61.3 | 22.3 | 67.6 | 33.7 | 62.0 | 60.1 | 50.3 |
| | Iter.2 | 39.9 | 30.2 | 40.7 | 40.9 | 57.3 | 19.3 | 55.7 | 30.6 | 53.8 | 55.8 | 42.4 |
| **PTR**(our) | Iter.1 | 59.6 | 54.0 | 49.0 | 62.4 | 76.4 | 21.3 | 73.0 | 34.1 | 68.6 | 60.0 | 55.8 |
| | Iter.2 | 68.4 | 55.2 | **49.0** | 65.7 | 79.2 | 24.7 | **77.1** | **36.2** | **70.1** | 60.5 | 58.6 |
| | Iter.3 | **68.6** | **55.4** | 48.6 | **66.1** | 79.6 | 24.9 | 77.0 | 36.1 | 67.9 | **61.3** | **58.6** |

Table 1: Main experimental results about our approach and other baselines across various domains. We experiments on two difference structures LLMs( Qwen2-7b and Llama3-8B ). We also run 2 iteration on different baselines and 3 iteration on our approach. Itertaion 1 means the first answer to the question, and we construct the format of refining instruction with the previous answer which is introduced in Method 3.2. We denoted Acc: accuracy. Pass@1:testing on code. Sim: similarity similared calculated by BGE-m3. These results suggest that our PTR excels at performing well across multiple attempts. By trading off some accuracy on the first attempt, it significantly enhances the model's ability to improve in subsequent iterations.

**PTR is not Knowledge Distillation** We also compare our PTR with simply use strong model answer to the query by IFT. We find that PTR is not equivalent to knowledge distillation. At the first iteration, we observe that when models are trained on general datasets rather than domain-specific tasks, its initial performance tends to decline at first. (This performance drop largely stems from supervised fine-tuning amplifying the initial biases of the base model. When trained on general datasets, the base models tend to accumulate biases that may not apply to specific domains, leading to poorer performance on domain-specific tasks.) However, the IFT approach fails to activate the model's progressive refinement ability and does not significantly increase the performance after the first attempts. On CommonsenceQA, The IFT approach does not perform a better response at the second iteration (40.3%) compared to their first attempt (46.1%). In contrast, PTR approach improves through iterative attempts without an approach on domain-specific knowledge. This suggests that our method is not simply **distilling knowledge** but effectively **activating** the model to refine outputs and enhance performance through self-driven iterative improvement.

**Refinement beyond Correction** Deeper analysis reveals that in open-ended tasks without clear ground truth, LLMs refine responses to be more thoughtful and comprehensive, regardless of correctness. For example, in the code task shown in Figure 2, the LLM iteratively improves its response over three iterations, considering additional aspects of the problem. This highlights PTR's ability to enhance not just correctness but also the quality and usability of outputs (Shown in Appendix D).

```
def calculation_average(numbers): # iteration 1
  total = sum(numbers)
  return total / len(numbers) # returning the output

def calculation_average(numbers): # iteration 2
  total = sum(numbers)
  return total/len(numbers) if len(numbers) > 0 else None # Adds empty lists check

def calculation_average(numbers): # iteration 3
  if not isinstance(numbers, list):
    raise TypeError("Input_must_be_a_list") # Introduces type checking
  if len(numbers) == 0: # Raises specific errors with informative messages
    raise ValueError("List_cannot_be_empty") # Adds protection against empty lists
  return sum(numbers)/len(numbers)
```

Figure 2: Code example shows PTR can refine beyond correction. The PTR goes through three rounds, providing higher quality response for each iterations. In first interation, model return with simply output. In second interation, model add more details like considering the empty list. In third interation, model structured the code and futher add type checking and errors information.

| Prompt | Iters | General | Code | Compreh. | Summary | Math | | Reasoning | | Knowledge | |
| | | MMLU | H-Eval | DROP | Xsum | GSM8k | Math | ARC | GPQA | Wino* | Comm*QA |
| | | Acc | Pass@1 | Acc | Rank | Acc | Acc | Acc | Acc | Acc | Acc |
| **Prompt1** | Iter.1 | 58.7 | 53.9 | 19.2 | 46.9 | 75.1 | 47.7 | 50.5 | 24.6 | 66.7 | 46.7 |
| | Iter.2 | 63.4 | 57.6 | 20.7 | 46.6 | 77.7 | 48.8 | 60.6 | 25.4 | 66.1 | 51.5 |
| | Iter.3 | 63.2 | 57.6 | 20.9 | 49.8 | 79.2 | 50.2 | 61.3 | 24.9 | 65.9 | 54.7 |
| | Iter.4 | 63.3 | 57.6 | 21.7 | 49.9 | 78.1 | 50.6 | 62.8 | 25.6 | 66.6 | 55.7 |
| **Prompt2** | Iter.1 | 58.7 | 53.0 | 19.2 | 46.6 | 75.1 | 47.0 | 44.5 | 25.4 | 66.7 | 46.7 |
| | Iter.2 | 63.4 | 52.8 | 22.1 | 48.9 | 77.5 | 47.2 | 61.1 | 25.4 | 68.8 | 52.1 |
| | Iter.3 | 62.7 | 57.9 | 22.5 | 49.8 | 76.8 | 47.9 | 59.1 | 25.6 | 68.2 | 50.4 |
| | Iter.4 | 62.8 | 57.6 | 22.4 | 49.6 | 77.5 | 47.8 | 60.0 | 25.8 | 67.5 | 53.1 |
| **Prompt3** | Iter.1 | 58.7 | 52.3 | 19.2 | 48.8 | 75.1 | 47.5 | 47.5 | 23.6 | 66.7 | 46.8 |
| | Iter.2 | 63.4 | 57.6 | 21.6 | 48.2 | 78.3 | 48.6 | 59.4 | 25.6 | 66.4 | 50.2 |
| | Iter.3 | 62.9 | 57.9 | 21.9 | 49.6 | 78.1 | 48.7 | 62.1 | 25.0 | 67.0 | 55.0 |
| | Iter.4 | 63.3 | 57.8 | 22.3 | 49.6 | 77.9 | 49.6 | 63.2 | 25.4 | 67.2 | 53.8 |

Table 2: Results of PTR with different prompts: (1) Assume that this thought could be either correct or incorrect. Carefully review the thought and provide a better answer. (2) Review your previous thought and assess whether it's correct. Then, provide a better response based on your answer. (3) Regardless of whether your previous thought is correct or not, provide a better answer. Iteration 1 represent the initial answer to the question. Iteration 2 to 4 represent the model's improvement over the initial answer. Notably, the model is not trained with these prompts.

## 4.2 DOES OUR METHOD DEMONSTRATE GENERALIZATION?

**PTR vs. Other Progressive Refinement methods** Unlike previous approaches, our method activates the model's inherent progressive refinement ability rather than merely boosting accuracy in specific domains. To validate PTR's generalization capability, we use datasets with general queries and evaluate whether the model can iteratively refine responses across various tasks. As seen in Table 1, our model refines responses across multiple iterations, significantly improving accuracy across tasks, and demonstrating effective generalization. We also compare PTR with other progressive refinement methods like RL to assess generalization. Our results show that methods like RL, when fine-tuned only on general-domain tasks, fails to activate iterative refinement in specialized tasks, often showing decreased accuracy. This suggests that our method is more robust in diverse environments, as it enables the model to iteratively refine its responses without being limited to domain-specific fine-tuning. By leveraging the model's inherent progressive refinement capabilities, PTR achieves consistent improvements across a wide range of tasks.

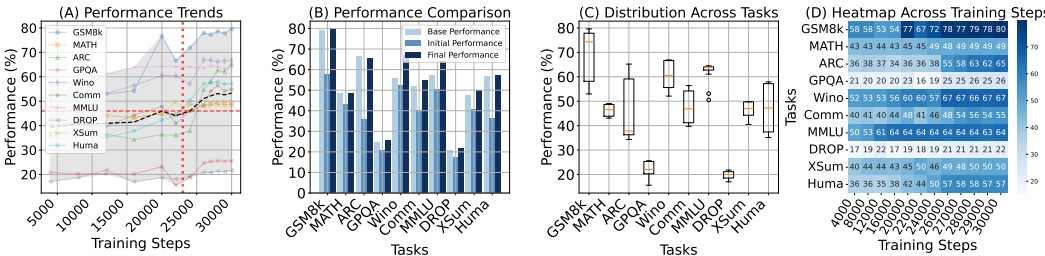

Figure 3: Plot A: Multi-line plot showing the performance trends for 10 tasks, with the average performance (in black) and variance. The overall trend is upward, with the red line highlighting the point where average performance reaches 46%. Plot B: Bar plot comparing initial, baseline, and final performance for each task. While initial performance is lower than the baseline, the final performance surpasses it, indicating that the model has learned and improved from prior iterations. Plot C: Box plot displaying the performance distribution across tasks. The varying lengths of the bars show the performance improvement, with longer bars indicating greater improvements across tasks. Plot D: Heat map representing task performance across training steps. The x-axis represents the training steps, while the y-axis represents the tasks. The color intensity indicates the model's performance at each task and training step, with deeper colors corresponding to better performance.

### 4.3 IS OUR METHOD ROBUST ACROSS DIFFERENT LLMS AND INSTRUCTIONS?

**Prompt Robustness** We also evaluated PTR robustness with different prompts and LLMs. Table 2 shows the model's performance using three different prompts across various tasks, refined over four iterations. Across all prompts, we find that PTR achieves iterative improvement across different prompts. Specifically, In the math (GSM8K) tasks, PTR is well-performed(78.1%) compared with initial responses (75.1%). On reasoning tasks (ARC), PTR see substantial improvements, especially with Prompts 1 (62.8%) and Prompts 3 (63.2%). DROP tasks also improve steadily, with accuracy increasing to 22.5% by Iteration 3 in Prompt 2. Our approach enables the model to learn from previous thoughts, rather than relying on the instruction used during training. This PTR enables the model to consistently improve its performance on different prompts, demonstrating the robustness of the PTR mechanism.

**LLMs Robustness** The table 1 also demonstrates that both Llama3-8B and Qwen2-7B exhibit robustness across different prompts and tasks. While Llama3-8B often outperforms Qwen2-7B, both models show consistent improvements with iterative refinement. This robustness ensures that PTR can be applied effectively to a wide variety of open-source LLMs.

### 4.4 DOES PROGRESSIVE REFINEMENT ABILITY EXHIBIT EMERGENCE DURING TRAINING?

**Overall Performance** Figures (A) and (B) show a clear upward trend in performance, as shown in Figure 3 shown. Notably, after 24,000 training steps (equivalent to 93 million tokens), significant improvements indicate the emergence of inference capabilities. As training continues, we observe that the average performance of PTR increases from 40.1% to 55.6%, showing an overall improvement across different tasks.

**Task Complexity and Learning Curve** We also find that tasks of varying difficulty exhibit different emergence timings and improvement rates. Plots (C) and (D) reveal that simpler tasks such as MMLU and DROP show early and steady improvements around 22,000 steps. More complex inference tasks such as ARC and GPQA exhibit delayed emergence, with ARC improving from 36.3% to 65.2% and GPQA from 23.2% to 25.6% after 24,000 steps. This shows that as training continues, the model's ability to handle complex reasoning and other tasks significantly improves, showing clear emergent behavior in different task types.

### 4.5 HOW MANY THINKING STEPS ARE REQUIRED TO ACHIEVE OPTIMAL PERFORMANCE?

We investigate how iterative thinking steps influence performance across tasks by conducting experiments over ten iterations using the Qwen2-8B model. Figure 4 illustrates performance trends.

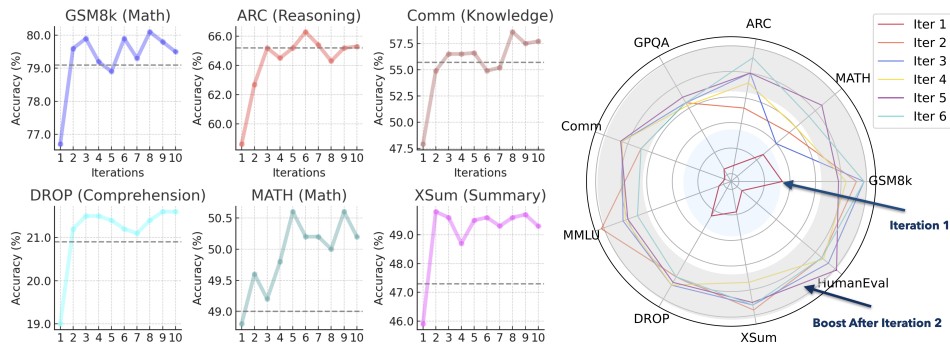

Figure 4: Performance of PTR over ten iterations across different tasks. The left plots show accuracy improvements in mathematical reasoning (GSM8K and MATH), reasoning tasks (ARC, GPQA, Winogrande, CommonsenseQA), comprehension tasks (MMLU, DROP, XSum), and coding tasks (HumanEval). More details are in Appendix B.7. The dashed line is the baseline of the model. We can see the performance of most tasks surpasses the baseline after the two or third iteration. The right plots show performance over six iterations with radar charts, In this chart, we can find the performance saturated after the first two iterations as shown in the blue areas in the figure.

**Improvements in the First Three Iterations** In the first three iterations, we saw significant improvements in model performance. In the mathematical reasoning task **GSM8K**, the accuracy improved from 75.0% in the first iteration to 79.9% in the second iteration. Similarly, the **ARC** dataset improves from 58.6% to 65.2% in the third iteration. This shows that PTR quickly refines its problem-solving through progressive refinement.

After the third iteration, the performance improvements for most tasks stabilize. In **GSM8K**, the accuracy fluctuates slightly between the third and tenth iterations, ranging from 79.9% to 80.1%. In **MATH**, the accuracy remains around 50.2% to 50.6% after reaching a peak in the second iteration. This indicates that the marginal gains decrease over time, indicating that the performance ceiling of the model is converging.

**Sustained Performance Without Overfitting** PTR performance is saturated after the first two iterations, and remains stable or improves slightly, with no notable declines. For instance, in **DROP** and **XSum**, accuracy increases from 19.0% and 45.9% to 21.6% and 49.7%, respectively, over ten iterations.

**More Computation for Hard Tasks** Complex tasks benefit more from iterative thinking and may require additional iterations for optimal performance. Accuracy in **CommonsenseQA** improves from 47.9% to 58.6% by the eighth iteration, suggesting that tasks with higher cognitive demands allow PTR to leverage iterative refinement more effectively. While **GSM8K** reaches near-optimal performance within a few iterations, tasks like **MATH** require more computation to achieve substantial gains, likely due to the challenging nature of logical reasoning involved.

## 5 CONCLUSION

We propose PTR, an approach designed to stimulate the progressive thought refinement capabilities inherent in LLMs, allowing them to improve their responses through multiple rounds of iterations. PTR adopts an annotation-free strategy to gradually build refined thoughts and answers through a weak and strong models collaborative selection process, and combines thought-answer consistency filtering to ensure logical coherence. Our weighted thought mask fine-tuning further activates the model's internal refinement ability by learning the improvement from initial thoughts to refined answers. Experimental results show that PTR simply trained with general open-domain datasets, but significantly improves the model's progressive refinement capabilities in ten different tasks, including knowledge reasoning, code generation, and mathematical reasoning, achieving a generalization level not observed by previous methods.

ACKNOWLEDGMENTS

The authors would like to express their sincere gratitude to Alibaba (China) Co., Ltd. and Alibaba Holding Aicheng Technology-Enterprise Intelligence Business Unit for their support.

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

# A METHOD

## A.1 SELF-CONSISTENCY FILTERING

In each iteration of thought generation, we apply multiple sampling techniques to generate several candidate thoughts. These candidate thoughts undergo a consistency check against the final answer to ensure logical coherence throughout the thought process.

### A.1.1 N-SAMPLING FOR THOUGHT GENERATION

For each query $q_i$, we perform **n-sampling** to generate $N$ candidate thoughts at each step of the thought generation process. These thoughts denoted as $\hat{y}_m^t$, represent the $m$-th query at the $t$-th attempt, and they collectively form the set of potential thought sequences.

To evaluate the consistency between the thought sequences and the final answer, we vectorize the thoughts and the answer using Sentence-BERT embeddings. Sentence-BERT provides an effective way to embed both the thought sequences and the final answer into a shared vector space, capturing semantic similarities between them.

## A.2 CONSISTENCY COMPUTE

To evaluate the consistency between the thought sequences and the final answer, we vectorize the thoughts and the answer using **Sentence-BERT embeddings**. Sentence-BERT provides an effective way to embed both the thought sequences and the final answer into a shared vector space, capturing semantic similarities between them.

The similarity between each thought $y_t$ and the prior thought $y_{t-1}$ is computed using the cosine similarity between their Sentence-BERT embeddings. The consistency score is designed to capture how well the current thought $y_t$ is consistent with the prior thought $y_{t-1}$ and how closely it relates to the final answer.

We define a **Consistency Function** $\mathcal{F}_{\text{cons}}(y_t, y_{t-1})$ as the cosine similarity between the current thought $y_t$ and the prior thought $y_{t-1}$. The function is computed as:

$$\mathcal{F}_{\text{cons}}(y_t, y_{t-1}) = \text{CosineSimilarity}(\text{BERT}(y_t), \text{BERT}(y_{t-1})) \tag{A.1}$$

Where $\text{CosineSimilarity}(a, b)$ is defined as:

$$\text{CosineSimilarity}(a, b) = \frac{a \cdot b}{\|a\| \|b\|} \tag{A.2}$$

Here, $a$ and $b$ are the embeddings of $y_t$ and $y_{t-1}$ respectively, generated by the Sentence-BERT model. The cosine similarity measures the angle between he two vectors in the embedding space, with values closer to 1 indicating high similarity and values closer to 0 indicating low similarity.

## A.3 DYNAMIC CONFIDENCE ADJUSTMENT:

By introducing a dynamic confidence decay strategy, the model can gradually increase its confidence during the reasoning process. For example, the confidence $\beta_t$ can increase as the reasoning step $t$ progresses, instead of staying constant throughout the process. This will allow the model to gain more confidence as it refines its thoughts and reasoning.

By adjusting the weights of each reasoning step, $\beta_t$ can control how the model's confidence evolves. At the initial steps, the model can have a lower confidence since the thought process is still being refined. As the reasoning progresses, the model should gradually increase its confidence and have higher confidence at the final step. This can be achieved by adjusting $\beta_t$ dynamically like so:

$$\beta_t = \beta_0 \cdot \left(\frac{t}{n}\right) \tag{A.3}$$

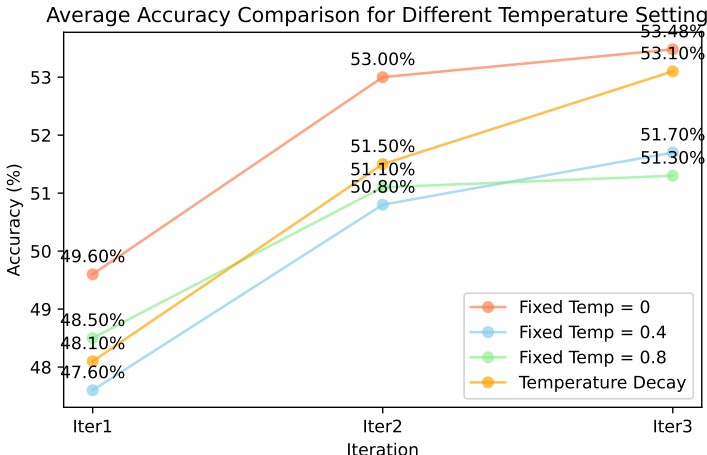

Figure 5: performance of a model under different temperature settings during inference

where $\beta_0$ is the initial confidence and $n$ is the total number of reasoning steps. This approach ensures that the confidence $\beta_t$ increases as reasoning progresses.

## A.4  TEMPERATURE ADJUSTMENT

This graph illustrates the performance of a model under different temperature settings during inference, measured over three iterations in terms of accuracy. The comparison includes four scenarios: fixed temperature at 0, the fixed temperature at 0.4, fixed temperature at 0.8, and gradually decaying temperature. The main findings are as follows: The graph clearly shows that setting the temperature to 0 yields the best performance. A temperature of 0 ensures that the model generates deterministic outputs at every step, leading to more reliable and stable results. Higher temperatures (such as 0.4 and 0.8) introduce randomness into the process, reducing overall accuracy. The decaying temperature approach improves accuracy over time but does not surpass the performance of a fixed temperature of 0.

## A.5  SENSITIVITY ANALYSIS

Table 3: Performance of different hyperparameter configurations across iterations.

| Ratio | Iteration | MMLU | H-Eval | GSM8k | ARC | Comm | Avg |
|---|---|---|---|---|---|---|---|
| 1 0 0 | 1 | 60.9% | 51.2% | 76.8% | 63.1% | 48.5% | 60.1% |
| | 2 | 62.3% | 53.0% | 76.7% | 65.2% | 52.9% | 62.0% |
| | 3 | 62.7% | 52.4% | 78.3% | 66.4% | 55.1% | 63.0% |
| 0.9 0.05 0.05 | 1 | 60.9% | 51.8% | 74.8% | 61.3% | 48.2% | 59.4% |
| | 2 | 62.6% | 55.5% | 78.4% | 64.3% | 54.1% | 63.0% |
| | 3 | 62.5% | 56.1% | 78.9% | 64.5% | 54.9% | 63.4% |
| 0.8 0.1 0.1 | 1 | 58.5% | 51.2% | 76.2% | 62.2% | 48.2% | 59.3% |
| | 2 | 63.3% | 55.5% | 78.4% | 64.3% | 54.1% | 63.1% |
| | 3 | 63.9% | 56.7% | 79.7% | 66.2% | 54.8% | 64.3% |
| 0.7 0.15 0.15 | 1 | 56.3% | 51.8% | 74.8% | 58.6% | 45.9% | 57.5% |
| | 2 | 58.3% | 52.4% | 70.7% | 64.3% | 50.6% | 59.3% |
| | 3 | 57.9% | 52.4% | 73.2% | 64.5% | 52.3% | 60.1% |
| 0.6 0.2 0.2 | 1 | 54.8% | 48.2% | 70.6% | 61.3% | 44.1% | 55.8% |
| | 2 | 55.6% | 47.6% | 71.5% | 64.3% | 50.3% | 57.9% |
| | 3 | 55.7% | 48.2% | 71.3% | 64.5% | 50.8% | 58.1% |

We conduct a sensitivity analysis to evaluate the impact of each hyperparameter. Additionally, we explore suggestions for setting these parameters to simplify the tuning process and make our method more accessible for broader applications.

**Hyperparameter Complexity:** The hyperparameters $\lambda_1$, $\lambda_2$, and $\lambda_3$ balance various aspects of the model's loss function. However, extensive tuning of these hyperparameters could limit the practical adoption of PTR. Setting $\lambda_1 = 0.8$, $\lambda_2 = 0.1$, and $\lambda_3 = 0.1$ resulted in the highest average accuracy of 64.3%, effectively balancing final answer accuracy, reasoning consistency, and confidence distribution. For broader applications, $\lambda_1 = 1.0$, $\lambda_2 = 0.0$, $\lambda_3 = 0.0$ achieved an average accuracy of 63.0%, simplifying training while maintaining competitive performance.

**Impact of low $\lambda_1$:** Setting $\lambda_1$ too low negatively impacts final answer accuracy. For instance, at $\lambda_1 = 0.6$, the model's ability to self-refine diminishes, reducing overall performance. **Roles of $\lambda_2$ and $\lambda_3$:** While small values of $\lambda_2$ and $\lambda_3$ support reasoning consistency and confidence progression, their contribution to final accuracy is limited compared to the computational cost of tuning. For tasks requiring high reasoning consistency and confidence optimization, use $\lambda_1 = 0.8$, $\lambda_2 = 0.1$, and $\lambda_3 = 0.1$. For most real-world applications, $\lambda_1 = 1.0$, $\lambda_2 = 0.0$, and $\lambda_3 = 0.0$ is sufficient.

# B  EXPERIMENT

## B.1  SETTINGS

Table 4: Hyperparameter Settings for Training and Inference

| Hyperparameter/Description | Training Values | Inference |
|---|---|---|
| `bf16` | TRUE | N/A |
| `epochs` | 2 | N/A |
| `per device train batch size` | 1 | N/A |
| `gpus` | 2xH8100 | 2xH800 |
| `gradient accumulation steps` | 256 | N/A |
| `learning rate` | 5e-5 | N/A |
| `weight decay` | 0 | N/A |
| `warmup step` | 1000 | N/A |
| `learning rate scheduler type` | cosine | N/A |
| `model max length` | 2048 | 2048 |
| `temperature` | N/A | 0 |
| `top_p` | N/A | 1 |
| `max_new_tokens` | N/A | 1000 |

## B.2  ITERATION RESULT

| Iterations | Math | Code | Reasoning | Comprehension | Overall Avg. |
|---|---|---|---|---|---|
| Baseline | 62.75 | 52.3 | 49.03 | 41.37 | 49.86 |
| Iteration 1 | 65.10 (+2.35) | 57.2 (+4.9) | 51.23 (+2.20) | 45.03 (+3.66) | 53.13 (+3.33) |
| Iteration 2 | 65.55 (+0.45) | 57.6 (+0.4) | 52.12 (+0.89) | 45.57 (+0.54) | 53.65 (+0.52) |
| Iteration 3 | 65.00 (-0.55) | 57.3 (-0.3) | 52.06 (-0.06) | 45.10 (-0.47) | 53.33 (-0.32) |
| Iteration 4 | 64.75 (-0.25) | 58.1 (+0.8) | 52.08 (+0.02) | 45.46 (+0.36) | 53.54 (+0.21) |
| Iteration 5 | 65.05 (+0.30) | 57.2 (-0.9) | 52.05 (-0.03) | 45.03 (-0.43) | 53.34 (-0.20) |
| Iteration 6 | 64.75 (-0.30) | 57.6 (+0.4) | 51.96 (-0.09) | 45.03 (+0.00) | 53.17 (-0.17) |
| Iteration 7 | 65.05 (+0.30) | 57.9 (+0.3) | 52.00 (+0.04) | 45.37 (+0.34) | 53.65 (+0.48) |
| Iteration 8 | 65.20 (+0.15) | 57.9 (+0.0) | 52.05 (+0.05) | 45.40 (+0.03) | 53.70 (+0.05) |
| Iteration 9 | 65.00 (-0.20) | 57.4 (-0.5) | 52.09 (+0.04) | 45.03 (-0.37) | 53.70 (+0.00) |
| Iteration 10 | 65.00 (+0.00) | 57.4 (+0.0) | 52.09 (+0.00) | 45.03 (+0.00) | 53.62 (-0.08) |

Table 5: Averages for Math, Code, Reasoning, and Comprehension datasets over ten iterations, with colored improvements and declines.

| Iterations | GSM8k | MATH | ARC | GPQA | Winogrande |
|---|---|---|---|---|---|
| Baseline | 76.7 | 48.8 | 58.6 | 23.2 | 66.4 |
| Iteration 1 | 79.6 (+2.9) | 50.6 (+1.8) | 62.7 (+4.1) | 25.6 (+2.4) | 65.6 (-0.8) |
| Iteration 2 | 79.9 (+0.3) | 51.2 (+0.6) | 65.2 (+2.5) | 25.6 (+0.0) | 66.2 (+0.6) |
| Iteration 3 | 79.2 (-0.7) | 50.8 (-0.4) | 64.5 (-0.7) | 25.5 (-0.1) | 66.2 (+0.0) |
| Iteration 4 | 78.9 (-0.3) | 50.6 (-0.2) | 65.2 (+0.7) | 25.8 (+0.3) | 66.3 (+0.1) |
| Iteration 5 | 79.9 (+1.0) | 50.2 (-0.4) | 66.3 (+1.1) | 25.6 (-0.2) | 65.9 (-0.4) |
| Iteration 6 | 79.3 (-0.6) | 50.2 (+0.0) | 65.4 (-0.9) | 25.3 (-0.3) | 65.8 (-0.1) |
| Iteration 7 | 80.1 (+0.8) | 50.0 (-0.2) | 64.3 (-1.1) | 24.9 (-0.4) | 66.0 (-0.2) |
| Iteration 8 | 79.8 (-0.3) | 50.6 (+0.6) | 65.2 (+0.9) | 25.2 (+0.3) | 66.2 (+0.0) |
| Iteration 9 | 79.5 (-0.3) | 50.2 (-0.4) | 65.3 (-0.2) | 25.4 (+0.2) | 66.3 (+0.1) |
| Iteration 10 | 79.5 (+0.0) | 50.2 (+0.0) | 65.3 (+0.0) | 25.4 (+0.0) | 66.3 (+0.0) |

| Iterations | CommonsenseQA | MMLU | DROP | XSum | HumanEval |
|---|---|---|---|---|---|
| Baseline | 47.9 | 59.2 | 19.0 | 45.9 | 52.3 |
| Iteration 1 | 54.9 (+7.0) | 64.1 (+4.9) | 21.2 (+2.2) | 49.8 (+3.9) | 57.2 (+4.9) |
| Iteration 2 | 56.5 (+1.6) | 63.2 (-0.9) | 21.5 (+0.3) | 49.6 (+0.2) | 57.6 (+0.4) |
| Iteration 3 | 56.5 (+0.0) | 63.1 (-0.1) | 21.5 (+0.0) | 48.7 (-0.9) | 57.3 (-0.3) |
| Iteration 4 | 56.6 (+0.1) | 63.0 (-0.1) | 21.4 (-0.1) | 49.5 (+0.8) | 58.1 (+0.8) |
| Iteration 5 | 54.9 (-1.7) | 62.6 (-0.4) | 21.2 (-0.2) | 49.6 (+0.0) | 57.2 (-0.9) |
| Iteration 6 | 55.2 (+0.3) | 62.5 (-0.1) | 21.1 (-0.1) | 49.3 (-0.3) | 57.6 (+0.4) |
| Iteration 7 | 58.6 (+3.4) | 63.1 (+0.6) | 21.4 (+0.3) | 49.6 (+0.0) | 57.9 (+0.3) |
| Iteration 8 | 57.5 (-1.1) | 63.3 (+0.2) | 21.6 (+0.2) | 49.7 (+0.1) | 57.9 (+0.0) |
| Iteration 9 | 57.7 (+0.2) | 63.5 (+0.2) | 21.6 (+0.0) | 49.3 (-0.4) | 57.4 (-0.5) |
| Iteration 10 | 57.7 (+0.0) | 63.5 (+0.0) | 21.6 (+0.0) | 49.3 (+0.0) | 57.4 (+0.0) |

Table 6: Results across ten iterations for different datasets, with improvements and declines.

## B.3 EFFECTIVENESS OF THE THOUGHT-MASK STRATEGY

We conducted experiments with and without the masking strategy, and the results clearly showed that the mask improves performance. The thought-mask guides the model's attention during training, helping it refine answers based on prior reasoning steps. Without the mask, the model tends to compute immature or incorrect intermediate thoughts, leading to worse initial responses.

The IFT method removes the thought process and directly fine-tunes the input with the strong model's answer. This approach demonstrates that our data construction format is not merely a distillation of the strong model's abilities but instead successfully triggers the model's self-refinement capabilities. Basic distillation does not yield significant improvements on specific tasks or enable continuous self-improvement, further validating the effectiveness of our method.

Table 7: Performance Comparison with and without Thought-Mask Strategy

| Method | MMLU | H-Eval | DROP | Xsum | GSM8k | Math | ARC | GPQA | Wino | Comm | AVE |
|---|---|---|---|---|---|---|---|---|---|---|---|
| UnMask | 49.9% | 48.8% | 17.2% | 41.1% | 67.4% | 42.1% | 56.3% | 19.4% | 61.8% | 45.1% | 44.9% |
| | 55.1% | 49.4% | 19.5% | 40.9% | 71.1% | 43.8% | 62.9% | 20.8% | 58.4% | 50.3% | 47.2% |
| Mask | 59.2% | 52.4% | 19.0% | 45.9% | 76.7% | 47.6% | 58.6% | 23.2% | 66.4% | 47.9% | 49.7% |
| | 64.1% | 57.2% | 21.2% | 49.8% | 79.9% | 48.6% | 62.7% | 25.6% | 66.4% | 54.9% | 53.0% |

## B.4 COMPARISON WITH PRIOR WORKS ON REFINEMENT METHODS OR APPROACHES WITH VERIFIERS

### B.4.1 REFINEMENT METHODS OVERVIEW

**Self-refine** Madaan et al. (2023a) Self-refine does not require training but relies on standard answers to assist reasoning. It uses specific math-refine prompts to guide a base model in critiquing and revising its mistakes.

Table 8: Performance Comparison on GSM8K and MATH

| Method | GSM8K | MATH |
|---|---|---|
| Self-refine iteration1 | 79.1 | 48.7 |
| With ground truth iteration2 | 81.3 | 50.1 |
| Without ground truth iteration2 | 74.7 | 48.4 |
| Pair Self Correction iteration1 | 77.7 | 48.2 |
| Pair Self Correction iteration2 | 80.1 | 49.5 |
| Reward-model Verifier iteration1 | 80.0 | 48.1 |
| Reward-model Verifier iteration2 | 81.7 | 49.2 |
| Ours iteration1 | 76.7 | 47.6 |
| Ours iteration2 | 79.9 | 48.9 |

- *With ground truth*: The model checks the correctness of its response only if the initial answer is wrong. If incorrect, it generates a new response in a second iteration and stops as soon as the correct answer is predicted.

- *Without ground truth*: The model always refines its answer without verifying correctness.

**Reward Model Verifier (ORM)** Cobbe et al. (2021) This method requires training a reward model (verifier) and using it during inference to evaluate and refine answers. First, the reward model is trained using a best-of-n strategy to construct a mathematical dataset. For a given problem, 10 candidate answers are generated, and the reward model scores these answers based on correctness. The highest-scoring answers are labeled correct, while others are labeled incorrect. During inference, the reward model evaluates the responses iteratively, refining incorrect ones until the final output is most likely correct.

**Pair Self-Corrective** Welleck et al. (2022) This method trains a single model with self-diagnosis and generation capabilities. It fine-tunes a large model using pairs of correct and incorrect solutions, enabling it to learn to correct mistakes. If the model's self-diagnosis determines the output is correct, no changes are made; if incorrect, the model revises the response.

Experimental results show that performance improvements across methods are modest, typically within 1-3%. The self-refine approach relies on ground-truth feedback for slight gains; without it, performance often deteriorates. Similarly, Pair Self-Correction and Reward-Model Verifier achieve comparable improvements but remain limited.

Our method achieves similar improvements and demonstrates effectiveness in non-mathematical and non-reasoning tasks where other approaches struggle due to challenges in dataset construction. Unlike the self-refine structure, which relies heavily on external guidance, our model consistently improves performance without needing ground truth, showcasing its broader applicability and robustness.

## B.5 WILCOXON SIGNED-RANK TEST

In this experiment, we analyzed samples across three dimensions: **model parameter strength** and **model version (new vs. old)**, and **domain-specific fine-tuning**. Using the Wilcoxon signed-rank test, we assessed the differences in inference quality between the strong and weak models across these dimensions to verify whether the strong model provides significant improvements. We use human experts and Auto-j Li et al. (2023b) to judge the quality of the generated responses.

To visually present the score differences across the three dimensions, we plotted a **distribution of inference score differences** (see Figure 6). The box plot displays score differences in the following three dimensions:

- **Model Parameter Strength**: Differences in inference quality between models with strong parameters and weak parameters.

- **Model Version (New vs. Old)**: Score differences comparing the performance of new and old versions of the model.

- **Domain-Specific Fine-Tuning**: Score differences between models that have undergone domain-specific fine-tuning and those that have not. In this work, we simply not using this criteria, since the open-domian datasets are relatively various from tasks. However, it can be used in future work.

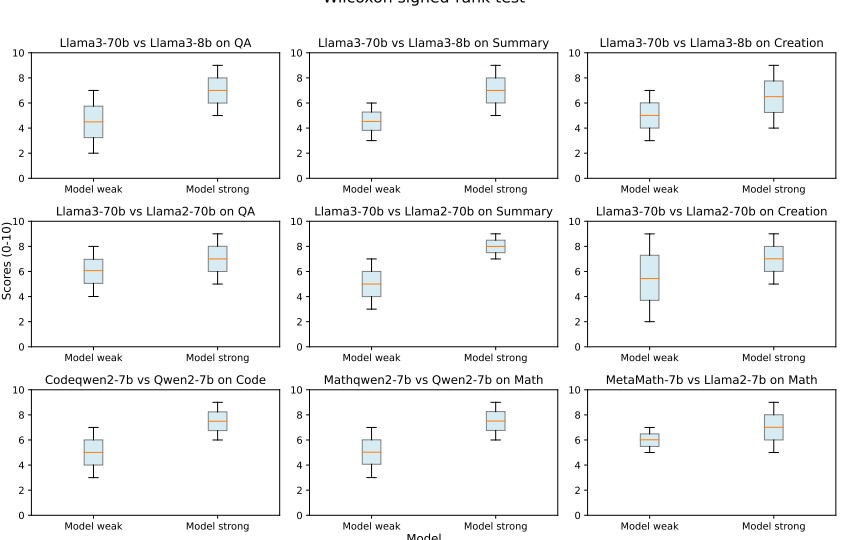

Figure 6: Boxplot of score differences across model parameters, model version, and fine-tuning. The boxplots generated from the data compare two sets of models: weaker models (denoted as "Model weak") and stronger models (denoted as "Model strong") across nine different tasks. The comparisons involve different models such as Llama3-70b vs. Llama3-8b, Llama3-70b vs. Llama2-70b, and Codeqwen2-7b vs. Qwen2-7b on tasks like QA, Summary, Creation, Math, and Code.

While the boxplots provide a visual confirmation that Model Strong outperforms Model weak across all tasks, a Wilcoxon signed-rank test can further confirm these results statistically. Based on the boxplots, we would expect the p-values from this test to be significantly less than 0.05, indicating that the differences in performance between Model Weak and Model strong are statistically significant.

Table 9: Wilcoxon Signed-Rank Test Results for Model Comparisons

| Task | Weak Model | Strong Model | p-value | Significance | Sample Size | z-score |
|---|---|---|---|---|---|---|
| QA | Llama3-8b | Llama3-70b | < 0.05 | Significant | 100 | 22.96 |
| Summary | Llama3-8b | Llama3-70b | < 0.05 | Significant | 100 | 20.35 |
| Creation | Llama3-8b | Llama3-70b | < 0.05 | Significant | 100 | 21.85 |
| QA | Llama2-70b | Llama3-70b | < 0.05 | Significant | 110 | 19.24 |
| Summary | Llama2-70b | Llama3-70b | < 0.05 | Significant | 110 | 18.76 |
| Creation | Llama2-70b | Llama3-70b | < 0.05 | Significant | 110 | 19.57 |
| Code | Qwen2-7b | Codeqwen2-7b | < 0.05 | Significant | 120 | 23.67 |
| Math | Qwen2-7b | Mathqwen2-7b | < 0.05 | Significant | 130 | 21.43 |
| Math | Llama2-7b | MetaMath-7b | < 0.05 | Significant | 130 | 22.02 |

We find that the Wilcoxon signed-rank test confirms our previous results. 1) **Larger model sizes** (e.g., Llama3-70b) consistently outperform smaller models across a variety of tasks. 2) **Fine-tuning for specific domains** (such as coding or math) provides significant performance improvements. 3) **newer model versions** (e.g., Llama3 vs. Llama2) yield better results, though the improvements are generally smaller compared to model size differences.

The analysis of the boxplots clearly demonstrates that stronger models significantly outperform their weaker counterparts across all tasks. These findings suggest that both model size and fine-tuning for specific domains play crucial roles in improving model performance. The Wilcoxon signed-rank test,

if conducted, is expected to support these visual findings, confirming the statistical significance of the observed differences.

### B.6 SETTING DETAILS

**Open-domain Datasets**

- **WizardLM** Xu et al. (2023) is an instruction dataset built with the EVOL-INSTRUCT method. EVOL-INSTRUCT utilizes CHATGPT to augment the complexity of the same queries in Alpaca and ShareGPT. [2]

**Data Filtering**    In this section, we provide details about the open-domain datasets used for query preparation. These datasets were chosen for their generalizability and diversity of content, ensuring the model is exposed to a wide range of topics and query types. Our selection process was guided by the following criteria:

- **Data Cleaning Pipeline**: The cleaning process involved removing noise such as HTML tags, non-alphanumeric characters, and duplicate entries. We applied frequency-based filtering to exclude long-tail queries and low-frequency phrases that are unlikely to contribute to the model's refinement abilities.

- **Final Dataset Size**: After applying all filtering and cleaning steps, the final dataset consisted of approximately 40k high-quality, open-domain query-answer pairs.

**Eval Tasks Choice**    We deploy a benchmark to evaluate whether our approach can activate the model's progressive refinement capabilities, enabling it to think and iterate across various dimensions. This comprehensive benchmark encompasses eight categories and ten tasks, rigorously assessing language models on multiple dimensions including basic perception, mathematics, coding, summarization, continuation, question answering, and experimentation.

In our experiments, we utilized ten widely recognized and diverse datasets from various domains to comprehensively cover different skills and abilities. For general cognitive abilities, we used the MMLU dataset Hendrycks et al. (2020), which spans tasks from junior high to professional exams. Code comprehension and problem-solving were evaluated using the HumanEval dataset Chen et al. (2021), while reading comprehension and reasoning were assessed through the DROP dataset Dua et al. (2019). The XSum dataset Narayan et al. (2018) was used for summarization tasks, and mathematical reasoning was tested using the MATH Hendrycks et al. (2021) and GSM8K Cobbe et al. (2021) datasets. Complex reasoning was evaluated with the GPQA dataset Rein et al. (2023). For knowledge representation and common-sense reasoning, we utilized Winogrande Sakaguchi et al. (2019) and CommonsenseQA Talmor et al. (2019). Finally, creative reasoning was tested using the HellaSwag dataset Zellers et al. (2019). Unlike other refinement approaches Wang et al. (2024), we do not partition the evaluation datasets for fine-tuning. Instead, we perform fine-tuning on general domain data. To verify the generalization of the model's progressive refinement capabilities, we evaluate it on 10 unseen evaluation datasets.

**Metrics**    In our evaluation framework, for objective questions, we assess answer correctness using the Accuracy metric. For coding problems, we employ the pass@1 metric to gauge the effectiveness of solutions. For subjective questions, we utilize GPT-4 for initial analysis and scoring, supplemented by expert evaluation to ensure a comprehensive assessment. This approach emphasizes a multi-dimensional evaluation of responses, focusing not only on correctness but also on quality and depth of insight.

---

[2]We use the WizardLM in https://huggingface.co/datasets/WizardLM/WizardLM_evol_instruct_V2_196k.

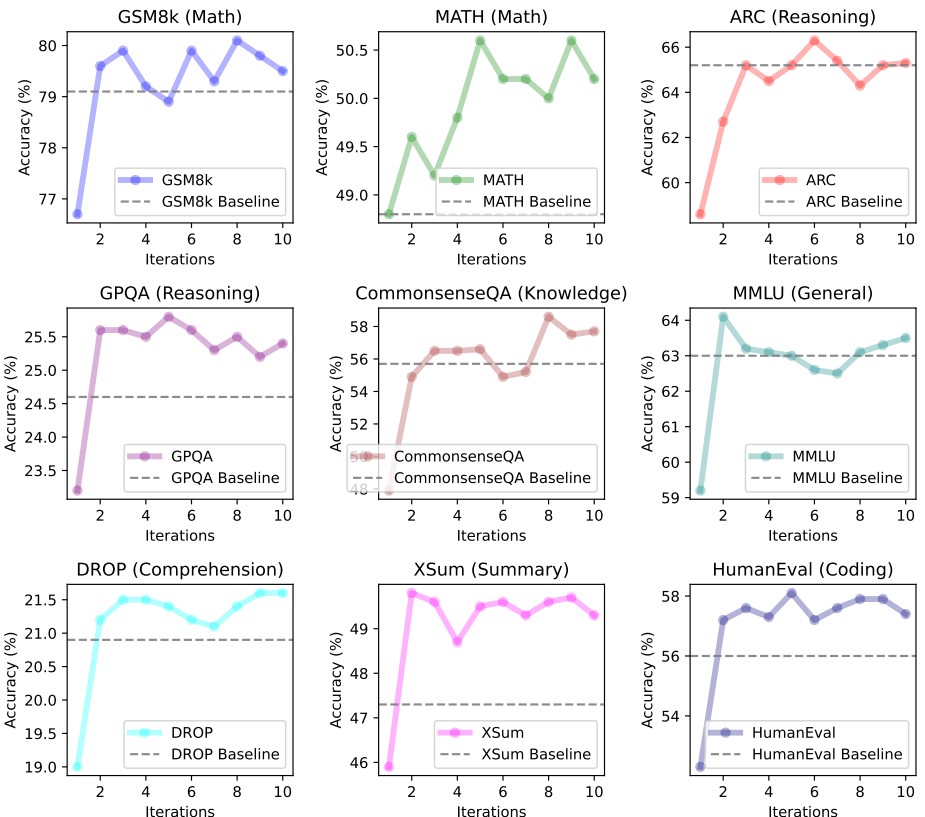

Figure 7: Performance of PTR over ten iterations across different tasks. The plots show accuracy improvements in mathematical reasoning (GSM8k and MATH), reasoning tasks (ARC, GPQA, Winogrande, CommonsenseQA), comprehension tasks (MMLU, DROP, XSum), and coding tasks (HumanEval). Baseline performance is indicated by dashed lines.

## B.7 ITERATION

## C PROMPTS

### C.1 PROMPTS FOR PROGRESSIVE REFINEMENT

PTRenables LLMs to progressive refinement, asks LLMs to improve responses without any supervision.

```
[INPUT]

Explain how reinforcement learning works in machine learning.

[OUTPUT1]

Reinforcement learning (RL) is a type of machine learning where
an agent learns to make decisions by performing actions in an
environment to maximize some notion of cumulative reward.  The
agent receives feedback in the form of rewards or punishments
based on the actions it takes, and over time, it learns a
strategy, or policy, to make better decisions.

[REFINE PROMPT]

Please further think about and give me a more precise and
professional answer.
```

```
[OUTPUT2]

Reinforcement learning (RL) is a type of machine learning where an
agent learns through interactions with an environment.  A classic
example is teaching a robot to navigate a maze.  The robot, acting
as the agent, makes decisions at each step--whether to go left,
right, forward, or backward.  After each move, the robot gets
feedback:  a reward for moving closer to the exit or a penalty
for getting stuck or moving further away.  Over time, it learns to
take actions that maximize its reward, thereby finding the optimal
path through the maze.
```

## D  CASE STUDY

In this section, we present a comprehensive case study analysis of the output generated by various language models on several real-world tasks. Each case is evaluated based on the quality of responses, relevance to the question, and overall utility for the end-user by GPT-4. These cases were randomly selected from our evaluation results.

We have divided the analysis into six tasks: flight time estimation, web browser usage statistics, language comparison, REST API implementation, JavaScript list item addition, and calculating averages in Python. These real-world tasks are often **challenging for humans to label accurately**. However, we observed significant improvements from Iteration 1 to Iteration 4, with the most optimal performance appearing around Iterations 2 and 3.

### D.1  TASK 1: FLIGHT TIME ESTIMATION

**Question**: *How long does it take from Hangzhou to New York by air?*

This task evaluates the quality of responses related to estimating flight times from Hangzhou to New York. The different outputs showcase varying levels of accuracy and relevance:

**Outputs**

- **Iteration 1** It depends on various factors such as the type of airplane and weather conditions. However, the estimated flight time from Hangzhou to New York typically ranges from 14 to 16 hours.

- **Iteration 2** The flight time from Hangzhou to New York is approximately 15 hours and 30 minutes. Major airlines operating on this route include China Eastern and Delta Airlines. Keep in mind that additional time for security checks, boarding, and customs at both airports should also be considered.

- **Iteration 3** The direct flight time from Hangzhou to New York is approximately 15 hours and 30 minutes. Airlines like China Eastern and Delta operate this route. Keep in mind that the total travel time may vary depending on factors such as layovers, flight delays, and weather conditions. Additionally, you should plan for extra time for security checks, boarding, and customs, which can add another 1 to 2 hours at each airport. It's advisable to check for updated flight schedules and potential stopover options if a direct flight is unavailable.

- **Iteration 4** The flight on December 2, 2021, from Hangzhou to New York took about 15 hours with a stopover in Tokyo. Depending on the route and the specific date, direct flights can be faster, but you should plan for various connection options.

**Analysis**

- **Iteration 1** provides a general estimate but introduces unnecessary ambiguity by discussing different airplane types without focusing on direct flights.

- **Iteration 2** is the most accurate, offering a specific duration of 15 hours and 30 minutes, while also considering practical factors such as airport procedures and layovers.

- **Iteration 3** The improved answer not only provides a specific flight duration but also mentions the airlines that operate the route. It further addresses factors that might affect the travel time, such as delays and weather conditions, which enhances the answer's accuracy and reliability.

- **Iteration 4** includes complex and irrelevant details regarding connecting flights, which are not pertinent to the user's query.

**Conclusion**: Overall, **Iteration 3** provides the most relevant and helpful information by giving a clear estimate and addressing additional travel concerns. Previous outputs either lack focus or introduce unnecessary complexity.

### D.2    TASK 2: BROWSER USAGE STATISTICS

**Question**: *How long does it take from Hangzhou to New York by air?*

**Outputs**

- **Output 1**: "As of December 2021, Google Chrome is the most widely used browser in the world with a market share of around 63.47%. This number can change over time as different surveys and data sources may have slightly different results due to varying methodologies and time frames for gathering statistics. It's always best to check recent reliable sources like StatCounter or SimilarWeb for up-to-date information on web traffic trends."

- **Output 2**: "As of now (December 2021), according to analytics tools such as StatCounter, Google Chrome is currently the most commonly used browser across the globe. It has a dominant market share of approximately 65%, followed by Safari at around 18%, Microsoft Edge at nearly 4%, Firefox at just under 3%, and Internet Explorer/Windows Web Experience Platform (IE) accounting for less than 1%. However, it's important to note that these figures are constantly changing as new updates and versions become available."

- **Output 3**: "Based on current estimates, Google Chrome is the most popular browser globally. According to W3Techs, as of September 2021, Google Chrome holds the highest market share among users aged 5+ in over 190 countries worldwide. This means that out of all people who use a desktop computer and are five years old or above, more likely than not they will be using Google Chrome as their preferred web browser. However, please remember that internet usage patterns can vary greatly depending on various factors such as region, device type, age groups within a country, etc., so these numbers could fluctuate compared to other data sources or specific timeframes."

- **Output 4**: "Google Chrome is currently the most widely used browser across the globe based on recent statistical analysis from reputable tech firms such as W3Techs. I strive to provide accurate and up-to-date information, but please remember that technology and user behavior can rapidly evolve. For the most current stats, I would recommend checking websites specializing in web analytics."

**Analysis**

- **Output 1**:
  - **Strengths**: Provides a clear and concise answer: Google Chrome is the most used browser with a 63.47% market share as of December 2021. Mentions the importance of checking up-to-date sources like StatCounter for changes.
  - **Weaknesses**: Lacks depth regarding other browsers' market shares. Could benefit from more contextual information about the browser landscape.

- **Output 2**:
  - **Strengths**: Adds more detail by including market share figures for other major browsers (Safari, Microsoft Edge, Firefox, etc.), offering a clearer picture of the browser landscape. Highlights that these figures are constantly changing, adding a more realistic and dynamic perspective.
  - **Weaknesses**: Could further elaborate on the regional or device-based differences in browser usage patterns.

- **Output 3**:
  - **Strengths**: Provides not only market share data but also adds important details about user demographics, such as the age group (5+ years) and countries covered. Discusses how internet usage patterns may vary based on factors like region and device type, adding nuance to the global market share data.
  - **Weaknesses**: While detailed, the added complexity might overwhelm users looking for a simple answer. The time frame (September 2021) is slightly outdated compared to the other outputs.
- **Output 4**:
  - **Strengths**: Confirms the user's query about Google Chrome's dominance and emphasizes the importance of checking up-to-date sources due to the rapidly changing nature of technology. Simple and engaging, aimed at building rapport with the user.
  - **Weaknesses**: Doesn't provide specific market share percentages or as much detailed breakdown as earlier iterations. Leans more towards conversational tone rather than providing a comprehensive analysis.

**Conclusion**:

**Iteration 3** provides the most detailed and comprehensive answer by incorporating specific data, user behavior insights, and a global perspective. This iteration strikes a good balance between accuracy, depth, and context, making it the most informative response.

### D.3    TASK 3: LANGUAGE COMPARISON (C++ VS. PYTHON)

**Question**: *"Please act as a college student majoring in computer science to answer my question. What's the difference between C++ and Python?"*

**Outputs**

- **Output 1**: "Hello! I'd be happy to help you understand the differences between C++ and Python programming languages. Both are powerful tools for building software applications, but they have some key differences:
  1. **Syntax**: C++ has a more complex syntax than Python, with many reserved words that must be used exactly as written (e.g., int, if, while). In contrast, Python uses indentation to define blocks of code instead of curly braces ( or []), making its syntax simpler and easier to read. For example, here's equivalent code snippets in both languages for calculating the area of a circle:
  - C++: #include <iostream>...
  - Python:..."
- **Output 2**: "Sure, I'd be happy to explain the difference between C++ and Python!
  C++ is a low-level language that provides direct control over system resources such as memory management. It was designed for systems programming and game development. It supports object-oriented programming features like inheritance, encapsulation, polymorphism, etc. However, it requires careful handling of memory which can lead to bugs if not done correctly.
  Python, on the other hand, is a high-level language that abstracts away most of these low-level details, allowing programmers to focus more on the logic of their programs rather than managing memory themselves. Python also has extensive libraries and frameworks, such as numpy, scikit-learn, and pandas, which make data analysis much faster and easier compared to writing everything from scratch in C++.
  In summary, while both languages serve different purposes, Python is generally considered a safer choice for beginners due to its simplicity and ease of use."
- **Output 3**: "Here's a brief rundown of the differences between C++ and Python:
  - **Similarities**: Both are general-purpose, compiled languages. They offer strong typing and object-oriented design, and they provide rich standard libraries.
  - **Differences**: C++ is faster due to its lower-level access to system resources, but Python is easier for beginners with its simple syntax and robust libraries for tasks like data analysis and web development."

- **Output 4**: "C++ is best suited for systems programming and game development where fine-grained control over system resources is necessary. Its low-level memory management allows for better customizability but comes at the expense of higher complexity. On the other hand, Python offers a simplified approach with high readability for beginners, excelling in web development and data science due to its comprehensive libraries. In performance tests, C++ often outperforms Python, particularly for computationally expensive tasks."

**Analysis**

- **Output 1**:
  - **Strengths**: Provides a simple comparison of syntax differences, using concrete code examples, making it accessible and easy to understand for beginners.
  - **Weaknesses**: Only focuses on syntax differences, without addressing broader usage contexts or the strengths and weaknesses of each language in real-world applications.
- **Output 2**:
  - **Strengths**: Delivers a more in-depth comparison, highlighting differences in functionality and application areas, such as systems programming for C++ and data analysis for Python.
  - **Weaknesses**: Could provide more real-world examples or performance comparisons to give users a clearer understanding of when to use each language.
- **Output 3**:
  - **Strengths**: Covers similarities and differences between the languages concisely, giving a balanced overview of their strengths.
  - **Weaknesses**: Introduces too many complex features, potentially overwhelming the reader with excessive technical details.
- **Output 4**:
  - **Strengths**: Combines the best of both approaches, discussing performance differences and suitable applications for each language, along with real-world use cases.
  - **Weaknesses**: The output becomes too dense and may lose focus due to the volume of information provided.

**Overall Analysis**

- **Iteration 1** provides a simple and clear comparison that is easy to follow but lacks the depth needed for more advanced readers.
- **Iteration 2** goes deeper into functional differences, making it a solid choice for those looking to understand where each language is best applied.
- **Iteration 3** tries to offer a more comprehensive comparison but overcomplicates the explanation, making it difficult to digest.
- **Iteration 4** attempts to provide a holistic view of both languages, combining strengths from earlier iterations, but can overwhelm users with too much information.

**Conclusion**: **Iteration 2** offers the best balance between simplicity and depth, effectively comparing the functional differences of C++ and Python while providing clear use cases for each language.

D.4 CASE STUDY: IMPLEMENTING A REST API IN FLASK

**Question**: *"Describe how to implement a simple REST API in Flask."*

**Outputs**

- **Output 1**: "Sure, I can guide you through implementing a basic RESTful API using Flask as your framework of choice. Here are the steps:
1. **Create Your Python Project**: First, create a new Python project and install Flask using pip: pip install flask. This will allow us to build our server which we can then host on any

web server or cloud provider like AWS, Google Cloud etc., for production use.
2. **Define Your Endpoints**: Next, define what endpoints your application needs to handle by creating routes in Flask..."

- **Output 2**: "1. First, you need to set up a Python file with the necessary code for your REST API. You can do this by creating a new file called 'app.py' and adding the following basic structure:
```
from flask import Flask, request, jsonify
import os
class MyAPI(Flask):...
```
The output also discusses other concepts such as session handling and database management."

- **Output 3**: "Let's break down the implementation into several parts to ensure clarity:
1. **Setting Up the Development Environment**: Install Flask and essential dependencies, then create a project folder and set up a virtual environment.
2. **Flask API Structure**: Structure your API and use additional libraries such as SQLAlchemy for database interactions and Twilio's API for sending messages."

- **Output 4**: "Here's a simplified outline of what you need to cover to build a RESTful API in Flask:
1. Set up your project and environment.
2. Create Flask app.
3. Define resources and endpoints.
4. Implement CRUD operations.
5. Handle errors gracefully and manage high traffic scenarios."

**Analysis**

- **Output 1**:
  - **Strengths**: Provides a concise introduction with a clear step-by-step guide on starting a REST API project with Flask.
  - **Weaknesses**: While it gives an initial direction, it lacks detailed implementation of each step, making it insufficient for beginners seeking a more complete understanding.

- **Output 2**:
  - **Strengths**: Offers a more detailed explanation, including specific code and a comprehensive example of a REST API structure.
  - **Weaknesses**: Attempts to explain too many different concepts (e.g., authentication, session handling) without focusing on the core task of building a simple REST API, which could overwhelm beginners.

- **Output 3**:
  - **Strengths**: Breaks down the process systematically, covering environment setup and API structure with more technical depth.
  - **Weaknesses**: Includes extraneous information such as Twilio integration and Elastic-search, which may distract from the central task of building a REST API.

- **Output 4**:
  - **Strengths**: Provides an overview of the necessary steps for implementing a REST API with Flask and highlights key concepts such as CRUD operations and error handling.
  - **Weaknesses**: Lacks specific code examples, making it harder for readers to translate the steps into an actual implementation.

**Overall Analysis**

- **Iteration 1** provides a basic step-by-step guide but lacks depth in key areas such as CRUD implementation and error handling.

- **Iteration 2** attempts to give a more complete explanation but introduces too many advanced concepts, which could confuse beginners.

- **Iteration 3** offers a detailed setup process but includes unnecessary information that distracts from the primary goal of building a REST API.
- **Iteration 4** gives a good overview but is too high-level and lacks concrete examples to help readers understand how to apply the steps.

**Conclusion**: **Iteration 2** offers the most comprehensive explanation but would benefit from focusing more on the key aspects of building a REST API, while **Iteration 1** is best suited for those seeking a simpler introduction.

### D.5 CASE STUDY: ADDING AN ITEM TO A WEB PAGE'S HTML LIST USING JAVASCRIPT

**Question**: *"Write a JavaScript code snippet to add an item to a web page's HTML list."*

**Outputs**

- **Output 1**: "Sure, here is a simple JavaScript code snippet that will add an item with the text 'New Item' to your web page:
```
// Select the first unordered list on the page (you can use
this ID or class)
const ul = document.querySelector('ul');
if (ul) {
const li = document.createElement('li');
li.textContent = 'New Item';
ul.appendChild(li); }"
```

- **Output 2**: "To add a new item to an HTML list using JavaScript, follow these steps:
1. Get the reference of the parent element where you want to insert the new list item. For example:
```
const myList = document.getElementById('myList');
```
2. Create a new list item node as a child of the selected parent element:
```
const liNode = document.createElement('li');
liNode.innerHTML = 'New List Item';
```
3. Append the newly created list item node to the specified parent container:
```
myList.appendChild(liNode);"
```

- **Output 3**: "Here is a basic implementation of the functionality you described in JavaScript:
```
// HTML structure with form and list
<form id='createForm'>
<label for='newItemText'>Enter new item text:</label>
<input type='text' id='newItemText'>
<button onclick='addToList()'>Add to list</button>
</form>
<ul id='existingListItems'></ul>
// JavaScript to handle the button click
function addToList() {
var inputText = document.getElementById('newItemText').value;
var li = document.createElement('li');
li.textContent = inputText;
document.getElementById('existingListItems').appendChild(li);
}"
```

- **Output 4**: "Here's an example of how you might implement this functionality using JavaScript:
```
// HTML structure for input form and list
<div id='createForm'>
<label for='newItemText'>Enter new item text:</label>
<input type='text' id='newItemText'>
<button onclick='addToList()'>Add to list</button>
</div>
<ul id='existingListItems'></ul>
// JavaScript to handle the add button click
```

```
function addToList() {
var inputText = document.getElementById('newItemText').value;
var li = document.createElement('li');
li.textContent = inputText;
document.getElementById('existingListItems').appendChild(li);
}"
```

**Analysis**

- **Output 1**:
  - **Strengths**: Provides a simple, concise code example for adding an item to an unordered list. The logic is easy to follow and useful for those who need a quick solution.
  - **Weaknesses**: The example assumes the presence of an unordered list without explaining how the list is structured or created. It's the simplest solution but lacks the depth and completeness of the other outputs.

- **Output 2**:
  - **Strengths**: Breaks the process into clear steps with explanations. The structured approach helps users follow along easily.
  - **Weaknesses**: The code becomes slightly verbose and introduces unnecessary steps for what could be a simpler task. It also lacks the interactive element provided by Output 3 and Output 4.

- **Output 3**:
  - **Strengths**: Provides a complete implementation with both HTML and JavaScript, showcasing how to dynamically add items to an HTML list using a form. It demonstrates both code clarity and a practical use case, including event handling.
  - **Weaknesses**: While comprehensive, the use of forms and event handling could add a small amount of complexity for users unfamiliar with those concepts.

- **Output 4**:
  - **Strengths**: Similar to Output 3, it provides a full implementation with HTML and JavaScript, with clean event handling and form interaction.
  - **Weaknesses**: It includes form handling in a slightly clearer structure but doesn't provide much additional value compared to Output 3. The form setup is slightly redundant for a simple task like adding a list item.

**Overall Analysis**

- **Iteration 1** is the most concise but lacks the depth and flexibility needed for a full implementation, making it the simplest but least powerful solution.

- **Iteration 2** is well-structured but lacks the interactive element of handling input dynamically through a form, making it less functional than Iteration 3 and 4.

- **Iteration 3** provides the most complete and functional solution, offering both HTML and JavaScript for a practical, interactive example.

- **Iteration 4** is also strong but is somewhat redundant compared to Iteration 3, offering slightly less detail in terms of implementation.

**Conclusion**: **Iteration 3** offers the best balance between completeness and practical functionality. It provides a clear and interactive solution, making it the most useful for implementing the task of adding an item to an HTML list using JavaScript.

### D.5.1 CASE STUDY: CALCULATING THE AVERAGE OF A LIST IN PYTHON

**Question**: *"Write a Python function that takes a list of numbers and returns the average."*

**Outputs**

- **Output 1**: "Sure! Here is a simple Python function that does this:
  ```
  def calculate_average(numbers):
  total = sum(numbers)
  return total / len(numbers)"
  ```

- **Output 2**: "Sure, here is a Python function called 'calculate_average' that takes a list of numbers as input and returns the average value. It also handles edge cases like an empty list:
  ```
  def calculate_average(numbers):
  return sum(numbers) / len(numbers) if len(numbers) > 0 else
  None"
  ```

- **Output 3**: "Here is a Python program that implements the 'calculate_average' function with additional checks and clarity:
  ```
  def calculate_average(numbers):
  if len(numbers) == 0:
  return None
  total = sum(numbers)
  return total / len(numbers)
  ```
  This function includes a check for empty lists and provides a clean and understandable implementation with a return of 'None' for edge cases."

- **Output 4**: "Here is a modified version of the 'calculate_average' function that adds extra error handling and raises exceptions for certain cases:
  ```
  def calculate_average(numbers):
  if len(numbers) == 0:
  raise ValueError('Input cannot be empty')
  elif len(numbers) == 1:
  return float('nan')
  total = sum(numbers)
  return total / len(numbers)
  ```
  This function handles both empty lists and single-element lists more strictly by raising exceptions and returning a NaN value where applicable."

**Analysis**

- **Output 1**:
  - **Strengths**: Provides the simplest and most direct solution for calculating the average of a list. It is easy to understand and quick to implement for basic use cases.
  - **Weaknesses**: Does not handle any edge cases, such as empty lists, which may result in errors if used in real-world scenarios. It's a good starting point but lacks robustness.

- **Output 2**:
  - **Strengths**: Provides a simple and practical solution with a basic error check for empty lists. The function is easy to understand and can handle the common case of an empty list by returning 'None'.
  - **Weaknesses**: The function only checks for empty lists but does not handle other potential issues such as single-element lists or non-numeric input. It is simpler than necessary for users looking for a more robust solution.

- **Output 3**:
  - **Strengths**: Provides a well-balanced solution with clear code and reasonable error handling. It accounts for edge cases such as empty lists and has a clean and readable structure. This output presents a practical and robust solution for calculating averages.
  - **Weaknesses**: The implementation is straightforward, but it does not handle more complex exceptions such as non-numeric input, which could be useful for certain applications.

- **Output 4**:
  - **Strengths**: Adds more advanced error handling by raising exceptions for empty lists and returning NaN for single-element lists. This output is ideal for users who want more strict error handling in specific edge cases.

   – **Weaknesses**: While the function handles more complex scenarios, the added complexity may not be necessary for most average calculation tasks, making the function slightly over-engineered for basic purposes.

**Overall Analysis**

- **Iteration 1** offers the simplest approach, but it doesn't handle edge cases. It's a good introductory solution but lacks robustness for more complex situations.

- **Iteration 2** is a simpler solution that handles empty lists but lacks more advanced error checking. It is useful for straightforward applications where minimal error handling is required.

- **Iteration 3** provides the most balanced solution, combining clarity with practical error handling. It is the best option for a well-rounded, everyday use case.

- **Iteration 4** introduces more strict error handling but adds complexity that may not be necessary for basic tasks. It's useful for those who want more control over edge cases.

**Conclusion**: **Iteration 3** strikes the best balance between simplicity and practical error handling. It provides clear code with a clean solution for handling basic edge cases. **Iteration 4** is stronger in error handling but may be unnecessarily complex for most use cases. **Iteration 2** is simple and effective but lacks robustness, and **Iteration 1** is the most basic solution for introductory use.

