# OpenReview forum: "Think Thrice Before You Act: Progressive Thought Refinement in Large Language Models"
_ICLR.cc/2025/Conference — ICLR 2025 Poster_

### Official Review · Reviewer_YHbN · 2024-10-31

**Soundness:** 3
**Presentation:** 3
**Contribution:** 3
**Rating:** 6
**Confidence:** 4

**Summary:**

The paper proposes Progressive Thought Refinement (PTR) for LLMs, focusing on iterative refinement without task-specific fine-tuning. PTR uses a weak-strong model collaboration and a unique thought-mask fine-tuning approach to refine model reasoning, showing generalization across diverse tasks.

**Strengths:**

+ PTR’s approach to iterative response improvement is novel, aiming to boost LLM generalization.
+ Benchmarking against established methods helps demonstrate PTR’s relative effectiveness.

**Weaknesses:**

- Critical information, such as data size, prompt details, and filtering strategies, is not presented clearly within the main text. Instead, these details are found only in the appendix, requiring readers to frequently refer back and forth for a complete understanding. Integrating these important details into the main text rather than the appendix will provide readers with immediate context and reduce the need for cross-referencing, enhancing readability and comprehension.

- The paper lacks clear guidelines on the optimal number of iterations for masking and fine-tuning. Accuracy fluctuates across iterations, indicating the model’s sensitivity to iteration count without sufficient explanation/justification. Establish clear guidelines on the optimal number of iterations for masking and fine-tuning.

- The paper would benefit from testing on datasets designed for reasoning and generalization evaluation, such as MMLU-Pro+ (arXiv:2409.02257), which could provide a more robust demonstration of PTR’s reasoning capabilities.

**Questions:**

See Weaknesses

---

> ### Author Response · Authors · 2024-11-21
>
> Thank you for your positive feedback.
>
> > Data size, prompt details, and filtering strategies
>
> We have ensured that the core content regarding **data size**, **prompt details**, and **filtering strategies** is included in the main text, so readers can access the necessary information directly.
>
> - In the **Experiments** section under **Dataset Sources**, we have specified the size of the dataset as 40k QA pairs. Additionally, we provide further details about the dataset size in the Appendix.
> - In the **Methods** section under **Thought Mask Mechanism**, we have briefly described the prompt details. For example, we provide refinement instructions such as, *"Please continue thinking and refine your answer."* Further specifics about the prompt instructions and inference results are included in the Appendix.
> - For filtering strategies , are discussed in  3.1.1 QUERY PREPARATION , where we detail the steps:
>   1. Removing noisy and irrelevant queries, such as URLs or image links.
>   2. Ensuring that test set questions are not mixed into the training set to prevent data leakage.
>   3. Employing **Thoughts-Answer Consistency Filtering** to further verify logical consistency and improve the quality of the training dataset. Details of this consistency verification method are also provided in the Appendix.
>
> Given the limited space in the main text, we have placed more detailed and technical information at the beginning of the Appendix, so readers can quickly find supplementary material. We hope this structure effectively helps readers access relevant content efficiently. If you have any further suggestions, we are happy to optimize this further.
>
>
>
>
>
> > Clear guidelines on the optimal number of iterations for masking and fine-tuning.
>
> We appreciate your feedback regarding the need for clearer guidelines on the optimal number of iterations.
>
> Based on the content of the article, we have already addressed both the number of fine-tuning steps and the number of reasoning iterations required for optimal performance:
>
> 1. **Fine-tuning Steps:**
>
> 1. In section 4.4, the article demonstrates that significant improvements in performance emerge after approximately **24,000 training steps** (equivalent to 93 million tokens) for our base model. At this point, the model begins to show refined capabilities. The performance continues to improve as training progresses, with \model{} achieving an overall improvement from 40.1% to 55.6%. This shows that, while there is a clear upward trend in performance during training, a substantial number of training steps (around 24,000) are needed for more complex reasoning tasks to emerge.
>
> 1. **Reasoning Iterations:**
>
> 1. In section 4.5, Regarding iterative thinking steps, the article reveals that **iterative refinement** plays a crucial role in achieving optimal performance. In particular, the first three iterations show significant performance improvements across various tasks, such as **GSM8K** (from 75.0% to 79.9%) and **ARC** (from 58.6% to 65.2%).
>
> > more robust demonstration of PTR’s reasoning capabilities.
>
> Thank you for your suggestion! We appreciate your recommendation to test reasoning and generalization capabilities using datasets like MMLU-Pro+ (arXiv:2409.02257). In fact, we have already conducted experiments on 10 different datasets, which we believe cover a wide range of capabilities, including general knowledge, code comprehension, summarization, math reasoning, and knowledge. These datasets include MMLU, Humaneval, DROP, XSum, GSM8K, Math, ARC, GPQA, Winogrande, and CommonsenseQA. We believe this range of tests is sufficient to demonstrate the robustness and generalization capabilities of our model.
>
> That being said, we have also added results from MMLU-Pro+ as follows:
>
> | Qwen2-7B                | MMLU PRO+ |
> | ----------------------- | --------- |
> | Base-model interation 1 | 30.1      |
> | Base-model interation 2 | 24.2      |
> | Our model  interation 1 | 28.8      |
> | Our model  interation 2 | 30.9      |
> | Our model  interation 3 | 31.3      |
>
> We observe similar improvement results on MMLU-Pro+, where the base model fails to improve for multiple iterations and shows a significant drop in performance after the second iteration. However, our model demonstrates self-improvement, increasing accuracy by 2.5%. These results further validate the effectiveness of our approach, demonstrating its ability to enhance reasoning capabilities. We are pleased that our findings align with the robustness requirements highlighted in your comment.
>
> We will include this experiment in the revision version of our paper and cite the MMLU-Pro+ (arXiv:2409.02257). We sincerely appreciate the reviewer’s constructive suggestions. The additional experiments, analyses, and explanations have significantly enhanced the quality of our submission. We hope this provides a strong basis for a score increase.

---

> > ### Comment · Reviewer_YHbN · 2024-11-26
> > **thank you for your response**
> >
> > I will increase my score

---

> > > ### Author Response · Authors · 2024-11-27
> > > **Thanks for your time and thoughtful feedback!**
> > >
> > > Thank you for your feedback and for increasing your score! Your thoughtful review and support are greatly appreciated. If you have any further questions, please feel free to contact us!

---

> ### Author Response · Authors · 2024-11-25
> **Re to Q1**
>
> ## data size
>
> - In the **Experiments** section under **Dataset Sources**, we have specified the size of the dataset as 40k QA pairs.
> In line 286
> Our model has trained on our PTR (Progressive Thought Refinement) dataset, and derived queries from the WizardLM dataset. After thorough cleaning, we reduced the original dataset from approximately 50k QA pairs to 40k high-quality QA pairs.
>
>
> |          | **MMLU** | **H-Eval** | **DROP** | **Xsum** | **GSM8k** | **Math** | **ARC** | **GPQA** | **Wino** | **Comm** |
> | -------- | -------- | ---------- | -------- | -------- | --------- | -------- | ------- | -------- | -------- | -------- |
> | **size** | 14042    | 164        | 9536     | 11334    | 1319      | 5000     | 1172    | 448      | 1767     | 1140     |
>
>
>
> ## prompt details
>
>
>
> - In the **Methods** section under **Thought Mask Mechanism**, we have briefly described the prompt details. For example, we provide refinement instructions such as, ***"Please continue thinking and refine your answer."*** Further specifics about the prompt instructions and inference results are included in the Appendix.
>
> ### For training stages:
>
> we simply use "Please continue thinking and refine your answer." to let model learn how to improve their answers.
>
>
>
> ### For inference stages in line324:
>
> For main experiements' results, we also use "Please continue thinking and refine your answer." to ask model to improve their answers.
>
>
>
> ### For inference stages in line397:
>
> we also tests robustness of our methods on different prompts. As in real world tasks, users may ask model to refine their answer in multiple ways.
>
> Results of our methods with different prompts to test the robostic of our trianing methods:
>
> - Assume that this thought could be either correct or incorrect. Carefully review the thought and provide a better answer.
>
> - Review your previous thought and assess whether it's correct. Then, provide a better response based on your answer.
>
> - Regardless of whether your previous thought is correct or not, provide a better answer.
>
> Notibly, the model is not trained with these prompts.
>
> The results of our methods shows that our methods can improve models performance on different prompts.
>
> ## filtering strategies
>
>
>
> Our filtering strategies include 3 parts:
>
> In 3.1.1 Query Preparation
>
> To enhance the model’s generalization, we avoid creating domain-specific datasets. Instead, we use queries from open-domain general datasets ensuring the model develops general refinement abilities rather than specializing in specific areas. Our data preprocessing involves three key steps. **First, we perform data cleaning to remove noise and irrelevant content,** such as images or URLs. Second, to prevent data leakage, **we exclude domain-specific testing queries during training**. Finally, we incorporate traditional SFT data (queries and answers) into our dataset to mitigate the risk of catastrophic forgetting.
>
> And in **Thoughts-Answer Consistency Filtering** in line 245
>
> To further ensure that the thought process exhibits logical coherence, we apply consistency filtering to remove inconsistent outputs. If the consistency score is below a certain threshold, the pair is considered inconsistent and removed, ensuring that only coherent thought sequences are used for the final output.
>
>
>
> ​

---

> ### Author Response · Authors · 2024-11-25
> **Re to Q2**
>
> ## clear guidelines on the optimal number of iterations for masking and fine-tuning.
>
> **Fine-tuning Steps:**
>
> 1. In section 4.4, the article demonstrates that significant performance improvements emerge after approximately **24,000 training steps** (equivalent to 93 million tokens) for our base model. At this point, the model begins to show refined capabilities. The performance continues to improve as training progresses, with our methods achieving an overall improvement from 40.1% to 55.6%. This shows that, while there is a clear upward trend in performance during training, a substantial number of training steps (around 24,000) are needed for more complex reasoning tasks to emerge.
>
>
>
> To be more specific
>
> > fine-tuning setting
>
> | Hyperparameter/Description   | Training Values | Inference |
> | :--------------------------- | :-------------- | :-------- |
> | bf16                         | TRUE            | N/A       |
> | epochs                       | 2               | N/A       |
> | per device train batch size  | 1               | N/A       |
> | gpus                         | 2xH8100         | 2xH800    |
> | gradient accumulation steps  | 256             | N/A       |
> | learning rate                | 5e-5            | N/A       |
> | weight decay                 | 0               | N/A       |
> | warmup step                  | 1000            | N/A       |
> | learning rate scheduler type | cosine          | N/A       |
> |                              |                 |           |
> | model max length             | 2048            | 2048      |
> | temperature                  | N/A             | 0         |
> | top_p                        | N/A             | 1         |
> | max_new_tokens               | N/A             | 1000      |
>
> Our fine-tuning costs are significantly lower than other methods. Only around 24000 steps of iteration with 40K data could achieve self-refinement capabilities across various domains. It is unnecessary to fine-tune and construct data for every specific task.

---

> > ### Author Response · Authors · 2024-11-25
> > **Fine-tuning and Inference Comparison Table with other methods**
> >
> > We also compared with other methods:
> >
> >
> >
> > We present a detailed comparison of our method with existing approaches, highlighting the key differences in cost, applicability, and performance across tasks such as GSM8K and general downstream tasks.
> >
> > ### **Comparison Table**
> >
> > | **Aspect**                                                  | **Our Method**                                               | **Self-Refine**                                              | **Pair Self-Correction**                                     | **Reward-Model Verifier**                                    |
> > | :---------------------------------------------------------- | :----------------------------------------------------------- | :----------------------------------------------------------- | :----------------------------------------------------------- | :----------------------------------------------------------- |
> > | **Iteration costs for Construct Data**                      | Requires only general domain queries; uses Qwen-2.7B and Qwen-70B to construct training datasets without external feedback. | No training required.                                        | Requires few-shot prompting with a 175B model, generating both correct and incorrect answers for training. The process involves at least two inferences per query. | Requires 100 inferences times with a 175B model to generate n-sampling responses for higher accuracy, filtering for correct and incorrect answers for dataset construction. |
> > | **Training Cost**                                           | 2 epochs on the base model.                                  | No training required.                                        | 2 epochs on the base model.                                  | 2 epochs on the verifier model.                              |
> > | **Additional Data Construction and Training for new Tasks** | Not required.                                                | Requires task-specific prompts with ground truth to assist evaluation. | Requires constructing task-specific datasets of correct and incorrect samples. Limited to tasks that make it easy to distinguish the correctness. | Requires constructing task-specific datasets of correct and incorrect samples. Limited to tasks that make it easy to distinguish the correctness. |
> > | **Need for External Feedback to Construct Data**            | Not required.                                                | Not applicable.                                              | Required correctness of answer.                              | Required correctness of answer.                              |
> > | **Number of Iterations During Inference**                   | 2–3 iterations.                                              | 2–3 iterations.                                              | 2–3 iterations.                                              | 2–3 iterations with 100 inference samples per iteration to improve response accuracy. |
> > | **Performance Improvement on GSM8K**                        | 2.9 /%.                                                      | 2.2 /%.                                                      | 2.4 /%.                                                      | 1.7 /%.                                                      |
> > | **Performance on Unseen Tasks**                             | Overall improvement of 3.9 points.                           | Requires task-specific prompts and ground truth for evaluation. | Requires additional dataset construction and fine-tuning, which is costly for subjective tasks, and impractical. | Requires additional dataset construction and fine-tuning, which is costly for subjective tasks, and impractical. |
> > | **Feedback During Inference**                               | Not required; generates an improved response regardless of initial correctness. | Requires feedback using ground truth to evaluate correctness. | Uses the model's self-verify to evaluate correctness. If incorrect, it iterates; if correct, it outputs the response. | Uses an additional verifier model to evaluate correctness. If incorrect, it iterates; if correct, it outputs the response. |
> >
> > ------
> >
> > [1] Self-refine：Self-refine: Iterative refinement with self-feedback https://arxiv.org/abs/2303.17651
> >
> > [2] Pair Self Correction： Generating Sequences by Learning to Self-Correct https://arxiv.org/abs/2211.00053
> >
> > [3] Reward-model Verifier: Training Verifiers to Solve Math Word Problems https://arxiv.org/abs/2110.14168

---

> > > ### Author Response · Authors · 2024-11-25
> > >
> > > ##  fluctuates across iterations
> > >
> > > > Accuracy fluctuates across iterations, indicating the model’s sensitivity to iteration count without sufficient explanation/justification.
> > >
> > > The primary contribution of our work lies in stimulating the model’s ability for self-refinement. From the main experiments, we observe that the model, using our method, can refine its previous responses, whereas the base model not only fails to achieve this but sometimes performs worse over iterations.
> > >
> > > Additionally, as discussed in Section 4.5 *HOW MANY THINKING STEPS ARE REQUIRED TO ACHIEVE OPTIMAL PERFORMANCE?*, the model shows significant improvements in the first three iterations, after which its performance tends to stabilize. This demonstrates that the model effectively refines its answers during the early self-refinement stages, validating the strength of our approach. The subsequent stabilization indicates that the model’s inherent capabilities have an upper limit. Even advanced systems like GPT-4 are not immune to errors, and it is unrealistic to expect perfect accuracy regardless of the number of iterations.
> > >
> > > As for why the model cannot achieve further refinement after the third or fourth iteration, we attribute this to two primary factors:
> > >
> > > 1. **Inherent Limitations of Model Capabilities**
> > >    The model’s performance is constrained by the knowledge and abilities it acquired during pretraining. While our method successfully stimulates self-refinement, the extent of this improvement is finite. Current generative models, including state-of-the-art systems like GPT-4, still exhibit errors in complex tasks. Consequently, even with iterative refinement, the model cannot surpass its inherent limitations, and opportunities for further improvement naturally diminish.
> > >
> > > 2. **Convergence of Response Quality**
> > >    During early iterations, there is substantial room for improvement, resulting in noticeable refinements. However, as the response quality approaches the model’s capability boundary, subsequent iterations yield diminishing returns or even stagnation. In other words, once the generated answers approach the model's performance ceiling, additional refinement can only result in marginal gains, which may be less noticeable or significant.
> > >
> > > We emphasize that the clear improvements observed during the first three iterations robustly demonstrate the effectiveness of our method. The stabilization seen in later iterations does not indicate a failure of the approach but rather reflects the model reaching its natural performance limit. This highlights that while achieving perfect responses may be infeasible, our method successfully activates the model’s self-refinement potential, enabling it to provide progressively better answers through iterative refinement. This offers users a practical tool for enhancing response quality in real-world applications.
> > >
> > >
> > >
> > > > Establish clear guidelines on the optimal number of iterations for masking and fine-tuning.
> > >
> > > ## Reasoning Iterations:
> > >
> > > 1. As discussed in Section 4.5, iterative refinement is critical for achieving optimal performance. The experimental results highlight that the first three iterations consistently yield significant improvements across various tasks. For instance, performance on **GSM8K** improves from 75.0% to 79.9%, while accuracy on **ARC** increases from 58.6% to 65.2%. This demonstrates that iterative reasoning effectively enhances the model’s output during the early stages, suggesting three iterations as a general benchmark for many tasks.
> > > 2. For more complex tasks requiring deeper logical reasoning, additional iterations may be necessary to achieve optimal results. These tasks often involve intricate relationships or multi-step problem-solving, which demand extended refinement processes to fully capture the nuances of the problem and deliver high-quality responses. Like MATH datasets, we can see the highest accuracy appears in the 5th iteration.
> > >
> > > **Recommendation:**
> > > Based on these findings, we recommend setting the number of iterations dynamically, considering task complexity. For tasks of moderate difficulty, three iterations provide a good balance between performance gains and efficiency. For highly complex tasks, additional iterations should be explored to ensure sufficient refinement while monitoring for performance stabilization to avoid unnecessary computation.

---

### Official Review · Reviewer_uz4P · 2024-11-04

**Soundness:** 3
**Presentation:** 2
**Contribution:** 2
**Rating:** 3
**Confidence:** 3

**Summary:**

The paper proposes Progressive Thought Refinement (PTR), a novel framework designed to improve the iterative reasoning capability of large language models (LLMs). PTR leverages a dual-phase approach to construct a progressive refinement dataset and then applies a thought-masking fine-tuning technique to teach LLMs to self-improve. PTR aims to enhance the generalization capability of LLMs across diverse tasks, including knowledge reasoning, code generation, and open-ended text generation, without task-specific fine-tuning. Experimental results indicate that PTR improves performance across a wide range of benchmarks, demonstrating its effectiveness in enabling models to progressively enhance response quality over multiple iterations.

**Strengths:**

- The paper addresses a highly relevant and timely topic within the machine learning community. Self-refinement has been perceived as an important approach for improving pre-trained LLMs.
- The authors have conducted extensive experiments, exploring the performance of their model and the baselines in different domains.

**Weaknesses:**

- The manuscript lacks sufficient clarity, making it challenging for readers to follow the paper
- 2 paragraphs of lines 358-467 are exactly duplicated in lines 510-520!
- Also, the claim in those paragraphs, "PTR consistently improves across iterations without signs of overfitting." is not the case in Figure 4. It looks like the performance is saturated after the first iteration and is just oscillating from iterations 2 to 10, especially in the XSsum experiment.
- Figure 1 which is supposed to show the core idea/method is unnecessarily complicated and overloaded making it hard to interpret.
- Figure 3 has 4 plots with lots of information, small numbers w/o proper take-home messages in the caption.
-  The description for \textbf{PTR vs. Knowledge Distillation (IFT)} in line 320 is not clear and hand-wavy.
- Claiming that LLMs have intrinsic/inherence progressive refinement ability is sort of a big claim that needs more detailed investigations.
- By looking at the average scores in Table 1, the Prompt baseline is already showing a very solid performance in the first iteration. When comparing this baseline with your method you should consider the complexity of your approach and the time analysis.

**Questions:**

- Have the authors investigated the effect of different losses of equation 3.4 in an ablation study?
- What is the "PRD" dataset mentioned in the paragraph of line 303?

---

> ### Author Response · Authors · 2024-11-21
> **Re to progressive refinement ability more detailed investigations.**
>
> Thank you for your valuable efforts and insightful comments. We have carefully addressed your concerns in detail below and hope our responses meet your expectations. As the discussion phase approaches its end, we look forward to any additional feedback or clarification requests and hope for a constructive dialogue to refine our work further.
>
> > Claiming that LLMs have intrinsic/inherence progressive refinement ability is sort of a big claim that needs more detailed investigations.
>
>
> To address the reviewer’s concern regarding the claim that Large Language Models (LLMs) have an intrinsic ability for progressive refinement, we offer the following clarifications and elaborations:
>
> ## 1. Distinguishing Between Model Capabilities and Self-Improvement Ability
>
> It is important to clarify that the inherent ability of a model to respond to tasks (its base performance) is different from its ability to self-improve or refine answers over time. While the base model's capabilities are largely determined by the knowledge it has accumulated during pre-training, we recognize that enabling progressive refinement is a separate, significant challenge. Our experiments show that while the base model does not exhibit any self-correction or refinement ability over multiple iterations, this ability can be triggered with our method. In our work, we demonstrate that, with just 50,000 data pairs and 24,000 training steps (equivalent to 93 million tokens), a 7B parameter model can exhibit significant self-improvement in 10 different unseen tasks. This process of fine-tuning activates the model’s ability to progressively refine its outputs, which is different from its original pre-trained capabilities. Thus, the self-improvement ability is not shown in the base model but rather emerges through additional fine-tuning, and this does not contradict the model's original base solid performance.
> ## 2. Comparison with Related Work
> We also compare our method with three other approaches in the "Costs Comparison Table for each methods" responses, analyzing the cost and effectiveness of each. Our findings show that, while other methods can achieve similar improvements (2-3%), they require more expensive training and inference processes. Specifically, methods like Verifier rely on extensive retraining, which involves using clear positive and negative samples to construct datasets. This process demands additional training data every time a new task or domain is encountered. In contrast, our approach allows for the model to demonstrate self-improvement across a variety of tasks without the need for task-specific datasets. Furthermore, our method requires significantly fewer inference steps during evaluation: while other methods like Verifier require 100 inference steps per iteration, we find that just two or three iterations with our method provide comparable or better performance, making our approach more practical and cost-effective.
>
>
> ## 3. Key Contribution: Enabling Self-Improvement Across Unseen Tasks
> The primary contribution of our work is not in improving scores in specific domains, but in enabling the model to learn self-improvement through this training approach, allowing it to demonstrate progressive refinement across different tasks.Since our method is not task-oriented, the model trained with our approach can still refine itself when encountering new scenarios or tasks it has never seen before, unlike the base model, which either does not improve or makes incorrect corrections.This significantly enhances the practical applicability of our method in real-world scenarios and also reduces the cost of training.
> We believe that, in real-world scenarios, even the most advanced models cannot achieve 100% accuracy, and there will always be instances where the model's responses are not ideal. Therefore, users may want the model to improve upon its previous answers. Existing methods mainly focus on specific domains, constructing task-oriented datasets to train models for improving self-refinement within those domains. However, in real life, tasks are not limited to just mathematics. Achieving self-improvement across a wider range of tasks is more challenging but also more valuable. Our method enables the model to demonstrate self-improvement across multiple tasks it has not been specifically trained on, which is the primary contribution of our work.
>
> Continued in https://openreview.net/forum?id=pUbbLHjCPM&noteId=aBIAkvxvtw

---

> > ### Comment · Reviewer_uz4P · 2024-11-27
> > **Response to the authors after the rebuttal**
> >
> > I would like to thank the authors for their response. Some of the concerns I raised have been addressed in their reply. However, I still believe there are three major issues with the paper:
> >
> > Presentation: The figures are vague and overly complex.
> > Results: As I mentioned, based on Table 1, the improvement over simple prompting is marginal, especially considering the added complexity, which makes it impractical for real-world use.
> > Iterative Performance: In Figure 4, while there is some improvement after a few iterations, saturation occurs too quickly. The claim that "PTR consistently improves across iterations without signs of overfitting" is not supported by the evidence.
> >
> > Hence, I'm keeping my score.

---

> > > ### Author Response · Authors · 2024-11-28
> > > **Re to Presentation**
> > >
> > > > **Presentation: The figures are vague and overly complex.**
> > >
> > > I would like to thank the reviewer for their valuable feedback. We appreciate the opportunity to clarify our work. Regarding the concern about the figures being vague and overly complex, we believe that the figures in our paper are carefully designed to clearly convey the key steps of our methodology.
> > >
> > > In **Figure 1**, we provide a clear and straightforward illustration of the entire process: (A) depicts how the data is constructed, (B) shows how the model is trained, and (C) demonstrates how the trained model performs inference. These details are integral to understanding our approach and the key steps involved, and we believe each part is necessary to ensure clarity in conveying our methodology.
> > >
> > > **Figures 3 and 4** focus on the overall improvements achieved across multiple datasets, highlighting the performance gains after training and iterations. These figures also consider the overall aesthetic layout of the paper. To avoid overwhelming the reader with excessive complexity, we have included additional figures in the appendix for a more detailed exploration. Furthermore, we have enlarged the font size and added more explanatory notes in the figure captions to enhance clarity and readability.
> > >
> > > We hope that these adjustments improve the accessibility of the paper, and we kindly ask the reviewer to reconsider these aspects in light of the changes made. We believe that these revisions contribute to a clearer, more intuitive presentation of our work.

---

> > > ### Author Response · Authors · 2024-11-28
> > > **Re to Results**
> > >
> > > > Results: As I mentioned, based on Table 1, the improvement over simple prompting is marginal, especially considering the added complexity, which makes it impractical for real-world use.
> > >
> > > Thank you for your thoughtful comments regarding the results and the improvement over simple prompting. We understand your concern about the marginal improvement in performance, especially considering the added complexity. However, we believe that the benefits of our approach, as highlighted in Table 1, go beyond what simple prompting can achieve.
> > >
> > > Re: as we mentioned in (https://openreview.net/forum?id=pUbbLHjCPM&noteId=SAMYrolxZC)
> > >
> > > We would like to emphasize the key contributions of our work, which are as follows:
> > >
> > > 1. **Base Models Lack Self-Improvement Capability** As shown in Table 1, the base model does not possess the ability for self-refinement, as it drops performance immediately after the second iteration. We refer to related work, such as [2] *Large language models cannot self-correct reasoning yet*(arXiv:2310.01798), which also discusses how current large language models struggle to self-correct reasoning errors without external feedback. Even large models face challenges in improving their responses to simple prompts, highlighting that the self-improvement capability is not inherent in pre-trained models, even for large-scale models.
> > > 2. **Improvement in Math Problems is Comparable to Existing Methods** Firstly, regarding math problems, our method achieves improvements similar to other refinement techniques, which typically yield gains in the range of 2-3%(https://openreview.net/forum?id=pUbbLHjCPM&noteId=nRyyXFHi6D).
> > > 3. **High Costs and Dataset Limitations of Existing Methods** However, existing refinement methods often require significantly higher computational costs. For instance, the generation costs are approximately 100 times higher in [3] https://openreview.net/forum?id=pUbbLHjCPM&noteId=nRyyXFHi6D) . And they are typically limited to specific datasets (https://openreview.net/forum?id=pUbbLHjCPM&noteId=Q0a1JD4Ioe). In contrast, our method improves performance efficiently and generalizes across a wider variety of real-world tasks, avoiding the high costs and dataset limitations inherent in other approaches.
> > > 4. **Self-Improvement Across a Variety of Unseen Tasks** Secondly, our method can still improve 4.1% on average of 10 unseen tasks without extra task-oriented training (in Table 1, we showed 10 cases vary from math, code, summary, reasoning, knowledge, comprehension tasks, and also added MMLU Pro+ in https://openreview.net/forum?id=pUbbLHjCPM&noteId=4YkxZGGW7X). Existing methods do not reach this performance, which is often confined to specific domains like math. For real-world applications, the ability to self-improve across a range of unseen, open-ended datasets is crucial and far more practical than methods focused solely on improving performance in a narrow domain.
> > >
> > > We hope this clarifies our contributions and addresses your concerns. We are confident that our approach offers meaningful improvements not only in math problems but also across various tasks, with the added benefit of being more efficient and broadly applicable in real-world settings.
> > >
> > > [3] Reward-model Verifier: Training Verifiers to Solve Math Word Problems https://arxiv.org/abs/2110.14168

---

> > > ### Author Response · Authors · 2024-11-28
> > > **Re to Performance of Iteration**
> > >
> > > > Iterative Performance: In Figure 4, while there is some improvement after a few iterations, saturation occurs too quickly. The claim that "PTR consistently improves across iterations without signs of overfitting" is not supported by the evidence.
> > >
> > >
> > >
> > > Thank you for your valuable feedback regarding the iterative performance and the concern about saturation occurring too quickly. We appreciate the opportunity to clarify our approach and results.
> > >
> > > **On the Main Contribution:**
> > >
> > > The main contribution of our work is to enable the model to self-refine without the need for domain-specific fine-tuning, allowing it to improve across various tasks. As shown in Table 1, the base model lacks this self-refinement ability, as it shows a performance drop immediately after the second iteration. We refer to related work, such as [2] *Large language models cannot self-correct reasoning yet* (arXiv:2310.01798), which highlights that even large models struggle to self-correct reasoning errors without external feedback. This demonstrates that self-improvement is not inherently present in pre-trained models, even large-scale ones.
> > >
> > > While we observe that our model stabilizes after four iterations, the key point is that it demonstrates **self-improvement** in the first three iterations, which we believe is a significant achievement. On average, the model’s performance improves by 3-4 points across different datasets, which is comparable to the improvements seen in other methods that focus on single-domain tasks. This provides strong support for the claim that our approach **successfully triggers the model’s self-improvement capability**.
> > >
> > >
> > >
> > > **On Iterative Improvement Speed:**
> > >
> > > We understand your concern that the improvement seems to plateau too quickly. However, we believe that the self-improvement observed after three to four iterations is not "too quick"; rather, it represents an appropriate and reasonable threshold. In fact, related work has typically performed self-improvement **for just two iterations,** without exploring the impact of further iterations on performance. Our method demonstrates that, after three iterations**, the model stabilizes and maintains its improved performance,** which we believe is a practical and effective outcome.
> > >
> > >
> > >
> > > **On the Claim of "Consistent Improvement":**
> > >
> > > Regarding the statement that "PTR consistently improves across iterations without signs of overfitting," we respectfully maintain that this claim is supported by the evidence. In some methods, such as those discussed in [3], overfitting is observed where the model shows improvements in the early iterations but then experiences performance degradation with subsequent iterations. In contrast, our method shows consistent improvement during the first three iterations. Once the model reaches a stable level of performance, it does not regress, further supporting the self-improvement capability of our approach.
> > >
> > > As shown in **Figure 4**, as the iteration steps increase, the model does not exhibit performance degradation, which would indicate overfitting. Instead, it stabilizes after a certain point, maintaining its improved performance without significant regression, demonstrating the effectiveness of the self-improvement process.
> > >
> > >
> > >
> > > **On Experimental Performance Validation:**
> > > To validate the robustness and effectiveness of our method, we conducted additional experiments. Firstly, as shown in **Table 1**, our method demonstrates consistent improvement across 10 different datasets, a result that other methods have not achieved. This broad applicability across diverse tasks highlights the versatility and robustness of our approach.
> > >
> > > Secondly, in **Table 2**, we show that our method is robust to variations in prompting. In real-world scenarios, user requirements are often diverse, and different refinement instructions can improve model performance in multiple ways. In contrast, as seen in [4] *Self-refine: Iterative refinement with self-feedback* (arXiv:2303.17651), existing methods require task-specific, customized prompts, which come with substantial costs. Our approach, however, avoids this overhead and remains effective across a variety of prompts, further demonstrating its practical advantages.
> > >
> > > **On Model Refinement Beyond Correctness:**
> > > Additionally, in **Figure 2**, we show that our method does not merely focus on improving accuracy but actually refines the model's output over multiple iterations. Even when the model's second response is correct, further refinement leads to even better answers. This ability to continue improving upon correct answers is a key feature of our method, and something that differentiates it from existing methods. The model does not simply transition from incorrect to correct, but rather refines its performance progressively, even when it starts with a correct response.

---

> > > > ### Author Response · Authors · 2024-11-28
> > > > **Re to Performance of Iteration**
> > > >
> > > > continued from https://openreview.net/forum?id=pUbbLHjCPM&noteId=25Mj7seERB
> > > >
> > > > Furthermore, as shown in the appendix, our method performs well on more open-ended tasks, not just focusing on correcting mistakes but demonstrating true self-improvement. This represents our primary contribution—an ability to continuously refine model performance, a capability that methods such as [4], [5], and [6] lack. We believe this aspect is crucial for real-world applications, where tasks are often open-ended and evolving.
> > > >
> > > > **Conclusion:**
> > > > We hope that these additional experimental results provide further evidence of the validity and strength of our approach. We kindly ask the reviewers to reconsider these contributions, as they highlight the practical advantages and robustness of our method over existing alternatives. We believe that the ability to refine models iteratively, with minimal additional cost and without requiring domain-specific fine-tuning, is a significant step forward in model self-improvement.
> > > >
> > > > Thank you once again for your thoughtful review, and we appreciate your consideration of these additional findings.
> > > >
> > > > [3]Progress or Regress? Self-Improvement Reversal in Post-training
> > > >
> > > > [4] Self-refine：Self-refine: Iterative refinement with self-feedback https://arxiv.org/abs/2303.17651
> > > >
> > > > [5] Pair Self Correction： Generating Sequences by Learning to Self-Correct https://arxiv.org/abs/2211.00053
> > > >
> > > > [6] Reward-model Verifier: Training Verifiers to Solve Math Word Problems https://arxiv.org/abs/2110.14168

---

> ### Author Response · Authors · 2024-11-25
> **Re to Figure3 Figure4 and description in line 320**
>
> > Figure 3 has 4 plots with lots of information, small numbers w/o proper take-home messages in the caption.
> more details showed in paper
> ## For Figure 3:
> Plot A: A multi-line plot illustrating performance trends across 10 tasks, with the average performance depicted in black and variance shown as shaded regions. The overall trend is upward, with a red line marking the point where the average performance reaches 46%.
> Plot B: A bar plot comparing each task's initial, baseline, and final performance. The initial performance is lower than the baseline, but the final performance surpasses it, indicating that the model effectively learns and improves through successive iterations.
> Plot C: A box plot showcasing the performance distribution across tasks. The varying lengths of the bars represent performance improvements, with longer bars indicating greater improvements across different tasks.
> Plot D: A heat map visualizing task performance over training steps. The x-axis represents training steps, and the y-axis represents tasks. The color intensity reflects performance levels, with darker colors indicating better performance.
> ## For Figure 4:
> The performance of our methods is evaluated over ten iterations across various tasks.
> Left Plots: These show accuracy improvements across categories such as mathematical reasoning (GSM8K, MATH), reasoning tasks (ARC, GPQA, Winogrande, CommonsenseQA), comprehension tasks (MMLU, DROP, XSum), and coding tasks (HumanEval). The dashed line represents the model's baseline performance. Most tasks exceed the baseline after the second or third iteration, demonstrating significant learning and adaptation.
> Right Plots: Radar charts illustrate performance over six iterations. The blue-shaded areas highlight remarkable performance improvements during the first two iterations, followed by saturation in subsequent iterations. This pattern demonstrates rapid early gains with diminishing returns as the iterations progress.
>
>
>
>
> > The description for \textbf{PTR vs. Knowledge Distillation (IFT)} in line 320 is not clear and hand-wavy.
>
> ## Description of PTR is not Knowledge Distillation
>
> We also compare our method with a straightforward approach that uses a strong model's answers to queries via IFT. Our findings show that our method is not equivalent to knowledge distillation.
>
> In the first iteration, we observe that when models are trained on general datasets rather than domain-specific tasks, their initial performance tends to decline. This performance drop primarily arises from supervised fine-tuning amplifying the base model’s inherent biases. When trained on general datasets, base models often accumulate biases that may not align with specific domains, resulting in poorer performance on domain-specific tasks.
>
> However, the IFT approach fails to activate the model’s self-refinement ability and does not significantly improve performance after the initial attempts. For example, on CommonsenseQA, the IFT approach yields a lower performance at the second iteration (40.3%) compared to the first (46.1%).
>
> In contrast, our approach demonstrates consistent improvement through iterative attempts without relying on domain-specific knowledge. This highlights that our method is not merely distilling knowledge but rather activating the model’s ability to refine its outputs and enhance performance through self-driven iterative improvement.

---

> ### Author Response · Authors · 2024-11-25
> **Re to time analysis**
>
> > When comparing this baseline with your method you should consider the complexity of your approach and the time analysis.
>
> We also compared with other methods:
>
>
>
> We present a detailed comparison of our method with existing approaches, highlighting the key differences in cost, applicability, and performance across tasks such as GSM8K and general downstream tasks.
>
> ### **Comparison Table**
>
> | **Aspect**                                                  | **Our Method**                                               | **Self-Refine**                                              | **Pair Self-Correction**                                     | **Reward-Model Verifier**                                    |
> | :---------------------------------------------------------- | :----------------------------------------------------------- | :----------------------------------------------------------- | :----------------------------------------------------------- | :----------------------------------------------------------- |
> | **Iteration costs for Construct Data**                      | Requires only general domain queries; uses Qwen-2.7B and Qwen-70B to construct training datasets without external feedback. | No training required.                                        | Requires few-shot prompting with a 175B model, generating both correct and incorrect answers for training. The process involves at least two inferences per query. | Requires 100 inferences times with a 175B model to generate n-sampling responses for higher accuracy, filtering for correct and incorrect answers for dataset construction. |
> | **Training Cost**                                           | 2 epochs on the base model.                                  | No training required.                                        | 2 epochs on the base model.                                  | 2 epochs on the verifier model.                              |
> | **Additional Data Construction and Training for new Tasks** | Not required.                                                | Requires task-specific prompts with ground truth to assist evaluation. | Requires constructing task-specific datasets of correct and incorrect samples. Limited to tasks that make it easy to distinguish the correctness. | Requires constructing task-specific datasets of correct and incorrect samples. Limited to tasks that make it easy to distinguish the correctness. |
> | **Need for External Feedback to Construct Data**            | Not required.                                                | Not applicable.                                              | Required correctness of answer.                              | Required correctness of answer.                              |
> | **Number of Iterations During Inference**                   | 2–3 iterations.                                              | 2–3 iterations.                                              | 2–3 iterations.                                              | 2–3 iterations with 100 inference samples per iteration to improve response accuracy. |
> | **Performance Improvement on GSM8K**                        | 2.9 /%.                                                      | 2.2 /%.                                                      | 2.4 /%.                                                      | 1.7 /%.                                                      |
> | **Performance on Unseen Tasks**                             | Overall improvement of 3.9 points.                           | Requires task-specific prompts and ground truth for evaluation. | Requires additional dataset construction and fine-tuning, which is costly for subjective tasks, and impractical. | Requires additional dataset construction and fine-tuning, which is costly for subjective tasks, and impractical. |
> | **Feedback During Inference**                               | Not required; generates an improved response regardless of initial correctness. | Requires feedback using ground truth to evaluate correctness. | Uses the model's self-verify to evaluate correctness. If incorrect, it iterates; if correct, it outputs the response. | Uses an additional verifier model to evaluate correctness. If incorrect, it iterates; if correct, it outputs the response. |
>
> ------
>
> [1] Self-refine：Self-refine: Iterative refinement with self-feedback https://arxiv.org/abs/2303.17651
>
> [2] Pair Self Correction： Generating Sequences by Learning to Self-Correct https://arxiv.org/abs/2211.00053
>
> [3] Reward-model Verifier: Training Verifiers to Solve Math Word Problems https://arxiv.org/abs/2110.14168

---

> > ### Author Response · Authors · 2024-11-25
> >
> > > Performance similar to other refine-work on math
> >
> > Experimental results show that performance improvements across methods are modest, typically within 3%. The *self-refine* approach relies on ground-truth feedback for slight gains; without it, performance often deteriorates. Similarly, Pair Self-Correction and Reward-Model Verifier achieve comparable improvements but remain limited.
> >
> > > General Task Improvement
> >
> > In contrast, my method achieves similar improvements and demonstrates effectiveness in other non-mathematical and non-reasoning tasks where their approach fails to tests and is hard to construct training datasets.  This highlights the limitation of the self-refine structure in effectively improving model performance without external guidance.
> >
> > The model can attain consistent performance improvements without relying on an ground truth. This showcases the effectiveness of our approach in comparison to the self-refine baseline, as it allows for more robust and consistent performance gains without the need for ground truth assistance and improves on more general tasks.
> >
> > > without external verifier or task-oriented fine-tuning
> >
> > - Constructing data for their approaches is relatively more challenging, and the data construction process takes significantly longer. This indirectly highlights the limitations of other methods, emphasizing that our contribution lies in not requiring domain-specific data construction.
> > - PTR does not require separate dataset construction or fine-tuning for each task, significantly reducing training costs.
> > - The Reward Model approach requires constructing positive and negative samples for each task and training a reward model to supervise the generation process. Similarly, the Pair Self-Correction method involves generating and labeling a large number of correct and incorrect sample pairs to achieve error correction capabilities. These processes are both costly and heavily dependent on specific domains.
> > - **Our method**, by employing a universal thought-mask strategy and leveraging the collaborative optimization of weak and strong models, does not rely on external labeled datasets, providing a clear advantage in terms of training costs.
> >
> > Additionally, while our method shows slightly lower absolute performance compared to some other methods, it proves effective across all tasks without relying on external verifiers or additional task-specific fine-tuning, which those methods require.
> >
> > > Inference Costs lower
> >
> > - The inference process of PTR is significantly less computationally expensive compared to other self-improvement methods, such as the Reward Model approach, which requires complex verifiers or multiple sampling and filtering steps.
> > - Specifically, as outlined in our response:
> >   - The **Reward Model Verifier** approach incurs high costs during data construction, as it requires best-of-n sampling and ground-truth answers to filter positive and negative examples. Additionally, its inference process is n times slower.
> >   - **Our method** requires only 2–3 iterations for inference and eliminates the need for external feedback, significantly reducing inference complexity.
> > - Furthermore, our data construction process requires only a single inference pass on a general dataset, allowing for performance improvements across all domains. In contrast, methods such as self-refine and Reward Model Verifier are task-oriented, requiring separate training datasets for different tasks and relying on domains like mathematics, where correctness can be clearly defined. These approaches face limitations in more open-ended domains. However, our method achieves improvements even in non-mathematical domains, highlighting its broader applicability and efficiency.
> >
> > Overall, our inference costs remain minimal, offering substantial advantages in terms of computational efficiency.

---

> ### Author Response · Authors · 2024-11-25
> **Re to ablation study**
>
> > Have the authors investigated the effect of different losses of equation 3.4 in an ablation study?
>
>
> 1. We conduct an ablation analysis to evaluate the impact of each hyperparameter.
> 2. Additionally, we also explore suggestions for setting these parameters, which could simplify the tuning process and make our method more accessible for broader applications.
>
>
>
> | ration        | iteraition | MMLU  | H-Eval | GSM8k | ARC   | Comm  | avg   |
> | ------------- | ---------- | ----- | ------ | ----- | ----- | ----- | ----- |
> | 1 0 0         | 1          | 60.9% | 51.2%  | 76.8% | 63.1% | 48.5% | 60.1% |
> |               | 2          | 62.3% | 53.0%  | 76.7% | 65.2% | 52.9% | 62.0% |
> |               | 3          | 62.7% | 52.4%  | 78.3% | 66.4% | 55.1% | 63.0% |
> | 0.9 0.05 0.05 | 1          | 60.9% | 51.8%  | 74.8% | 61.3% | 48.2% | 59.4% |
> |               | 2          | 62.6% | 55.5%  | 78.4% | 64.3% | 54.1% | 63.0% |
> |               | 3          | 62.5% | 56.1%  | 78.9% | 64.5% | 54.9% | 63.4% |
> | 0.8 0.1 0.1   | 1          | 58.5% | 51.2%  | 76.2% | 62.2% | 48.2% | 59.3% |
> |               | 2          | 63.3% | 55.5%  | 78.4% | 64.3% | 54.1% | 63.1% |
> |               | 3          | 63.9% | 56.7%  | 79.7% | 66.2% | 54.8% | 64.3% |
> | 0.7 0.15 0.15 | 1          | 56.3% | 51.8%  | 74.8% | 58.6% | 45.9% | 57.5% |
> |               | 2          | 58.3% | 52.4%  | 70.7% | 64.3% | 50.6% | 59.3% |
> |               | 3          | 57.9% | 52.4%  | 73.2% | 64.5% | 52.3% | 60.1% |
> | 0.6 0.2 0.2   | 1          | 54.8% | 48.2%  | 70.6% | 61.3% | 44.1% | 55.8% |
> |               | 2          | 55.6% | 47.6%  | 71.5% | 64.3% | 50.3% | 57.9% |
> |               | 3          | 55.7% | 48.2%  | 71.3% | 64.5% | 50.8% | 58.1% |
>
>
>
>
>
> In our experiments, we conducted sensitivity studies to evaluate the impact of these hyperparameters on model performance. The results indicate that different configurations of λ1,λ2, and λ3 significantly affect the balance between accuracy, reasoning consistency, and confidence distribution.
>
> We observed that setting the weights to λ1=0.8, λ2=0.1, and λ3=0.1 resulted in the highest average accuracy of 64.3%, effectively balancing final answer accuracy, reasoning consistency, and confidence distribution.
>
> From a broader application perspective, however, using λ1=1.0,λ2=0.0,λ3=0.0 (focusing solely on final answer accuracy) also achieved an average accuracy of 63.0%. To simplify training and reduce complexity for real-world scenarios, we recommend focusing solely on λ1, as it can still achieve competitive performance.
>
> We also identified the following findings:
>
> 1. **Impact of low λ1**: Experiments show that setting λ1 too low negatively impacts the accuracy of the final answer. For example, when λ1=0.6, the model’s training deviates, significantly reducing the accuracy of the initial response and diminishing its ability to self-refine. This indicates that λ1 should not be set too low in practice.
> 2. **Roles of λ2 and λ3**: Adding small weights for λ2 (reasoning consistency) and λ3 (confidence progression) supports the model’s ability to self-refine during reasoning, thereby improving overall performance. Logically, these parameters help the model refine its reasoning over time. However, their contribution to final accuracy is relatively limited compared to the cost of hyperparameter tuning and computational overhead. Therefore, unless the task explicitly requires higher interpretability or improved confidence in subsequent reasoning steps, excessive tuning of these parameters may not be worthwhile.
>
> For tasks requiring high reasoning consistency and confidence distribution optimization, we recommend λ1=0.8,λ2=0.1,λ3=0.1. However, for most real-world applications, the simpler configuration of λ1=1.0,λ2=0.0,λ3=0.0 is also sufficient.

---

> ### Author Response · Authors · 2024-11-26
> **Re to "PRD" dataset**
>
> > What is the "PRD" dataset mentioned in the paragraph of line 303?
>
> **We clarify the PTR dataset in line286**
> Our model has trained on our PTR (Progressive Thought Refinement) dataset, and derived queries from the WizardLM dataset[1]. After thorough cleaning in Section 3.1.1, we reduced the original dataset from approximately 50k QA pairs to 40k high-quality QA pairs.
>
> **More detail in Section 3.1.1 Query Preparation**
>
> Our filtering strategies include 3 parts:
>
> To enhance the model’s generalization, we avoid creating domain-specific datasets. Instead, we use queries from open-domain general datasets ensuring the model develops general refinement abilities rather than specializing in specific areas. Our data preprocessing involves three key steps. First, we perform data cleaning to remove noise and irrelevant content, such as images or URLs. Second, to prevent data leakage, we exclude domain-specific testing queries during training. Finally, we incorporate traditional SFT data (queries and answers generated by a strong model) into our dataset to mitigate the risk of catastrophic forgetting.
>
> **In line 203 Thought-Answer responses in section 3.1.2**
>
> We also construct  Thought-Answer responses, to construct thoughts via the weak model and final answer via the strong model.
> The objective is to ensure that the final answer is progressively improved through multiple iterations rather than relying on a single-step response.
>
> Through this approach, we aim to enable the model to learn the ability of self-refinement, gradually enhancing its answers from initial, imperfect responses to more refined ones. Our experiments demonstrate that, by using a self-improvement prompt, our model can significantly improve the quality of its answers. However, the base model lacks this self-improvement capability and even performs worse in multiple iterations. This is one of the key contributions of our paper.
>
> **In line 239 section 3.2 Progressive Weighted Thought-Mask Fine-tuning**
>
>
> and finally, our dataset is constructed  below:
>
>  $ D = { ( q_i, S_{i, thought},\hat{ y_{i,s,icl}} ) }_{i=1}^N $
>
>
> Consisting of the input query $q_i$, the initial thought sequence $S_{\text{i, thought}}$, and the final answer $\hat{ y_{i,s,icl}}$.
>
> [1] WizardLM: Empowering Large Language Models to Follow Complex Instructions

---

> ### Author Response · Authors · 2024-11-26
> **More clarification and Results for related work**
>
> ## 4. Avoiding Explicit Labeling for Dataset Construction
>
> Another significant innovation in our work is that we do not require explicitly labeled data (such as correct/incorrect labels) to train the model. Many existing methods rely on external feedback to construct labeled datasets, a process that can be costly and labor-intensive.
> In contrast, We use a strong-weak model collaborative filtering approach, which eliminates the need for external feedback to construct datasets, addressing the high costs of using external feedback for dataset construction. We believe that, in general, a stronger model performs better than a weaker one, and we have demonstrated this through the Wilcoxon Signed-Rank Test analysis in Appendix B.5. Even though our dataset does not provide explicit feedback or labeling for correctness, the model can still learn to improve itself through our datasets and training approach. The experiments confirm that the model can indeed refine its previous answers.
>
>
> ## 5. Practical Value of Self-Improvement in Real-World Scenarios
> Furthermore, since fine-tuning for each task is not a feasible solution, our method offers greater practical value when encountering new or difficult-to-annotate tasks. Our method offers a practical solution by enabling the model to self-improve across diverse tasks after fine-tuning. This reduces the need for task-specific fine-tuning, making it more feasible to apply our approach in real-world settings. Our method demonstrates that models can self-refine even when encountering tasks they have never seen before, enhancing their utility and applicability in dynamic environments.
>
> In summary, we acknowledge that claiming intrinsic self-improvement ability is a significant assertion; however, we believe our detailed investigations and experiments provide strong evidence that such an ability can be activated and leveraged through fine-tuning. Our work does not merely improve model performance in specific tasks but empowers the model to refine its answers across a wide range of tasks, even those it has not encountered during training. This is a key contribution to the field, offering both practical and cost-effective benefits for real-world applications.
>
>
>
> Below are the experimental results comparing performance on mathematical tasks.
>
> |               Method                | GSM8K | MATH |
> | :---------------------------------: | :---: | :--: |
> |      Self-refine[1] iteration1      | 79.1  | 48.7 |
> |    with ground truth  iteration2    | 81.3  | 50.1 |
> |    w/o ground truth  iteration2     | 74.7  | 48.4 |
> |                                     |       |      |
> | Pair Self Correction[2] iteration1  | 77.7  | 48.2 |
> |   Pair Self Correction iteration2   | 80.1  | 49.5 |
> |                                     |       |      |
> | Reward-model Verifier[3] iteration1 | 80.0  | 48.1 |
> |  Reward-model Verifier iteration2   | 81.7  | 49.2 |
> |                                     |       |      |
> |           Ours iteration1           | 76.7  | 47.6 |
> |           Ours iteration2           | 79.6  | 48.9 |
>
> We compare our method to prior and baseline approaches, evaluating performance on GSM8K and MATH.
>
>
> We sincerely thank the reviewer once again for their valuable comments and constructive feedback. We have worked diligently to address all concerns and improve the clarity, quality, and depth of our work. We hope that our revisions meet your expectations and address the issues raised. We are open to further discussion and would greatly appreciate any additional feedback to enhance the manuscript. Your insights have been instrumental in refining our work, and we look forward to the possibility of an improved evaluation. Thank you for your time and thoughtful review.

---

> > ### Author Response · Authors · 2024-11-26
> > **Detail for related work**
> >
> > - **Self-refine**[1] does not require training but relies on standard answers to assist reasoning. It uses specific math-refine prompts to guide a base model in critiquing and revising its mistakes.
> >   - *"With ground truth"* refers to a setup where the model checks the correctness of its response only if the initial answer is wrong. If incorrect, the model generates a new response in a second iteration and stops as soon as the correct answer is predicted.
> >   - *"Without ground truth"* refers to a setup where the model always refines its answer without verifying correctness.
> >
> > > The specific math refine prompt is :
> > >
> > > if the answer is wrong:
> > >
> > > ```
> > > There is an error in the math above because of a lack of understanding of the question. What is the error? To find the error, go through semantically complete blocks of the code, and check if everything looks good.
> > > Let us go through the error and check step-by-step
> > > ```
> > >
> > > if the answer is right:
> > >
> > > ```
> > > looks good
> > > ```
> >
> >
> >
> >
> >
> > - **Pair Self Correction**[2] trains a single model to integrate both self-verify and generation capabilities.
> >
> > In the data construction phase, the original study used **Davinci (≈175B)** and **text-davinci-002 (≈175B)** models with few-shot prompting to construct the dataset. In our work, we substituted these models with the **Qwen2-70B** model for data generation (same as our methods settings).
> >
> > The method involves fine-tuning a large model using pairs of correct and incorrect mathematical solutions. This enables the model to learn to correct mistakes effectively. The model first diagnoses its initial output: if the self-diagnosis determines the answer is correct, no changes are made; if the answer is incorrect, the model revises the response accordingly.
> >
> >
> >
> > - **Reward-model Verifier** [3] requires training a Verifier to assess the correctness of model-generated answers. And inference on another base model.
> >
> > During the data construction phase, the original study used a 175B model to generate datasets, as explained in paper that larger models typically perform better on GSM8K(Here we use  **Qwen2-70B** model , the same settings with our methods). Labels were used to filter correct and incorrect responses. These inputs, along with their corresponding outputs and correctness judgments, were used to train the model to distinguish between correct and incorrect answers.
> >
> > The 175B model samples 100 answers for each training question. These answers were labeled as correct or incorrect (based on whether they reached the correct final answer) and used to train the reward model for 2 epochs.
> >
> > During inference, the base model model (here we use when 7b same as our methods) samples 100 responses for each question. If the initial response is incorrect, the model iteratively refines and generates new answers. This process is guided by the reward model’s scoring system, judging the output correctness.
> >
> >
> >
> > [1] Self-refine：Self-refine: Iterative refinement with self-feedback https://arxiv.org/abs/2303.17651
> >
> > [2] Pair Self Correction： Generating Sequences by Learning to Self-Correct https://arxiv.org/abs/2211.00053
> >
> > [3] Reward-model Verifier: Training Verifiers to Solve Math Word Problems https://arxiv.org/abs/2110.14168

---

### Official Review · Reviewer_DjDQ · 2024-11-04

**Soundness:** 3
**Presentation:** 3
**Contribution:** 2
**Rating:** 8
**Confidence:** 3

**Summary:**

This paper introduces Progressive Thought Refinement (PTR), a framework designed to enable LLMs to iteratively improve their responses through self-refinement. PTR operates in two phases: (1) a thought data construction phase, using a weak-strong model collaboration to build high-quality thought-answer pairs, and (2) a weighted thought-mask fine-tuning phase, which encourages models to enhance responses across multiple iterations. Experimental results demonstrate PTR's effectiveness across ten diverse tasks.

**Strengths:**

1. The paper presents a novel framework, PTR, that enables LLMs to iteratively self-refine. The concept of self-refine/correct is a very important research direction and has attracted many recent research efforts. The paper has shown reasonable effectiveness and performance improvement in experiments, with a new and interesting framework to achieve this.

2. The two-phase methodology, combining weak-strong model collaboration with thought-mask fine-tuning, is robustly validated across ten diverse datasets. While the demand for strong model is costly, it remains a reasonable approach in application.

3. The paper is well-organized, with clear visuals and structured explanations. The paper also provided a thorough discussion on related concepts and papers.

**Weaknesses:**

- The function $F_{\text{cons}}$ and $\beta_t$ in equation 3.4 is not clearly defined. Clear definitions for the components of the equation are needed in the main paper.

- Hyperparameter complexity. while hyperparameters $\lambda_1$, $\lambda_2$, $\lambda_3$ help balance various aspects of the model's loss function, extensive tuning could limit the practical adoption of PTR. The paper would benefit from a sensitivity analysis or heuristics for setting these parameters to make the method more accessible for broader application.

- Computational overhead. The reliance on a 70B model and the multi-iteration inference adds significant computational demands for both training and inference. It would be beneficial for the authors to discuss the cost of this method (the trade-offs in performance versus efficiency).

- Despite the complexity of the PTR procedure, the performance improvement over direct prompting (iteration 1 baseline) is relatively modest. On average, PTR achieves only a 1.88% improvement by iteration 3 for Qwen. For Llama3, the most significant gains are seen in the CommonsenseQA dataset, whereas tasks such as mathematical reasoning show only minimal benefits.

- Minor typos (e.g., correct alignment of PTR and Prompt1/2/3 in Tables 1 and 2).

**Questions:**

See above in weaknesses.

---

> ### Author Response · Authors · 2024-11-21
> **Answer to Q1 Q2**
>
> > Q1 clearly defined on 3.4 equation in main paper
>
> We acknowledge that the components of Equation 3.4 could benefit from more explicit definitions in main pages. We clarify each element of the equation to ensure it is comprehensible. We clarify each element of the equation to ensure it is comprehensible. Due to space constraints, more details can be included in the appendix.
>
>
>
> > Q2 broader application advice for hyperparameter complexity, ablation study
>
> We appreciate your observation regarding the hyperparameter complexity. Indeed, balancing the model's loss function with multiple hyperparameters can introduce challenges.
>
> 1. We conduct a sensitivity analysis to evaluate the impact of each hyperparameter.
> 2. Additionally, we also explore suggestions for setting these parameters, which could simplify the tuning process and make our method more accessible for broader applications.
>
>
>
> | ration        | iteraition | MMLU  | H-Eval | GSM8k | ARC   | Comm  | avg   |
> | ------------- | ---------- | ----- | ------ | ----- | ----- | ----- | ----- |
> | 1 0 0         | 1          | 60.9% | 51.2%  | 76.8% | 63.1% | 48.5% | 60.1% |
> |               | 2          | 62.3% | 53.0%  | 76.7% | 65.2% | 52.9% | 62.0% |
> |               | 3          | 62.7% | 52.4%  | 78.3% | 66.4% | 55.1% | 63.0% |
> | 0.9 0.05 0.05 | 1          | 60.9% | 51.8%  | 74.8% | 61.3% | 48.2% | 59.4% |
> |               | 2          | 62.6% | 55.5%  | 78.4% | 64.3% | 54.1% | 63.0% |
> |               | 3          | 62.5% | 56.1%  | 78.9% | 64.5% | 54.9% | 63.4% |
> | 0.8 0.1 0.1   | 1          | 58.5% | 51.2%  | 76.2% | 62.2% | 48.2% | 59.3% |
> |               | 2          | 63.3% | 55.5%  | 78.4% | 64.3% | 54.1% | 63.1% |
> |               | 3          | 63.9% | 56.7%  | 79.7% | 66.2% | 54.8% | 64.3% |
> | 0.7 0.15 0.15 | 1          | 56.3% | 51.8%  | 74.8% | 58.6% | 45.9% | 57.5% |
> |               | 2          | 58.3% | 52.4%  | 70.7% | 64.3% | 50.6% | 59.3% |
> |               | 3          | 57.9% | 52.4%  | 73.2% | 64.5% | 52.3% | 60.1% |
> | 0.6 0.2 0.2   | 1          | 54.8% | 48.2%  | 70.6% | 61.3% | 44.1% | 55.8% |
> |               | 2          | 55.6% | 47.6%  | 71.5% | 64.3% | 50.3% | 57.9% |
> |               | 3          | 55.7% | 48.2%  | 71.3% | 64.5% | 50.8% | 58.1% |
>
>
>
> **Hyperparameter Complexity**: While hyperparameters λ1,λ2,λ3 help balance various aspects of the model's loss function, extensive tuning hyperparameters could limit PTR's practical adoption.
>
> In our experiments, we conducted ablation studies to evaluate the impact of these hyperparameters on model performance. The results indicate that different configurations of λ1,λ2,λ3 significantly affect the balance between accuracy, reasoning consistency, and confidence distribution.
>
> We observed that setting the weights to λ1=0.8,λ2=0.1,λ3=0.1 resulted in the highest average accuracy of 64.3%, effectively balancing final answer accuracy, reasoning consistency, and confidence distribution.
>
> From a broader application perspective, however, using λ1=1.0,λ2=0.0,λ3=0.0 (focusing solely on final answer accuracy) also achieved an average accuracy of 63.0%. To simplify training and reduce complexity for real-world scenarios, we recommend focusing solely on λ1, as it can still achieve competitive performance.
>
> We also identified the following findings:
>
> 1. **Impact of low λ1**: Experiments show that setting λ1 too low negatively impacts the accuracy of the final answer. For example, when λ1=0.6, the model’s training deviates, significantly reducing the accuracy of the initial response and diminishing its ability to self-refine. This indicates that λ1 should not be set too low in practice.
> 2. **Roles of λ2 and λ3**: Adding small weights for λ2 (reasoning consistency) and λ3 (confidence progression) supports the model’s ability to self-refine during reasoning, thereby improving overall performance. Logically, these parameters help the model refine its reasoning over time. However, their contribution to final accuracy is relatively limited compared to the cost of hyperparameter tuning and computational overhead. Therefore, unless the task explicitly requires higher interpretability or improved confidence in subsequent reasoning steps, excessive tuning of these parameters may not be worthwhile.
>
> For tasks requiring high reasoning consistency and confidence distribution optimization, we recommend λ1=0.8,λ2=0.1,λ3=0.1. However, for most real-world applications, the simpler configuration of λ1=1.0,λ2=0.0,λ3=0.0 is also sufficient.
>
> We will include the ablation study results and further recommendations in the appendix of the next version of our paper.

---

> ### Author Response · Authors · 2024-11-21
> **Answer to Q3 Q4**
>
> > Q4 modest improvements
>
> 1. **Key Contribution**:
>    The primary contribution of our work lies in enabling models to develop self-refinement capabilities through training. Our experiments show that without training, open-source models tend to worsen their initial answers when attempting to refine them. However, our training method equips the model with the ability to refine its responses effectively. The activation of self-refinement capabilities in models is an exciting development, and we plan to explore methods to further enhance the magnitude of improvement in future work.
>
>    **Generalization Across Domains**:
>    Even in related self-refinement approaches, the improvements achieved through fine-tuning in mathematical domains are relatively modest. In contrast, our method demonstrates improvements across a broader range of domains. For instance, the overall performance of our model increases from 49.6% to 53.5%. This improvement in the model's self-refinement ability, particularly its generalization across diverse fields, is a major contribution to our work.
>
>
>
> > Q3 cost of this method and our main contribution
>
> Thanks for pointing out the problem of training costs, here I will further elaborate on the reasons and compare other approaches. Other methods have higher computational costs, but they don't perform as well as ours.
>
> We did a comparison to previous work in the field of mathematics in Reviewer 9DXQ **Q2** responses, our model also has several strengths below:
>
> > Constructing data Costs lower
>
> PTR does not require separate dataset construction or fine-tuning for each task, significantly reducing costs.
>
> Constructing data for other approaches is relatively more challenging, and the data construction process takes significantly longer. The **Reward Model Verifier** approach incurs high costs during data construction, as it requires best-of-n sampling and ground-truth answers to filter positive and negative examples. Additionally, its inference process is n times slower.
>
>
> Additionally, while our method shows slightly lower absolute performance compared to some other methods, it proves effective across all tasks without relying on external verifiers or additional task-specific fine-tuning, which those methods require.
>
>
>
> > Inference Costs lower
>
> - As outlined in our response to reviewer 9DXQ:
>
>   - The **Reward Model Verifier** approach incurs extra costs during inference, as it requires verifier and ground-truth answers to check the responses.
>   - **Our method** requires only 2–3 iterations for inference and eliminates the need for external feedback, significantly reducing inference complexity.
>
>
>
> > Training Costs lower
>
> - our data construction process requires only a single inference pass on a general dataset, allowing for performance improvements across all domains.
>
> -  In contrast, methods such as self-refine and Reward Model Verifiers are task-oriented, requiring separate training datasets for different tasks and relying on domains like mathematics, where correctness can be clearly defined.
>
> - These approaches face limitations in more open-ended domains. However, our method achieves improvements even in non-mathematical domains, highlighting its broader applicability and efficiency.
>
>
>
> We sincerely thank the reviewer for their thoughtful feedback and encouraging comments on our work. We have carefully addressed each concern and provided detailed clarifications and additional insights in our responses. We hope these revisions meet your expectations and effectively highlight the contributions of our work. Your insights have been invaluable in refining the manuscript, and we welcome any further feedback or suggestions to improve it further. We look forward to the possibility of an improved evaluation and continued discussion. Thank you again for your time and constructive review!

---

> > ### Comment · Reviewer_DjDQ · 2024-11-25
> > **Thank you for the rebuttal**
> >
> > Thank you for the rebuttal and the additional sensitivity analysis. I appreciate the effort. However, the rebuttal is disorganized and somewhat difficult to follow, as it frequently directs to answers provided to other reviewers. Since ICLR rebuttals do not have a character limit, I suggest elaborating on these points directly in the reply or at least provide a link to the relevant response for easier navigation. Regarding specific points:
> >
> > > We clarify each element of the equation to ensure it is comprehensible. We clarify each element of the equation to ensure it is comprehensible. Due to space constraints, more details can be included in the appendix.
> >
> > I can not locate these clarifications or revisions in either the main paper or the appendix. Can you point me to where these details are provided?
> >
> > > We did a comparison to previous work in the field of mathematics in Reviewer 9DXQ Q2 responses
> >
> > Can you clarify which baselines were compared with references to the paper? Providing explicit references to each method would be beneficial. For example, does "Reward Model Verifier" refer to [1]? If so, my understanding is that their method does not require a 70B model as the method proposed in this paper does. If the authors aim to claim a superior cost-performance trade-off, it's necessary to rigorously compare these aspects (for example include a detailed table to compare all methods on cost), with each baseline method clearly cited.
> >
> > [1] Generative Verifiers: Reward Modeling as Next-Token Prediction

---

> > > ### Author Response · Authors · 2024-11-25
> > >
> > > Thank you for your timely feedback! We will reorganize the rebuttal promptly and provide more detailed responses to each reviewer’s comments.

---

> > > ### Author Response · Authors · 2024-11-25
> > > **Clarify baselines with references to the paper**
> > >
> > > Below are the experimental results comparing performance on mathematical tasks.
> > >
> > > |               Method                | GSM8K | MATH |
> > > | :---------------------------------: | :---: | :--: |
> > > |      Self-refine[1] iteration1      | 79.1  | 48.7 |
> > > |    with ground truth  iteration2    | 81.3  | 50.1 |
> > > |    w/o ground truth  iteration2     | 74.7  | 48.4 |
> > > |                                     |       |      |
> > > | Pair Self Correction[2] iteration1  | 77.7  | 48.2 |
> > > |   Pair Self Correction iteration2   | 80.1  | 49.5 |
> > > |                                     |       |      |
> > > | Reward-model Verifier[3] iteration1 | 80.0  | 48.1 |
> > > |  Reward-model Verifier iteration2   | 81.7  | 49.2 |
> > > |                                     |       |      |
> > > |           Ours iteration1           | 76.7  | 47.6 |
> > > |           Ours iteration2           | 79.6  | 48.9 |
> > >
> > > We compare our method to prior and baseline approaches, evaluating performance on GSM8K and MATH.
> > >
> > > - **Self-refine**[1] does not require training but relies on standard answers to assist reasoning. It uses specific math-refine prompts to guide a base model in critiquing and revising its mistakes.
> > >   - *"With ground truth"* refers to a setup where the model checks the correctness of its response only if the initial answer is wrong. If incorrect, the model generates a new response in a second iteration and stops as soon as the correct answer is predicted.
> > >   - *"Without ground truth"* refers to a setup where the model always refines its answer without verifying correctness.
> > >
> > > > The specific math refine prompt is :
> > > >
> > > > if the answer is wrong:
> > > >
> > > > ```
> > > > There is an error in the math above because of a lack of understanding of the question. What is the error? To find the error, go through semantically complete blocks of the code, and check if everything looks good.
> > > > Let us go through the error and check step-by-step
> > > > ```
> > > >
> > > > if the answer is right:
> > > >
> > > > ```
> > > > looks good
> > > > ```
> > >
> > >
> > >
> > >
> > >
> > > - **Pair Self Correction**[2] trains a single model to integrate both self-verify and generation capabilities.
> > >
> > > In the data construction phase, the original study used **Davinci (≈175B)** and **text-davinci-002 (≈175B)** models with few-shot prompting to construct the dataset. In our work, we substituted these models with the **Qwen2-70B** model for data generation (same as our methods settings).
> > >
> > > The method involves fine-tuning a large model using pairs of correct and incorrect mathematical solutions. This enables the model to learn to correct mistakes effectively. The model first diagnoses its initial output: if the self-diagnosis determines the answer is correct, no changes are made; if the answer is incorrect, the model revises the response accordingly.
> > >
> > >
> > >
> > > - **Reward-model Verifier** [3] requires training a Verifier to assess the correctness of model-generated answers. And inference on another base model.
> > >
> > > During the data construction phase, the original study used a 175B model to generate datasets, as explained in paper that larger models typically perform better on GSM8K(Here we use  **Qwen2-70B** model , the same settings with our methods). Labels were used to filter correct and incorrect responses. These inputs, along with their corresponding outputs and correctness judgments, were used to train the model to distinguish between correct and incorrect answers.
> > >
> > > The 175B model samples 100 answers for each training question. These answers were labeled as correct or incorrect (based on whether they reached the correct final answer) and used to train the reward model for 2 epochs.
> > >
> > > During inference, the base model model (here we use when 7b same as our methods) samples 100 responses for each question. If the initial response is incorrect, the model iteratively refines and generates new answers. This process is guided by the reward model’s scoring system, judging the output correctness.
> > >
> > >
> > >
> > > [1] Self-refine：Self-refine: Iterative refinement with self-feedback https://arxiv.org/abs/2303.17651
> > >
> > > [2] Pair Self Correction： Generating Sequences by Learning to Self-Correct https://arxiv.org/abs/2211.00053
> > >
> > > [3] Reward-model Verifier: Training Verifiers to Solve Math Word Problems https://arxiv.org/abs/2110.14168

---

> > > > ### Author Response · Authors · 2024-11-25
> > > > **Cost-performance for each methods**
> > > >
> > > > We present a detailed comparison of our method with existing approaches, highlighting the key differences in cost, applicability, and performance across tasks such as GSM8K and general downstream tasks.
> > > >
> > > > ### **Comparison Table**
> > > >
> > > > | **Aspect**                                                  | **Our Method**                                               | **Self-Refine**                                              | **Pair Self-Correction**                                     | **Reward-Model Verifier**                                    |
> > > > | ----------------------------------------------------------- | ------------------------------------------------------------ | ------------------------------------------------------------ | ------------------------------------------------------------ | ------------------------------------------------------------ |
> > > > | **Iteration costs for Construct Data**                      | Requires only general domain queries; uses Qwen-2.7B and Qwen-70B to construct training datasets without external feedback. | No training required.                                        | Requires few-shot prompting with a 175B model, generating both correct and incorrect answers for training. The process involves at least two inferences per query. | Requires 100 inferences times with a 175B model to generate n-sampling responses for higher accuracy, filtering for correct and incorrect answers for dataset construction. |
> > > > | **Training Cost**                                           | 2 epochs on the base model.                                  | No training required.                                        | 2 epochs on the base model.                                  | 2 epochs on the verifier model.                              |
> > > > | **Additional Data Construction and Training for new Tasks** | Not required.                                                | Requires task-specific prompts with ground truth to assist evaluation. | Requires constructing task-specific datasets of correct and incorrect samples. Limited to tasks that make it easy to distinguish the correctness. | Requires constructing task-specific datasets of correct and incorrect samples. Limited to tasks that make it easy to distinguish the correctness. |
> > > > | **Need for External Feedback to Construct Data**            | Not required.                                                | Not applicable.                                              | Required correctness of answer.                              | Required correctness of answer.                              |
> > > > | **Number of Iterations During Inference**                   | 2–3 iterations.                                              | 2–3 iterations.                                              | 2–3 iterations.                                              | 2–3 iterations with 100 inference samples per iteration to improve response accuracy. |
> > > > | **Performance Improvement on GSM8K**                        | 2.9 /%.                                                      | 2.2  /%.                                                     | 2.4  /%.                                                     | 1.7  /%.                                                     |
> > > > | **Performance on Unseen Tasks**                             | Overall improvement of 3.9 points.                           | Requires task-specific prompts and ground truth for evaluation. | Requires additional dataset construction and fine-tuning, which is costly for subjective tasks, and impractical. | Requires additional dataset construction and fine-tuning, which is costly for subjective tasks, and impractical. |
> > > > | **Feedback During Inference**                               | Not required; generates an improved response regardless of initial correctness. | Requires feedback using ground truth to evaluate correctness. | Uses the model's self-verify to evaluate correctness. If incorrect, it iterates; if correct, it outputs the response. | Uses an additional verifier model to evaluate correctness. If incorrect, it iterates; if correct, it outputs the response. |
> > > >
> > > > ---
> > > >
> > > > ###

---

> > > > > ### Author Response · Authors · 2024-11-25
> > > > >
> > > > > ### **Advantages of Our Approach**
> > > > >
> > > > > #### 1. **Lower Cost**:
> > > > >
> > > > >    - Both *Pair Self-Correction* and *Reward-Model Verifier* rely on large-scale models (e.g., 175B parameters) to generate correct and incorrect answers for dataset construction. *Reward-Model Verifier* further requires 100 inference samples per query to improve the probability of a correct answer.
> > > > >    - In contrast, our method avoids stringent requirements for model accuracy during data generation. We use a strong model (Qwen-70B) and a weaker model (Qwen-2.7B) to generate data with only two inferences, allowing the model to learn self-improvement from the responses of both.
> > > > >
> > > > > #### 2. **Broader Applicability**:
> > > > >
> > > > >    - Existing approaches are limited to tasks with clear, objective correctness criteria, such as mathematical problems. They require task-specific datasets and fine-tuning, which are costly or infeasible for subjective or open-ended tasks.
> > > > >    - Our method, however, maintains strong adaptability across diverse tasks without additional fine-tuning. This enables the model to generalize effectively to unseen tasks.
> > > > >
> > > > > ---
> > > > >
> > > > > ### **Key Contributions of Our Work**
> > > > >
> > > > > 1. **Eliminating the Need for External Feedback**:
> > > > >    - Our method addresses the high cost of dataset construction by removing the reliance on external feedback for labeling correctness. Instead, the model learns self-improvement from general domain queries and the interactions between strong and weak models.
> > > > >    - Even without explicit correctness labels, our datasets enable the model to improve previous answers. Wilcoxon signed-rank tests (Appendix B.5) confirm that strong models consistently outperform weaker models, allowing the dataset to drive self-correction effectively.
> > > > >
> > > > > 2. **Practical Value Across Tasks**:
> > > > >    - Unlike task-oriented methods, which require fine-tuning for each specific task, our approach is highly practical for new or difficult-to-annotate tasks. The model undergoes a single fine-tuning step, after which it demonstrates self-improvement across all unseen tasks.
> > > > >
> > > > > 3. **Real-World Adaptability**:
> > > > >    - Our method reduces the reliance on clear correctness criteria, making it more accessible and effective in real-world scenarios where explicit answers are not always available. By fostering self-improvement, our approach ensures performance gains across diverse, unseen domains—a contribution that other methods cannot achieve.
> > > > >
> > > > > ---
> > > > >
> > > > >
> > > > >
> > > > > Our approach not only achieves comparable accuracy improvements in mathematical tasks but also significantly reduces inference and data construction costs. Furthermore, it extends beyond task-specific limitations, providing a scalable and adaptive solution for a wide range of real-world applications. This makes our method a more versatile and impactful contribution to self-corrective learning.

---

> > > ### Author Response · Authors · 2024-11-25
> > > **Additional notes: cost for 70B model**
> > >
> > > Regarding the cost-performance trade-off, we also would like to clarify an important point: while our method uses a 70B model for data construction, we rely on open-source models, which are free to use in our local server. In contrast, related work often relies on paid APIs, such as OpenAI's GPT models, to achieve higher correctness by generating datasets or performing inferences. This distinction makes our approach significantly more cost-efficient in terms of practical implementation.
> > >
> > > To address this, we included a detailed table comparing all methods. This comparison will underscore the cost-effectiveness of our method, particularly for researchers and practitioners constrained by resource budgets. Thank you again for this suggestion—it will help us make our contributions clearer and more impactful.

---

> > > > ### Comment · Reviewer_DjDQ · 2024-12-01
> > > > **Thank you for elaborating**
> > > >
> > > > Thank you for the more detailed follow-up and clarifications. I recommend that the authors incorporate the additional discussion on baselines and the comparison table into the revision, as this would enhance the clarity of this work's contributions. The rebuttal has addressed my concerns, and I updated my score accordingly.

---

> > > > > ### Author Response · Authors · 2024-12-02
> > > > >
> > > > > Thank you for your thoughtful and detailed feedback, as well as for raising your score to 8. We greatly appreciate your support and the time you dedicated to reviewing our work. Your decision to increase the score has given us significant confidence, and we are motivated to continue improving. We will incorporate further discussion into the revision to strengthen the clarity of our contributions. We are grateful for your constructive feedback, which has significantly improved the quality of our work. If you have any additional comments or questions, please feel free to reach out. Thank you once again for your valuable support.

---

> ### Author Response · Authors · 2024-11-25
> **Re to revision of equation**
>
> If these clarifications or revisions pdf are not visible in the current stages, I apologize for the oversight. It’s possible that the revised edition is not yet accessible to reviewers at this stage. To address this, I will provide the relevant details here for your reference.
>
> # Main paper line:259
>
> $L_{PTR}(\theta) = \sum\limits_{(q_i, S_{i, \text{thought}}, y_n) \in \tilde{D}} = \Bigg[ -\lambda_1 \log \Pr(y_n \mid q_i, S_{i,\text{thought}}; \theta) \nonumber$
> $+ \lambda_2 \sum\limits_{t=2}^{n} F_{\text{cons}}(y_t, y_{t-1}) + \lambda_3 \sum\limits_{t=1}^{n} \beta_t \left(1 - \Pr(y_t \mid q_i, S_{i,\text{thought}}; \theta)\right) \Bigg].$
>
>
>
>
> Unlike standard supervised fine-tuning, which trains the model to produce a single response $\hat{y}$ given $x$,
>
> $-\lambda_1 \log \Pr(y_n \mid q_i, S_{i,\text{thought}}; \theta)$ focuses exclusively on the accuracy of the final response generated after the thought refinement process.
>
> $F_{\text{cons}}(y_t, y_{t-1})$ ensures that the current response remains logically consistent with the previous thought sequence by computing the Cosine Similarity with the Sentence-BERT [1] (See Appendix A.2)
>
> $\left(1 - \Pr(y_t \mid q_i, S_{i,\text{thought}}; \theta)\right) $ represents the model’s uncertainty or error probability at each reasoning step, which measures the confidence of the model’s predictions. The term $\beta_t $ represents the confidence score at each reasoning step, which increases as reasoning progresses to encourage higher certainty in later steps (see Appendix A.3).
>
> Here, $\lambda_1$, $\lambda_2$, and $\lambda_3$ are dynamically adjusted according to the model's needs. We set λ1=0.8, λ2=0.1, and λ3=0.1, with their sum weighted to 1.
>
>
>
>
>
> # Appendix A.2 line 773
>
> **A.2 Consistency Compute**
>
> To evaluate the consistency between the thought sequences and the final answer, we vectorize the thoughts and the answer using **sentence-BERT embeddings**. Sentence-BERT provides an effective way to embed both the thought sequences and the final answer into a shared vector space, capturing semantic similarities between them.
>
> The similarity between each thought  $y_t$ and the prior thought $y_{t−1}$ is computed using the cosine similarity between their Sentence-BERT embeddings. The consistency score is designed to capture how well the current thought $y_t$ is consistent with the prior thought  $y_{t−1}$ and how closely it relates to the final answer.
>
> We define a **Consistency Function** $F_{\text{cons}}(y_t, y_{t-1})$ as the cosine similarity between the current thought $y_t$ and the prior thought $y_{t−1}$. The function is computed as:
> $$
> F_{\text{cons}}(y_t, y_{t-1}) = \text{CosineSimilarity}(\text{BERT}(y_t), \text{BERT}(y_{t-1}))
> $$
> Where  $ \text{CosineSimilarity}(a, b)$ is defined as:
> $$
> \text{CosineSimilarity}(a, b) = \frac{a \cdot b}{\|a\| \|b\|}
> $$
> Here, a and b are the embeddings of $y_t$ and $y_{t−1}$ respectively, generated by the Sentence-BERT model. The cosine similarity measures the angle between the two vectors in the embedding space, with values closer to 1 indicating high similarity and values closer to 0 indicating low similarity.
>
> **A.3 Dynamic Confidence Adjustment**
>
> By introducing a dynamic confidence decay strategy, the model can gradually increase its confidence during the reasoning process. For example, the confidence $\beta_t$ can increase as the reasoning step tt progresses, instead of staying constant throughout the process. This will allow the model to gain more confidence as it refines its thoughts and reasoning.
>
> By adjusting the weights of each reasoning step, $\beta_t$ can control how the model’s confidence evolves. At the initial steps, the model can have a lower confidence since the thought process is still being refined. As the reasoning progresses, the model should gradually increase its confidence and have higher confidence at the final step. This can be achieved by adjusting  $\beta_t$ dynamically like so:
>
>
> $$
> \beta_t = \beta_0 \cdot \left(\frac{t}{n}\right)
> $$
>
>
> where  $\beta_0$ is the initial confidence and n is the total number of reasoning steps. This approach ensures that the confidence $\beta_t$  increases as reasoning progresses.

---

### Official Review · Reviewer_9DXQ · 2024-11-04

**Soundness:** 3
**Presentation:** 3
**Contribution:** 3
**Rating:** 6
**Confidence:** 3

**Summary:**

Previous methods of self-improvement tried to enhance performance within specific domains, because supervision signals are vague and hard to define and evaluate response quality in open-ended tasks. However, their key limitation lies in lack of generalizability due to high reliance on task-specific pipelines or reward models. Hence, the authors introduce PTR method, which consists of “thought data construction phase” and “weighted thought-mask fine-tuning phase”.

**Strengths:**

- It's easy to follow, and clear writing and presentations (including nice figures.)
- Performance improvement over iterations looks very good.
- Extensive experimental results and analysis of the proposed methods, particularly in Sections 4.3 - 4.5, and qualitative examples.

**Weaknesses:**

I have some concerns, and ask the authors to explain and mitigate these.

- What are the main contributions and novel points of this paper over some prior refinement works? Weak-strong model to construct refinement data without using rewards, and thought-mask strategy?
- If yes, I wonder why the authors do not compare with prior works on refinement method or approaches with verifiers. There are so many related works (as also mentioned in Related Work sections), but three methods that the authors showed look very naive. If extra feedbacks are not proper for some datasets, I think comparing in math or code reasoning, where we can easily define the rewards or feedbacks by comparing with ground truth, would be needed at least.
- As a followup question, why do all three baselines showed worse performance after iterations? I believe these baselines are not good and not well-designed, which makes it very hard to confirm the contributions of this paper.
- As I understood, thought-mask strategy is using some training dataset with masked on thought parts. How were the results when you didn't use this masking strategy, and only used refinement loss parts? I believe IFT baseline is not this case---if yes, what is the IFT method exactly? (in L.304)
- What were the experimental settings for fine-tuning? If it costs a lot, would it be better to use some extra feedbacks during inference in perspective of test-time compute efficiency?

**Questions:**

- To be clear, what do the authors mean about "extra feedback" exactly? Using some other tools, human or compiler feedback, or verifiers (reward models) during *inference* means extra feedback? But this method trained models that can refine their answers without any other feedbacks during inference (though they used extra feedbacks from other models during *training*.)

---

> ### Author Response · Authors · 2024-11-21
> **Answer to Q1 Q3 Q6**
>
> > Q1 main contributions and novel points of this paper over some prior refinement works
>
> Prior works typically rely on external feedback to assess the quality of model responses and to construct labeled datasets for training. This approach is costly, especially for open-ended tasks where evaluating generated responses is challenging. As a result, most benchmarks in previous work have been confined only to mathematical or coding tasks. In contrast, our goal is to empower the model's self-improvement capabilities, allowing it to continually refine itself without the need for rewards or explicit correctness indicators. This method enables the low-cost and large-scale construction of datasets, ultimately improving the model's performance in all fields.
>
> Additionally, earlier self-improvement approaches, such as designing prompts or fine-tuning with specific preference datasets, allow the model to improve in certain domains. However, this comes with significant challenges when facing unseen tasks. Limited generalization beyond their training domains.
>
> In contrast, our method does not rely on domain-specific datasets and no labeling for correctness. Our experimental results confirm that this approach can enhance the model's capability without being limited by the task-specific fine-tuning required by previous methods.
>
>
>
> > Q3 our baseline showed worse performance
>
> Our contribution lies in demonstrating that our reasoning mask strategy and the optimization of the dataset using strong and weak model pairs can effectively unlock the model's self-improvement capabilities. This self-improvement ability is then exhibited consistently across all tasks.
>
> The baseline models used for comparison include an initial model without fine-tuning, which serves to demonstrate that models lack inherent self-improvement capabilities without additional fine-tuning. Additionally, we compared against an SFT (supervised fine-tuning) model, which is explained in detail in the experimental section. The SFT model was fine-tuned using answers directly taken from a strong model, aimed at proving that our dataset does not suffer from data leakage.
>
> Moreover, our training approach goal is not about memorizing the correct answers to specific questions but about enabling the model to learn how to iteratively refine its responses.
>
>
>
> > Q6 misunderstanding of extra feedback
>
> There seems to be some misunderstanding here, particularly regarding the role of external verifiers providing feedback. These verifiers include compiler tools and fine-tuned models.
>
> Let me elaborate further on this feedback. External feedback can be categorized into several types:
>
> 1. **Before training (during dataset construction):** External feedback is used to label correct and incorrect answers to construct the dataset. This requires external tools or models to annotate data accurately.
> 2. **After training:** External feedback supervises the model's responses to decide whether a response is correct and whether the model should generate further responses.
> 3. **During training (online training):** Feedback in this stage often involves a reward model that scores the model's outputs and dynamically updates the weights. However, training a reward model requires a pre-labeled dataset, and these reward models are typically designed for specific domains with specialized discriminative capabilities.
>
> Thus, external feedback, whether used before, during, or after training, has significant limitations. Firstly, constructing a discriminative model or dataset is costly and often domain-specific, requiring specialized feedback models. Secondly, it adds extra inference overhead, requiring more complex pipelines to evaluate and refine model responses.
>
> In contrast, our approach eliminates the need for external feedback at any stage, whether before or after training. This is because we recognize that in many scenarios, the boundaries of correctness are ambiguous, making external feedback both costly and less practical. Instead, we rely on self-improving datasets to enable the model to continually enhance its performance across all domains. This universality and simplicity represent our key contribution.

---

> ### Author Response · Authors · 2024-11-21
> **Answer to Q2**
>
> > Q2 compare with prior works on refinement methods or approaches with verifiers
>
> We appreciate your valuable suggestion. As noted by the reviewer, existing methods that rely on additional feedback for comparison are not applicable to all datasets and require domain-specific data construction, such as in mathematics. However, our approach does not involve fine-tuning on mathematical datasets, which introduces a potential unfairness in comparison. To address this, we have supplemented our experiments and found that even when additional feedback is leveraged, the improvements are similar. In contrast, for non-mathematical and non-reasoning tasks, such methods struggle to generalize, whereas our approach continues to achieve improvements.
>
> This highlights the primary contribution of our work: it does not require task-specific fine-tuning, which distinguishes it from previous self-refinement methods.
>
> Below are the experimental results comparing performance on mathematical tasks.
>
> |              Method              | GSM8K | MATH |
> | :------------------------------: | :---: | :--: |
> |      Self-refine iteration1      | 79.1  | 48.7 |
> |  with ground truth  iteration2   | 81.3  | 50.1 |
> |   w/o ground truth  iteration2   | 74.7  | 48.4 |
> |                                  |       |      |
> | Pair Self Correction iteration1  | 77.7  | 48.2 |
> | Pair Self Correction iteration2  | 80.1  | 49.5 |
> |                                  |       |      |
> | Reward-model Verifier iteration1 | 80.0  | 48.1 |
> | Reward-model Verifier iteration2 | 81.7  | 49.2 |
> |                                  |       |      |
> |         Ours iteration1          | 76.7  | 47.6 |
> |         Ours iteration2          | 79.9  | 48.9 |
>
> We compare our method to prior and baseline approaches, evaluating performance on GSM8K and MATH.
> Reward-model Verifier https://arxiv.org/abs/2110.14168 Training Verifiers to Solve Math Word Problems
> Self-refine：arXiv preprint arXiv:2303.17651 Self-refine: Iterative refinement with self-feedback
> Pair Self Correction：https://openreview.net/forum?id=hH36JeQZDaO Generating Sequences by Learning to Self-Correct
>
>
> > Performance similar to other refine-work on math
>
> Experimental results show that performance improvements across methods are modest, typically within 1-2%. The *self-refine* approach relies on ground-truth feedback for slight gains; without it, performance often deteriorates. Similarly, Pair Self-Correction and Reward-Model Verifier achieve comparable improvements but remain limited.
>
>
> > General Task Improvement
>
> In contrast, my method achieves similar improvements and demonstrates effectiveness in other non-mathematical and non-reasoning tasks where their approach fails to test and hard to construct training datasets.  This highlights the limitation of the self-refine structure in effectively improving model performance without external guidance.
>
> > Without external verifier or task-oriented fine-tuning
>
> - Our method does not require separate dataset construction or fine-tuning for each task, significantly reducing training costs.
> - The Reward Model approach requires constructing positive and negative samples for each task and training a reward model to supervise the generation process.
> - Similarly, the Pair Self-Correction method involves generating and labeling a large number of correct and incorrect sample pairs to achieve error correction capabilities. These processes are both costly and heavily dependent on specific domains.
>
> > Inference Costs lower
>
> - The **Reward Model Verifier** approach incurs high costs during data construction, as it requires best-of-n sampling and ground-truth answers to filter positive and negative examples. Additionally, its inference process is n times slower.
> - **Our method** requires only 2–3 iterations for inference and eliminates the need for external feedback, significantly reducing inference complexity.
>
> - Furthermore, our data construction process requires only a single training pass on a general dataset, allowing for performance improvements across all domains.
>
> > Coding task
>
> The data construction for coding tasks is significantly more complex, requiring the creation of a large number of positive and negative samples to train the verifiers. We plan to further evaluate and validate the effectiveness of our approach compared to previous methods in future work. This also highlights the higher applicability and feasibility of our method, as it does not rely on fine-tuning for specific domains but instead enhances self-improvement capabilities across a wide range of tasks.

---

> > ### Author Response · Authors · 2024-11-25
> > **More specific detail in Q2**
> >
> > We compare our method to prior and baseline approaches, evaluating performance on GSM8K and MATH.
> >
> > - **Self-refine**[1] does not require training but relies on standard answers to assist reasoning. It uses specific math-refine prompts to guide a base model in critiquing and revising its mistakes.
> >   - *"With ground truth"* refers to a setup where the model checks the correctness of its response only if the initial answer is wrong. If incorrect, the model generates a new response in a second iteration and stops as soon as the correct answer is predicted.
> >   - *"Without ground truth"* refers to a setup where the model always refines its answer without verifying correctness.
> >
> > > The specific math refine prompt is :
> > >
> > > if the answer is wrong:
> > >
> > > ```
> > > There is an error in the math above because of a lack of understanding of the question. What is the error? To find the error, go through semantically complete blocks of the code, and check if everything looks good.
> > > Let us go through the error and check step-by-step
> > > ```
> > >
> > > if the answer is right:
> > >
> > > ```
> > > looks good
> > > ```
> >
> >
> >
> >
> >
> > - **Pair Self Correction**[2] trains a single model to integrate both self-verify and generation capabilities.
> >
> > In the data construction phase, the original study used **Davinci (≈175B)** and **text-davinci-002 (≈175B)** models with few-shot prompting to construct the dataset. In our work, we substituted these models with the **Qwen2-70B** model for data generation (same as our methods settings).
> >
> > The method involves fine-tuning a large model using pairs of correct and incorrect mathematical solutions. This enables the model to learn to correct mistakes effectively. The model first diagnoses its initial output: if the self-diagnosis determines the answer is correct, no changes are made; if the answer is incorrect, the model revises the response accordingly.
> >
> >
> >
> > - **Reward-model Verifier** [3] requires training a Verifier to assess the correctness of model-generated answers. And inference on another base model.
> >
> > During the data construction phase, the original study used a 175B model to generate datasets, as explained in paper that larger models typically perform better on GSM8K(Here we use  **Qwen2-70B** model , the same settings with our methods). Labels were used to filter correct and incorrect responses. These inputs, along with their corresponding outputs and correctness judgments, were used to train the model to distinguish between correct and incorrect answers.
> >
> > The 175B model samples 100 answers for each training question. These answers were labeled as correct or incorrect (based on whether they reached the correct final answer) and used to train the reward model for 2 epochs.
> >
> > During inference, the base model model (here we use when 7b same as our methods) samples 100 responses for each question. If the initial response is incorrect, the model iteratively refines and generates new answers. This process is guided by the reward model’s scoring system, judging the output correctness.
> >
> >
> >
> > [1] Self-refine：Self-refine: Iterative refinement with self-feedback https://arxiv.org/abs/2303.17651
> >
> > [2] Pair Self Correction： Generating Sequences by Learning to Self-Correct https://arxiv.org/abs/2211.00053
> >
> > [3] Reward-model Verifier: Training Verifiers to Solve Math Word Problems https://arxiv.org/abs/2110.14168

---

> > ### Author Response · Authors · 2024-11-25
> > **Costs Comparison Table for each methods**
> >
> > We present a detailed comparison of our method with existing approaches, highlighting the key differences in cost, applicability, and performance across tasks such as GSM8K and general downstream tasks.
> >
> > ### **Comparison Table**
> >
> > | **Aspect**                                                  | **Our Method**                                               | **Self-Refine**                                              | **Pair Self-Correction**                                     | **Reward-Model Verifier**                                    |
> > | ----------------------------------------------------------- | ------------------------------------------------------------ | ------------------------------------------------------------ | ------------------------------------------------------------ | ------------------------------------------------------------ |
> > | **Iteration costs for Construct Data**                      | Requires only general domain queries; uses Qwen-2.7B and Qwen-70B to construct training datasets without external feedback. | No training required.                                        | Requires few-shot prompting with a 175B model, generating both correct and incorrect answers for training. The process involves at least two inferences per query. | Requires 100 inferences times with a 175B model to generate n-sampling responses for higher accuracy, filtering for correct and incorrect answers for dataset construction. |
> > | **Training Cost**                                           | 2 epochs on the base model.                                  | No training required.                                        | 2 epochs on the base model.                                  | 2 epochs on the verifier model.                              |
> > | **Additional Data Construction and Training for new Tasks** | Not required.                                                | Requires task-specific prompts with ground truth to assist evaluation. | Requires constructing task-specific datasets of correct and incorrect samples. Limited to tasks that make it easy to distinguish the correctness. | Requires constructing task-specific datasets of correct and incorrect samples. Limited to tasks that make it easy to distinguish the correctness. |
> > | **Need for External Feedback to Construct Data**            | Not required.                                                | Not applicable.                                              | Required correctness of answer.                              | Required correctness of answer.                              |
> > | **Number of Iterations During Inference**                   | 2–3 iterations.                                              | 2–3 iterations.                                              | 2–3 iterations.                                              | 2–3 iterations with 100 inference samples per iteration to improve response accuracy. |
> > | **Performance Improvement on GSM8K**                        | 2.9 /%.                                                      | 2.2  /%.                                                     | 2.4  /%.                                                     | 1.7  /%.                                                     |
> > | **Performance on Unseen Tasks**                             | Overall improvement of 3.9 points.                           | Requires task-specific prompts and ground truth for evaluation. | Requires additional dataset construction and fine-tuning, which is costly for subjective tasks, and impractical. | Requires additional dataset construction and fine-tuning, which is costly for subjective tasks, and impractical. |
> > | **Feedback During Inference**                               | Not required; generates an improved response regardless of initial correctness. | Requires feedback using ground truth to evaluate correctness. | Uses the model's self-verify to evaluate correctness. If incorrect, it iterates; if correct, it outputs the response. | Uses an additional verifier model to evaluate correctness. If incorrect, it iterates; if correct, it outputs the response. |
> >
> > ---
> >
> > ###

---

> > > ### Author Response · Authors · 2024-11-25
> > >
> > > ### **Advantages of Our Approach**
> > >
> > > #### 1. **Lower Cost**:
> > >
> > >    - Both *Pair Self-Correction* and *Reward-Model Verifier* rely on large-scale models (e.g., 175B parameters) to generate correct and incorrect answers for dataset construction. *Reward-Model Verifier* further requires 100 inference samples per query to improve the probability of a correct answer.
> > >    - In contrast, our method avoids stringent requirements for model accuracy during data generation. We use a strong model (Qwen-70B) and a weaker model (Qwen-2.7B) to generate data with only two inferences, allowing the model to learn self-improvement from the responses of both.
> > >
> > > #### 2. **Broader Applicability**:
> > >
> > >    - Existing approaches are limited to tasks with clear, objective correctness criteria, such as mathematical problems. They require task-specific datasets and fine-tuning, which are costly or infeasible for subjective or open-ended tasks.
> > >    - Our method, however, maintains strong adaptability across diverse tasks without additional fine-tuning. This enables the model to generalize effectively to unseen tasks.
> > >
> > > ---
> > >
> > > ### **Key Contributions of Our Work**
> > >
> > > 1. **Eliminating the Need for External Feedback**:
> > >    - Our method addresses the high cost of dataset construction by removing the reliance on external feedback for labeling correctness. Instead, the model learns self-improvement from general domain queries and the interactions between strong and weak models.
> > >    - Even without explicit correctness labels, our datasets enable the model to improve previous answers. Wilcoxon signed-rank tests (Appendix B.5) confirm that strong models consistently outperform weaker models, allowing the dataset to drive self-correction effectively.
> > >
> > > 2. **Practical Value Across Tasks**:
> > >    - Unlike task-oriented methods, which require fine-tuning for each specific task, our approach is highly practical for new or difficult-to-annotate tasks. The model undergoes a single fine-tuning step, after which it demonstrates self-improvement across all unseen tasks.
> > >
> > > 3. **Real-World Adaptability**:
> > >    - Our method reduces the reliance on clear correctness criteria, making it more accessible and effective in real-world scenarios where explicit answers are not always available. By fostering self-improvement, our approach ensures performance gains across diverse, unseen domains—a contribution that other methods cannot achieve.
> > >
> > > ---
> > >
> > >
> > >
> > > Our approach not only achieves comparable accuracy improvements in mathematical tasks but also significantly reduces inference and data construction costs. Furthermore, it extends beyond task-specific limitations, providing a scalable and adaptive solution for a wide range of real-world applications. This makes our method a more versatile and impactful contribution to self-corrective learning.

---

> ### Author Response · Authors · 2024-11-21
> **Answer Q4 Q5**
>
> > Q4 Effectiveness of the Thought-Mask Strategy
>
> **Effectiveness of the Thought-Mask Strategy** We appreciate your interest in the role of the thought-mask strategy.
>
> We also conducted experiments without the masking strategy, and the results were worse compared to when the mask was applied. This is because our mask covers the intermediate thought process, and we aim to guide the model's attention during training on how to refine its answer based on previous thoughts to arrive at a better response. Without the mask, the model also computes immature thoughts that arise after the question, but these are often incorrect, leading the model to lean toward providing a worse answer initially, which is unnecessary.
>
> In our case, the IFT method involves removing the thought process and directly fine-tuning using the input and the strong model's answer. This method helps demonstrate that our data construction format is not just a simple distillation of the large model's answering ability into a smaller model. We observe that basic distillation using general data does not lead to significant improvements on specific tasks and cannot enable the model to continuously self-improve. On the contrary, it validates that our method successfully triggers the model's ability to self-refine.
>
> |        | MMLU  | H-Eval | DROP  | Xsum  | GSM8k | Math  | ARC   | GPQA  | Wino  | Comm  | AVE   |
> | ------ | ----- | ------ | ----- | :---: | ----- | ----- | ----- | ----- | ----- | ----- | ----- |
> | UnMask   | 49.9% | 48.8%  | 17.2% | 41.1% | 67.4% | 42.1% | 56.3% | 19.4% | 61.8% | 45.1% | 44.9% |
> |        | 55.1% | 49.4%  | 19.5% | 40.9% | 71.1% | 43.8% | 62.9% | 20.8% | 58.4% | 50.3% | 47.2% |
> | Mask | 59.2% | 52.4%  | 19.0% | 45.9% | 76.7% | 47.6% | 58.6% | 23.2% | 66.4% | 47.9% | 49.7% |
> |        | 64.1% | 57.2%  | 21.2% | 49.8% | 79.9% | 48.6% | 62.7% | 25.6% | 66.4% | 54.9% | 53.0% |
>
>
>
> > Q5 fine-tuning setting
>
> | Hyperparameter/Description   | Training Values | Inference |
> | ---------------------------- | --------------- | --------- |
> | bf16                         | TRUE            | N/A       |
> | epochs                       | 2               | N/A       |
> | per device train batch size  | 1               | N/A       |
> | gpus                         | 2xH8100         | 2xH800    |
> | gradient accumulation steps  | 256             | N/A       |
> | learning rate                | 5e-5            | N/A       |
> | weight decay                 | 0               | N/A       |
> | warmup step                  | 1000            | N/A       |
> | learning rate scheduler type | cosine          | N/A       |
> |                              |                 |           |
> | model max length             | 2048            | 2048      |
> | temperature                  | N/A             | 0         |
> | top_p                        | N/A             | 1         |
> | max_new_tokens               | N/A             | 1000      |
>
> Our fine-tuning costs are significantly lower than other methods. Only around 24000 steps iteration with 40K data could achieve self-refinement capabilities across various domains. It is unnecessary to fine-tune and construct data for every specific task.
>
> Inference requires only two to three iterations, without needing feedback for supervision.
>
> However, other method use best-of-n to construct datasets and use ground-truth to confirm the correctness, which is much higher than our methods, typically 10 times larger when using best-of-10 to construct data.
>
> In contrast, other methods involve generating multiple answers during data construction and using external feedback mechanisms to distinguish correct and incorrect responses. This reliance on external feedback has two major drawbacks:
>
> 1. External feedback mechanisms are task-specific, making it difficult to ensure accurate feedback for more open-ended tasks.
> 2. Training a separate feedback model for supervision needs higher costs.
>
> PTR avoids these issues while maintaining lower training and inference costs with comparable or superior results.
>
>
> We sincerely thank the reviewer for their thorough evaluation and thoughtful feedback on our work. We have addressed each of the points raised, clarifying our contributions, methodologies, and experimental results in detail. Your comments have been instrumental in helping us refine our manuscript, and we greatly appreciate the opportunity to further improve our work. We look forward to any additional feedback or suggestions you may have and remain hopeful for a positive evaluation. Thank you again for your time and constructive insights!

---

> ### Comment · Reviewer_9DXQ · 2024-11-25
>
> Sorry for the late responses, and thanks for the authors' detailed responses!
>
> Most of things get clear for me, especially the novelty points of this paper compared to prior works.
>
> To be clear, as I understood, this paper introduce PTR framework that enables the models to self-refine with the proposed PTR dataset. This dataset is constructed through two weak and strong models, thus; the cost is relatively small to make this and the task is agnostic as it doesn't need the correctness label compared to previous verifier-based methods.
>
> Related to specific questions I asked:
>
> **Q1.** Thanks to clarifying the novel points.
>
> **Q2.** Thanks, and I think this experiment will make the contribution of this paper much clear. Though the model is not fine-tuned to specific datasets, which makes the authors' method suffer from lower initial performance.
>
> **Q3.** There is one followup question: other three methods show worse performance after refinement, because this model is not fine-tuned on downstream tasks? If they are fine-tuned on each tasks, will they possibly show good capability in refinement, as you mentioned in your answer to Q1?
>
> **Q4.** Thanks.
>
> **Q5.** Thanks.
>
> **Q6.** One followup question: can we say PTR does not use external feedback, given they used external strong model while constructing dataset? It's not feedback and very cheap, but could be quite vague?---Though I like the contributions of this method.
>
> ---
>
> I raised my score to 6. Sorry for misunderstanding the novel points for the paper at the first time. But I think the authors could enhance the writing and provide much clear explanation to compare to prior related works (also please add results and discussions related to Q2 and Q3).

---

> > ### Author Response · Authors · 2024-11-25
> > **Re to Re:Q2**
> >
> > Thank you for emphasising the contribution of this paper. We will further improve this part and include it in the revised version.

---

> > ### Author Response · Authors · 2024-11-25
> > **Re to Re:Q3**
> >
> > Yes, if the model is not fine-tuned for specific domains but rather fine-tuned on a general-domain dataset, it will lose accuracy in the specific domain at first. However, our method could refine the answer and gain a better result in refinement, while other methods could not. We will further investigate how to mitigate the negative impact of our method in future work.
> >
> > First of all, constructing datasets for each task is a very costly process, especially for subjective tasks where the quality and accuracy of answers cannot be simply defined. Creating such datasets for all tasks would require substantial external feedback and manual annotation, which is why previous work has not attempted to fine-tune models on each individual task (and it is practically impossible to do so). Instead, prior experiments were mostly conducted on specific domains.
> >
> > The first key contribution of our work is that it eliminates the need for external feedback to construct datasets, addressing the issue of high costs in dataset construction through external feedback. We believe that, in general, a strong model performs better than a weaker model, and we have demonstrated this through the Wilcoxon Singned-Rank Test analysis in Appendix B.5. Even though our dataset does not provide explicit feedback or labeling for correctness, the model can still learn to improve itself through our datasets and training approach. The experiments confirm that the model is indeed able to refine its previous answers.
> >
> > Furthermore, because fine-tuning each task is not a feasible solution, our method offers more practical value when we encounter new or difficult-to-annotate tasks.
> >
> > So another key contribution of our work is that we do not require fine-tuning for specific tasks in particular domains. Instead, we enable the model to refine itself, reducing the additional costs associated with fine-tuning. We cannot expect a model to fine-tune for every new scenario it encounters, especially in real-world applications. Our method requires only one fine-tuning step, after which the model can refine itself across all unseen tasks. This is a major contribution to our approach.

---

> > ### Author Response · Authors · 2024-11-25
> > **Re to Re:Q6**
> >
> > Indeed, our method does not use external feedback.
> >
> > Firstly, constructing a dataset requires leveraging a external model to perform reasoning, whether it's our method or other self-improvement methods. The difference lies in that other methods construct datasets within specific domains and require additional feedback to label the responses. For example, in mathematical tasks, responses need to be labeled as correct or incorrect to create sample pairs.
> >
> > This approach presents two main issues:
> >
> > 1. **Higher Cost**: Obtaining explicit labels for correctness increases the overall cost of dataset construction.
> > 2. **Limited Applicability**: Datasets can only be constructed for tasks with clear right or wrong answers, such as mathematical problems. But limited in more general downstream tasks.
> >
> > In contrast, our method uses an external strong model and a weak model to construct refinement data. While this approach may result in some instances where the correctness of responses is somewhat ambiguous, it allows us to create a dataset where, overall, responses are required to improve upon previous answers. This enables the model to learn how to enhance its answers without relying on explicit correctness labels.
> >
> > As a result, our method is more accessible and applicable to a broader range of real-world scenarios where clear-cut answers are not always available. By fostering the model's ability to self-improve, our approach ensures performance enhancements across various unseen domains, which other works could not achieve. This represents a more significant contribution.
> >
> > We really appreciate your recognition of our approach contribution!

---

> > ### Author Response · Authors · 2024-11-25
> > **Thank you for your valuable feedback and the increase in our score! We sincerely appreciate your support and generosity.**
> >
> > Thank you for your positive feedback and for the generous increase in our score. We sincerely appreciate your insightful and constructive comments, as well as the time and effort you invested in reviewing our work. Your thorough engagement and multiple rounds of communication have greatly contributed to improving the quality of our research. If you have any further questions or suggestions, please don’t hesitate to reach out.

---

> ### Comment · Reviewer_9DXQ · 2024-11-29
>
> Thanks for elaborating more the contributions and comparison table with prior works. I believe that these explanations would enhance the paper much stronger. I maintain my updated score at this point, but I'm quite positive for the acceptance of this paper.

---

> > ### Author Response · Authors · 2024-12-02
> > **Thanks for your comments and support.**
> >
> > Thank you very much for your thoughtful comments and continued support. We are pleased to hear that the additional explanations and the updated comparison table have strengthened the paper. Your positive feedback is greatly appreciated, and we are grateful for the time you have dedicated to reviewing our work. We will continue to make any necessary improvements based on your suggestions. If you have any further recommendations or questions, please feel free to reach out. Thank you again for your valuable input.

---

### Author Response · Authors · 2024-11-21
**General Response to All Reviewers**

We appreciate the valuable comments from the reviewers and are encouraged that reviewers found that our research question is of "vital importance" and "is an important research issue", that our method "has significant value in both research and applications", that our experiments "provide strong empirical evidence demonstrating the effectiveness of the proposed methods", that our paper is "well-organized", "well-organized and reasonable", "clearly structured and logically presented" and "well-written and easy to follow".
#  **Key Strengths Highlighted by Reviewers**

1. **novel framework and timely topic** (DjDQ, uz4P, YHbN)
2. **thorough discussion on related concepts and papers** (DjDQ)
3. **Well-organized, and clear writing and clear visuals** (9DXQ, DjDQ)
4. **Performance robustly and good, reasonable in application** (9DXQ, DjDQ, YHbN)
5. **Extensive experimental results and analysis** (9DXQ, uz4P)



# Key Updates
To address the reviewers’ concerns, we have made the following updates:

1. **Add comparison with some prior refinement works Experiments**
   1. Demostrate similar improvement in math within 3%.
   2. Our method even showed improvement in 10 unseen tasks without additional fine-tuning. While other methods fail to show generalizations on other tasks, and require high costs for constructing task-oriented datasets.(https://openreview.net/forum?id=pUbbLHjCPM&noteId=92XttfP1DM).
   3. Our method showed lower inference costs and training costs in comparison with other methods(https://openreview.net/forum?id=pUbbLHjCPM&noteId=Q0a1JD4Ioe).
2. **Add ablation experiment for fine-tuning hyperparameters Experiments**
   1. We explore suggestions for setting these parameters, which could simplify the tuning process and make our method more accessible for broader applications
   2. For tasks requiring high reasoning consistency and confidence distribution optimization, we recommend λ1=0.8,λ2=0.1,λ3=0.1. (https://openreview.net/forum?id=pUbbLHjCPM&noteId=7B8yF9ZHyQ)
   3. However, for most real-world applications, the simpler configuration of λ1=1.0,λ2=0.0,λ3=0.0 is also sufficient.
3. **Add ablation experiment for Thought-Mask Strategy Experiments**
   1. Showing that our Thought-Mask Strategy is necessary to perform better.(https://openreview.net/forum?id=pUbbLHjCPM&noteId=KC6x1Ak8ci)
4. **MMLU** **PRO****+ Experiments**: Added experiments on supervised finetuning for **MMLU PRO+** datasets to demonstrate generalizability (https://openreview.net/forum?id=pUbbLHjCPM&noteId=4YkxZGGW7X).
5. **Additional Explanations**:
   1. Clarified novel points of this paper over some prior refinement works (https://openreview.net/forum?id=pUbbLHjCPM&noteId=snHG4vh9QG).
   2. Clarified external feedback(https://openreview.net/forum?id=pUbbLHjCPM&noteId=snHG4vh9QG)
   3. Clarified fine-tuning setting (https://openreview.net/forum?id=pUbbLHjCPM&noteId=KC6x1Ak8ci)
   4. Clarified the fine-tuning hyperparameters (https://openreview.net/forum?id=pUbbLHjCPM&noteId=MBxeoIQDPb)
   5. Clarified the lower costs of our methods (https://openreview.net/forum?id=pUbbLHjCPM&noteId=Q0a1JD4Ioe)
   6. Clarified the PTR datasets（https://openreview.net/forum?id=pUbbLHjCPM&noteId=aBIAkvxvtw）
   7. Enhanced the explanation of figure 3,4, and the description of section "Description of PTR is not Knowledge Distillation"(https://openreview.net/forum?id=pUbbLHjCPM&noteId=EZ7pkNuozc).
   8. Clarified prompt details and filtering strategies (https://openreview.net/forum?id=pUbbLHjCPM&noteId=ovUYlbOX4m)




We hope these revisions and additional experiments address the reviewers' concerns and further substantiate the significance of our contributions.

---

> ### Author Response · Authors · 2024-11-26
> **Main contribution and novel points**
>
> As 2 reviewers asked about the costs and effeciency of our method compared with other prior works, we clairfy the main contribution and advantages here to show our novel points and strength.
>
> ### **Advantages of Our Approach**
>
> #### 1. **Lower Cost**:(https://openreview.net/forum?id=pUbbLHjCPM&noteId=XL2MjH2S3C)
>
> - Both *Pair Self-Correction* and *Reward-Model Verifier* rely on large-scale models (e.g., 175B parameters) to generate correct and incorrect answers for dataset construction. *Reward-Model Verifier* further requires 100 inference samples per query to improve the probability of a correct answer. (https://openreview.net/forum?id=pUbbLHjCPM&noteId=nRyyXFHi6D)
> - In contrast, our method avoids stringent requirements for model accuracy during data generation. We use a strong model (Qwen-70B) and a weaker model (Qwen-2.7B) to generate data with only two inferences, allowing the model to learn self-improvement from the responses of both.
>
> #### 2. **Broader Applicability**:
>
> - Existing approaches are limited to tasks with clear, objective correctness criteria, such as mathematical problems. They require task-specific datasets and fine-tuning, which are costly or infeasible for subjective or open-ended tasks. (https://openreview.net/forum?id=pUbbLHjCPM&noteId=nRyyXFHi6D)
> - Our method, however, maintains strong adaptability across diverse tasks without additional fine-tuning. This enables the model to generalize effectively to unseen tasks.
>
> ### **Key Contributions to Our Work**
>
> The primary contribution of our work is not in improving scores in specific domains, but in enabling the model to learn self-improvement through this training approach, allowing it to demonstrate progressive refinement across different tasks. Since our method is not task-oriented, the model trained with our approach can still refine itself when encountering new scenarios or tasks it has never seen before, unlike the base model, which either does not improve or makes incorrect corrections.
>
> We believe that, in real-world scenarios, even the most advanced models cannot achieve 100% accuracy, and there will always be instances where the model's responses are not ideal. Therefore, users may want the model to improve upon its previous answers. Existing methods mainly focus on specific domains, constructing task-oriented datasets to train models for improving self-refinement within those domains. However, in real life, tasks are not limited to just mathematics. Achieving self-improvement across a wider range of tasks is more challenging but also more valuable. Our method enables the model to demonstrate self-improvement across multiple tasks it has not been specifically trained on, which is the primary contribution of our work.
>
> 1. **Eliminating the Need for External Feedback**:
>    1. Our method addresses the high cost of dataset construction by removing the reliance on external feedback for labeling correctness. Instead, the model learns self-improvement from general domain queries and the interactions between strong and weak models.
>    2. Even without explicit correctness labels, our datasets still can enable the model to improve previous answers.
> 2. **Practical** **Value** **Across Tasks**:
>    1. Unlike task-oriented methods, which require fine-tuning for each specific task, our approach is highly practical for new or difficult-to-annotate tasks. The model undergoes a single fine-tuning step, after which it demonstrates self-improvement across all unseen tasks.
> 3. **Real-World Adaptability**:
>    1. Our method reduces the reliance on clear correctness criteria, making it more accessible and effective in real-world scenarios where explicit answers are not always available. By fostering self-improvement, our approach ensures performance gains across diverse, unseen domains — a contribution that other methods cannot achieve.

---

### Meta-Review · Area_Chair_ShmS · 2024-12-24

**Metareview:**

The paper proposes Progressive Thought Refinement (PTR), a two-phase framework enabling LLMs to self-refine responses iteratively. Phase one constructs thought-data via weak-strong model cooperation, and phase two employs weighted thought-mask fine-tuning for multi-iteration response enhancement. Experimental results affirm PTR's efficacy across ten tasks.

Reviewers overall consider the paper pertinent and timely in the ML community. Self-refinement is a crucial LLM improvement approach, and PTR's concept and performance, backed by extensive experiments and analysis, are promising. The two-phase methodology is well-validated on diverse datasets, though the strong model demand incurs cost but is still practical. However, concerns exist. The manuscript's lack of clarity impedes reader comprehension, and the method and hyperparameter settings are complex. Despite these, AC regards it as a valuable contribution in an essential direction and suggests acceptance as a poster.

**Additional Comments On Reviewer Discussion:**

The principal concerns put forward by the reviewers are as follows:

1. It is unclear what the main contributions and novel aspects of this paper are in comparison to previous refinement works.
2. The lack of clarity in the manuscript makes it difficult for readers to understand.
3. The method and hyperparameter settings are overly complex.
4. The improvement over simple prompting is minimal and saturation occurs rapidly.

Following the rebuttal discussions, the majority of issues and misunderstandings have been resolved. Consequently, most reviewers have a favorable view of this work and have adjusted their scores upwards. However, one reviewer remains unconvinced regarding the presentation and the experiment.

In my opinion, the authors have performed well in the rebuttal discussion. The generalization of the proposed method represents a crucial and novel contribution to LLM self-refinement.

---

### Decision · Program_Chairs · 2025-01-22

Accept (Poster)